# Optimized Tradeoffs for Private Prediction with Majority Ensembling

**Shuli Jiang**                                                                                               *shulij@andrew.cmu.edu*
*Robotics Institute, Carnegie Mellon University*
**Qiuyi (Richard) Zhang**                                                                                        *qiuyiz@google.com*
*Google DeepMind*
**Gauri Joshi**                                                                                                *gaurij@andrew.cmu.edu*
*Electrical and Computer Engineering, Carnegie Mellon University*

**Reviewed on OpenReview:** *https://openreview.net/forum?id=dwJluAakM8*

## Abstract

We study a classical problem in private prediction, the problem of computing an $(m\epsilon, \delta)$-differentially private majority of $K$ $(\epsilon, \Delta)$-differentially private algorithms for $1 \leq m \leq K$ and $1 > \delta \geq \Delta \geq 0$. Standard methods such as subsampling or randomized response are widely used, but do they provide optimal privacy-utility tradeoffs? To answer this, we introduce the Data-dependent Randomized Response Majority (DaRRM) algorithm. It is parameterized by a data-dependent noise function $\gamma$, and enables efficient utility optimization over the class of all private algorithms, encompassing those standard methods. We show that maximizing the utility of an $(m\epsilon, \delta)$-private majority algorithm can be computed tractably through an optimization problem for any $m \leq K$ by a novel structural result that reduces the infinitely many privacy constraints into a polynomial set. In some settings, we show that DaRRM provably enjoys a privacy gain of a factor of 2 over common baselines, with fixed utility. Lastly, we demonstrate the strong empirical effectiveness of our first-of-its-kind privacy-constrained utility optimization for ensembling labels for private prediction from private teachers in image classification. Notably, our DaRRM framework with an optimized $\gamma$ exhibits substantial utility gains when compared against several baselines.

## 1 Introduction

Differential privacy (DP) is a widely applied framework for formally reasoning about privacy leakage when releasing statistics on a sensitive database Erlingsson et al. (2014); Cormode et al. (2018). Differential privacy protects data privacy by obfuscating algorithmic output, ensuring that query responses look similar on adjacent datasets while preserving utility as much as possible Dwork et al. (2006).

Privacy in practice often requires aggregating or composing multiple private procedures that are distributed for data or training efficiency. For example, it is common to aggregate multiple private algorithmic or model outputs in methods such as boosting or calibration (Sagi & Rokach, 2018). In federated learning, model training is distributed across multiple edge devices. Those devices need to send local information, such as labels or gradients Konečný et al. (2016), to an aggregating server, which is often honest but curious about the local training data. Hence, the output from each model at an edge device needs to be privatized locally before being sent to the server. When translating from a local privacy guarantee to a centralized one, one needs to reason about the composition of the local privacy leakage Naseri et al. (2020). Therefore, we formally ask the following:

**Problem 1.1** (Private Majority Ensembling (Illustrated in Figure 1))**.** *Consider $K \geq 1$ $(\epsilon, \Delta)$-differentially private mechanisms $M_1, \ldots, M_K$ for $K$ odd. Given a dataset $\mathcal{D}$, each mechanism outputs a binary answer — that is, $M_i : \mathcal{D} \to \{0, 1\}$, $\forall i \in [K]$. Given a privacy **allowance** $1 \leq m \leq K$, $m \in \mathbb{R}$ and a failure probability*

*$\delta \geq \Delta \geq 0$, $\delta, \Delta \in [0, 1)$, how can one maximize the utility of an $(m\epsilon, \delta)$-differentially private mechanism $\mathcal{A}$ to compute the majority function $g(S_1, S_2, \ldots, S_K)$, where $S_i \sim M_i(\mathcal{D})$?*

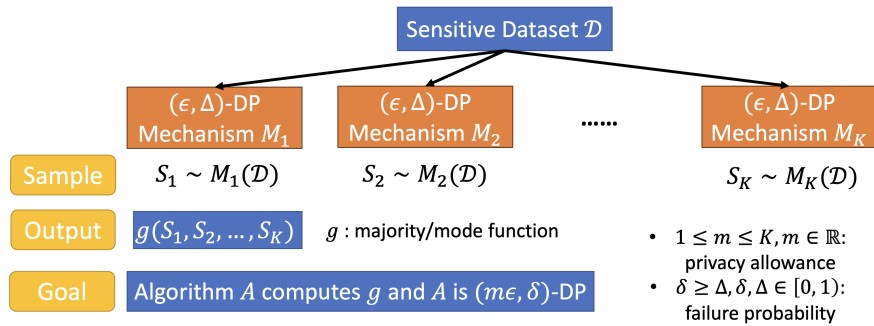

Figure 1: An illustration of the problem setting. The inputs are the dataset $\mathcal{D}$ and $K$ $(\epsilon, \Delta)$-differentially private mechanisms $M_1, \ldots, M_K$. One draws samples $S_i \sim M_i(\mathcal{D})$ and computes an aggregated output $g(S_1, \ldots, S_K)$ based on all observed samples. Our goal is to design a randomized algorithm $\mathcal{A}$ that approximately computes $g$ and is $(m\epsilon, \delta)$-differentially private for $1 \leq m \leq K$ and $\delta \geq \Delta \geq 0$. We focus on $g$ being the majority function .

The majority function $g$ is often used in private prediction, where one studies the privacy cost of releasing one prediction Dwork & Feldman (2018) and exploits the fact that releasing only the aggregated output on sharded models is significantly more private than releasing each prediction. For example, this occurs in semi-supervised knowledge transfer with private aggregated teacher ensembles (PATE) Papernot et al. (2017; 2018), in ensemble learning algorithms Jia & Qiu (2020); Xiang et al. (2018), machine unlearning Bourtoule et al. (2021), private distributed learning algorithms such as Stochastic Sign-SGD Xiang & Su (2023), and in ensemble feature selection Liu et al. (2018). Private prediction is also shown to be a competitive technique in data-adaptive settings, where the underlying dataset is changing slowly over time, to quickly adjust to online dataset updates Zhu et al. (2023). Furthermore, to address the large privacy loss of private prediction under the many-query regime, there has been recent works in everlasting private prediction that extends privacy guarantees with repeated, possibly infinite, queries without suffering a linear increase in privacy loss Naor et al. (2023); Stemmer (2024).

These works, however, rely often on the standard sensitivity analysis of $g$ to provide a private output and thus generally provide limited utility guarantees. This is because the maximum sensitivity of $g$ can be too pessimistic in practice, as observed in the problem of private hyperparameter optimization (Liu & Talwar, 2019). On the other hand, for private model ensembling, a naive way to bound privacy loss without restrictive assumptions is to apply simple composition (Theorem 2.2) or general composition (Theorem 2.3, a tighter version compared to advanced composition) to reason about the final privacy loss after aggregation. A black-box application of the simple composition theorem to compute $g$ would incur a $K\epsilon$ privacy cost in the pure differential privacy setting, that is, $\delta = 0$, or if one is willing to tolerate some failure probability $\delta$, general composition would yield a $O(\sqrt{K}\epsilon)$ privacy cost Kairouz et al. (2015). Thus, a natural baseline algorithm $\mathcal{A}$ that is $(m\epsilon, m\Delta)$-differentially private applies privacy amplification by subsampling and randomly chooses $m$ of the $K$ mechanisms to aggregate and returns the majority of the subsampled mechanisms. This technique is reminiscent of the subsampling procedure used for the maximization function $g$ (Liu & Talwar, 2019) or some general techniques for privacy amplification in the federated setting via shuffling (Erlingsson et al., 2019).

However, standard composition analysis and privacy amplication techniques can be suboptimal for computing a private majority, in terms of both utility and privacy. Observe that if there is a clear majority among the outputs of $M_1(\mathcal{D}), \ldots, M_K(\mathcal{D})$, one can add less noise. This is because each mechanism $M_i$ is $(\epsilon, \Delta)$-differentially private already, and hence, is less likely to change its output on a neighboring dataset by definition. This implies the majority outcome is unlikely to change based on single isolated changes in $\mathcal{D}$. Furthermore, composition theorems make two pessimistic assumptions: 1) the worst-case function $g$ and the dataset $\mathcal{D}$ are considered, and 2) all intermediate mechanism outputs $M_1(\mathcal{D}), \ldots, M_K(\mathcal{D})$ are released,

rather than just the final aggregate. Based on these observations, is it possible then to improve the utility of computing a private majority, under a fixed privacy loss?

## 1.1 Our Contributions

We give a (perhaps surprising) affirmative answer to the above question by using our novel data-dependent randomized response framework (DaRRM), which captures all private majority algorithms, we introduce a tractable noise optimization procedure that maximizes the privacy-utility tradeoffs. Furthermore, we can provably achieve a constant factor improvement in utility over simple subsampling by applying data-dependent noise injection when $M_i$'s are i.i.d. and $\delta = 0$. To our knowledge, this is the first of its work of its kind that gives a tractable utility optimization over the possibly infinite set of privacy constraints.

**Data-dependent Randomized Response Majority (DaRRM).** We generalize the classical Randomized Response (RR) mechanism and the commonly used subsampling baseline for solving Problem 1.1 and propose a general randomized response framework DaRRM (see Algorithm 1), which comes with a customizable noise function $\gamma$. We show that DaRRM actually captures all algorithms computing the majority whose outputs are at least as good as a random guess (see Lemma 3.3), by choosing different $\gamma$ functions.

**Designing $\gamma$ with Provable Privacy Amplification.** The choice of the $\gamma$ function in DaRRM allows us to explicitly optimize noise while trading off privacy and utility. Using structural observations, we show privacy amplification by a factor of 2 under mild conditions over applying simple composition in the pure differential privacy setting when the mechanisms $M_i$'s are i.i.d. (see Theorem 4.1).

**Finding the Best $\gamma$ through Dimension-Reduced Optimization.** We further exploit the generality of DaRRM by applying a novel optimization-based approach that applies constrained optimization to find a data-dependent $\gamma$ that maximizes some measure of utility. One challenge is that there are infinitely many privacy constraints, which are necessary for DaRRM with the optimized $\gamma$ to satisfy the given privacy loss. We show that we can reformulate the privacy constraints, which are infinite dimensional, to a finite polynomial-sized constraint set, allowing us to efficiently constrain the optimization problem to find the best $\gamma$, even for approximate differential privacy (see Lemma 5.1). Empirically, we show that with a small $m$ and $\epsilon$, the optimized $\gamma$ (see $\gamma_{opt}$ in Figure 2) achieves the best utility among all $\gamma$ functions, even compared to the subsampling and the data-independent baseline. To our knowledge, this is the first utility maximization algorithm that optimizes over all private algorithms by constrained optimization with dimension reduction.

**Experiments.** In downstream tasks, such as semi-supervised knowledge transfer for private image classification, we compare our DaRRM with an optimized $\gamma$ to compute the private label majority from private teachers against PATE Papernot et al. (2018), which computes the private label majority from non-private teachers. We fix the privacy loss of the output of both algorithms to be the same and find that when the number of teachers $K$ is small, DaRRM indeed has a higher utility than PATE, achieving 10%-15% and 30% higher accuracy on datasets `MNIST` and `Fashion-MNIST`, respectively.

## 2 Background

### 2.1 Related Work

**Private Composition.** Blackbox privacy composition analysis often leads to pessimistic utility guarantees. In the blackbox composition setting, one can do no better than the $O(K\epsilon)$ privacy analysis for pure differential privacy Dwork et al. (2014). For approximate differential privacy, previous work has found optimal constants for advanced composition by reducing to the binary case of hypothesis testing with randomized response; and optimal tradeoffs between $\epsilon, \delta$ for black box composition are given in Kairouz et al. (2015), where there could be a modest improvement 20%.

Thus, for specific applications, previous work has turned to white-box composition analysis for improved utility. This includes, for example, moment accountant for private SGD Abadi et al. (2016) and the application of contractive maps in stochastic convex optimization Feldman et al. (2018). For the specific case of model ensembles, Papernot et al. (2018) shows a data-dependent privacy bound that vanishes as the probability of

disagreement goes to 0. Their method provides no utility analysis but they empirically observed less privacy loss when there is greater ensemble agreement.

When $g$ is the maximization function, some previous work shows that an approximately maximum value can be outputted with high probability while incurring $O(\epsilon)$ privacy loss, independently of $K$. Liu & Talwar (2019) proposed a random stopping mechanism for $m = 1$ that draws samples uniformly at random from $M_i(\mathcal{D})$ at each iteration. In any given iteration, the sampling halts with probability $\gamma$ and the final output is computed based on the samples collected until that time. This leads to a final privacy cost of only $3\epsilon$ for the maximization function $g$, which can be improved to $2\epsilon$ (Papernot & Steinke, 2022). In addition to the aforementioned works, composing top-k and exponential mechanisms also enjoy slightly improved composition analysis via a bounded-range analysis Durfee & Rogers (2019); Dong et al. (2020).

**Bypassing the Global Sensitivity.** To ensure differential privacy, it is usually assumed the query function $g$ has bounded global sensitivity — that is, the output of $g$ does not change much on *any* adjacent input datasets differing in one entry. The noise added to the output is then proportional to the global sensitivity of $g$. If the sensitivity is large, the output utility will thus be terrible due to a large amount of noises added. However, the worst case global sensitivity can be rare in practice, and this observation has inspired a line of works on designing private algorithms with data-dependent sensitivity bound to reduce the amount of noises added.

Instead of using the maximum global sensitivity of $g$ on any dataset, the classical Propose-Test-Release framework of Dwork Dwork & Lei (2009) uses a local sensitivity value for robust queries that is tested privately and if the sensitivity value is too large, the mechanism is halted before the query release. The halting mechanism incurs some failure probability but deals with the worst-case sensitivity situations, while allowing for lower noise injection in most average-case cases.

One popular way to estimate average-case sensitivity is to use the Subsample-and-Aggregate framework by introducing the notion of *perturbation stability*, also known as *local sensitivity* of a function $g$ on a dataset $\mathcal{D}$ Thakurta & Smith (2013); Dwork et al. (2014), which represents the minimum number of entries in $\mathcal{D}$ needs to be changed to change $g(\mathcal{D})$. One related concept is *smooth sensitivity*, a measure of variability of $g$ in the neighborhood of each dataset instance. To apply the framework under *smooth sensitivity*, one needs to privately estimate a function's local sensitivity $L_s$ and adapt noise injection to be order of $O(\frac{L_s}{\epsilon})$, where $L_s$ can often be as small as $O(e^{-n})$, where $n = |\mathcal{D}|$, the total dataset size Nissim et al. (2007). Generally, the private computation of the smooth sensitivity of a blackbox function is nontrivial but is aided by the Subsample and Aggregate approach for certain functions.

These techniques hinge on the observation that a function with higher stability on $\mathcal{D}$ requires less noise to ensure worst case privacy. Such techniques are also applied to answer multiple online functions/queries in model-agnostic learning Bassily et al. (2018). However, we highlight two key differences in our setting with a weaker stability assumption. First, in order to estimate the *perturbation stability* of $g$ on $\mathcal{D}$, one needs to downsample or split $\mathcal{D}$ into multiple blocks Thakurta & Smith (2013); Dwork et al. (2014); Bassily et al. (2018), $\hat{\mathcal{D}}_1, \ldots, \hat{\mathcal{D}}_B$ , and estimate the *perturbation stability* based on the mode of $g(\hat{\mathcal{D}}_1), \ldots, g(\hat{\mathcal{D}}_B)$. This essentially reduces the amount of change in the output of $g$ due to a single entry in $\mathcal{D}$, with high probability and replaces the hard-to-estimate *perturbation stability* of $g$ with an easy-to-compute *perturbation stability* of the mode. Such a notion of stability has also been successfully applied, along with the sparse vector technique, for model-agnostic private learning to handle exponentially number of queries to a model Bassily et al. (2018). Note that in these cases, since a private stochastic test is applied, one cannot achieve pure differential privacy Dwork et al. (2014). In practice, e.g. federated learning, however, one does not have direct access to $\mathcal{D}$, and thus it is impractical to draw samples from or to split $\mathcal{D}$. Second, to ensure good utility, one relies on a key assumption, i.e. the *subsampling stability* of $g$, which requires $g(\hat{\mathcal{D}}) = g(\mathcal{D})$ with high probability over the draw of subsamples $\hat{\mathcal{D}}$.

Although our intuition in designing DaRRM also relies on the stability of the mode function $g$, previous usage of stability to improve privacy-utility tradeoffs, e.g., propose-test-release Vadhan (2017); Dwork et al. (2014), requires the testing of such stability, based on which one adds a larger (constant) noise $\gamma$. This can still lead to adding redundant noise in our case.

**Optimal Randomized Response.** Holohan et al. (2017) and Kairouz et al. (2015) show that the classical Randomized Response (RR) mechanism with a constant probability of faithfully revealing the true answer is optimal in certain private estimation problems. Our proposed DaRRM framework and our problem setting is a generalized version of the ones considered in both Holohan et al. (2017) and Kairouz et al. (2015), which not only subsumes RR but also enables a data-dependent probability, or noise addition.

While RR with a constant probability can be shown optimal in problems such as private count queries or private estimation of trait possession in a population, it is not optimal in other problems, such as private majority ensembling, since unlike the former problems, changing one response of the underlying mechanisms does not necessarily change the output of the majority. To explicitly compute the minimum amout of noise required, one needs the output distributions of the underlying mechanisms but this is unknown. To resolve this, our proposed DaRRM framework adds the amount of noise dependent on the set of observed outcomes from the underlying private mechanisms, $\mathcal{S}$, which is a random variable of the dataset and is hence a proxy. This enables DaRRM to calibrate the amount of noise based on whether the majority output is likely to change. The amount of noise is automatically reduced when the majority output is not likely to change.

Second, Holohan et al. (2017) and Kairouz et al. (2015) both consider a special case in our setting where all $K$ private mechanisms are i.i.d., while our approach focuses on the more general setting where each private mechanism can have a different output distribution.

**Learning A Good Noise Distribution.** There have been limited works that attempt to derive or learn a good noise distribution that improves the utility. For deep neural networks inference, Mireshghallah et al. (2020) attempts to learn the best noise distribution to maximizing utility subject to an entropy Lagrangian, but no formal privacy guarantees were derived. For queries with bounded sensitivity, Geng & Viswanath (2015) demonstrate that the optimal noise distribution is in fact a staircase distribution that approaches the Laplacian distribution as $\epsilon \to 0$.

**Private Prediction.** Instead of releasing a privately trained model as in private learning, private prediction hides the models and only releases private outputs. Private prediction has been shown as a practical alternative compared to private learning, as performing private prediction is much easier compared to private learning on a wide range of tasks Dwork & Feldman (2018); Naor et al. (2023); van der Maaten & Hannun (2020). Although a privately trained model can make infinitely many predictions at the inference time without incurring additional privacy loss, since differential privacy is closed under post-processing, it has been shown recently that it is indeed possible to make infinitely many private predictions Naor et al. (2023) with a finite privacy loss for specific problems.

## 2.2 Preliminaries

We first introduce the definition of differential privacy, simple composition and general composition as follows. The general composition Kairouz et al. (2015) gives a near optimal and closed-form bound on privacy loss under adaptive composition, which improves upon advanced composition Dwork et al. (2014).

**Definition 2.1** (Differential Privacy (DP) Dwork et al. (2014)). *A randomized mechanism $\mathcal{M} : \mathcal{D} \to \mathcal{R}$ with a domain $\mathcal{D}$ and range $\mathcal{R}$ satisfies $(\epsilon, \delta)$-differential privacy for $\epsilon, \delta \geq 0$ if for any two* **adjacent datasets** *$\mathcal{D}, \mathcal{D}'$ and for any subset of outputs $S \subseteq \mathcal{R}$ it holds that $\Pr[\mathcal{M}(\mathcal{D}) \in S] \leq e^\epsilon \Pr[\mathcal{M}(\mathcal{D}') \in S] + \delta$. $\delta = 0$ is often called pure differential privacy; while $\delta > 0$ is often called approximate differential privacy.*

**Theorem 2.2** (Simple Composition Dwork et al. (2014)). *For any $\epsilon > 0$ and $\delta \in [0, 1]$, the class of $(\epsilon, \delta)$-differentially private mechanisms satisfy $(k\epsilon, k\delta)$-differential privacy under $k$-fold adaptive composition.*

**Theorem 2.3** (General Composition (Theorem 3.4 of Kairouz et al. (2015))). *For any $\epsilon > 0, \delta \in [0, 1]$ and $\delta' \in (0, 1]$, the class of $(\epsilon, \delta)$-differentially private mechanisms satisfies $(\epsilon', 1 - (1 - \delta)^k (1 - \delta'))$-differential privacy under $k$-fold adaptive composition for*

$$\epsilon' = \min \left\{ k\epsilon, \frac{(e^\epsilon - 1)\epsilon k}{e^\epsilon + 1} + \epsilon \sqrt{2k \log(e + \frac{\sqrt{k\epsilon^2}}{\delta'})}, \frac{(e^\epsilon - 1)\epsilon k}{e^\epsilon + 1} + \epsilon \sqrt{2k \log(1/\delta')} \right\}$$

We then formalize the error and utility metric in our problem as follows:

**Definition 2.4** (Error Metric and Utility Metric). *For the problem setting in Definition 1.1, let the observed (random) outcomes set be $\mathcal{S} = \{S_1, .., S_k\}$, where $S_i \sim M_i(\mathcal{D})$. For a fixed $\mathcal{D}$, we define the error of an algorithm $\mathcal{A}$, i.e., $\mathcal{E}(\mathcal{A})$, in computing the majority function $g$ as the Total Variation (TV) distance between $g(\mathcal{S})$ and $\mathcal{A}(\mathcal{D})$. Specifically,*

$$\mathcal{E}(\mathcal{A}) = \mathcal{D}_{TV}(g(\mathcal{S}) \parallel \mathcal{A}(\mathcal{D})) = |\Pr[\mathcal{A}(\mathcal{D}) = 1] - \Pr[g(\mathcal{S}) = 1]|$$

*and the utility is defined as $1 - \mathcal{E}(\mathcal{A})$.*

**Notation.** Throughout the paper, we use the same notations defined in Problem 1.1 and Definition 2.4. Furthermore, let $\mathcal{D}$ and $\mathcal{D}'$ to denote a pair of adjacent datasets with one entry being different. Also, let $p_i = \Pr[M_i(\mathcal{D}) = 1]$ and $p_i' = \Pr[M_i(\mathcal{D}') = 1]$, $\forall i \in [K]$. We omit the subscript $i$ when all $p_i$'s or $p_i'$'s are equal. $\mathbb{I}\{\cdot\}$ denotes the indicator function and $[K] = \{1, 2, \ldots, K\}$. For the purpose of analysis, let $\mathcal{L}(\mathcal{D}) = \sum_{i=1}^{K} M_i(\mathcal{D}) \in \{0, 1, \ldots, K\}$, i.e. the (random) sum of all observed outcomes on dataset $\mathcal{D}$. $\mathcal{D}$ is omitted when the context is clear. Unless specified, we use the noise function $\gamma : \{0, 1, \ldots, K\} \to [0, 1]$ as input to our algorithms to calibrate the probabilistic noise injection. Unless specified, the privacy allowance $m \in \mathbb{R}$.

## 3 Private Majority Algorithms

The very first approach to consider when solving private majority ensembling (Problem 1.1), since the output is binary, is the classical Randomized Response (RR) mechanism Dwork et al. (2014), where one flips a biased coin with a *constant* probability $p_{const} \in [0, 1]$. If the coin lands on head with probability $p_{const}$, output the true majority base on $K$ samples; if not, then simply output a noisy random answer. However, to make the output $(m\epsilon, \delta)$-differential private, the success probability $p_{const}$ can be at most $O(\frac{m}{K})$ (or $O(\frac{m}{\sqrt{K}})$) when $\delta = 0$ (or $\delta > 0$) (see Appendix A.1), which is too small for any reasonable utility.

The key observation for improved utility is that the probability of success should not be a *constant*, but should depend on the *unpublished* set of observed outcomes from the mechanisms $\mathcal{S}$. If we see many 1's or 0's in $\mathcal{S}$, then there should be a clear majority even on adjacent datasets. On the other hand, if we see about half 1's and half 0's, this means the majority is highly volatile to data changes, which implies we need more noise to ensure privacy. In summary, if we can calibrate the success probability based on $\mathcal{S}$ to smoothly increase when there is a clear majority, we can improve the utility without affecting privacy.

**Subsampling.** One natural baseline is outputting the majority of $m$ out of $K$ randomly subsampled mechanisms (without replacement), given a privacy allowance $m \in [K]$. Suppose $\delta \geq m\Delta$, the privacy loss of the aggregated output can be reasoned through simple composition or general composition. Interestingly, we show outputting the majority of $m$ out of $K$ subsampled mechanisms corresponds to RR with a *non-constant* probability $p_\gamma = \gamma_{Sub}(\mathcal{L}(\mathcal{D}))$, which is set by a polynomial function $\gamma_{Sub} : \{0, \ldots, K\} \to [0, 1]$ based on the sum of observed outcomes $\mathcal{L}(\mathcal{D})$ in Lemma 3.1 (see a full proof in Appendix A.2). Intuitively, subsampling may be seen as implicitly adding noise by only outputting based on a randomly chosen subset of the mechanisms; therefore this implicit noise is inherently *data-dependent* on $\mathcal{L}(\mathcal{D})$.

**Lemma 3.1.** *Consider Problem 1.1, with the privacy allowance $m \in [K]$. Consider the data-dependent algorithm that computes $\mathcal{L}(\mathcal{D})$ and then applies RR with probability $p_\gamma$. If $p_\gamma = \gamma_{Sub}(l)$, where $l \in \{0, 1, \ldots, K\}$ is the value of $\mathcal{L}(\mathcal{D})$, i.e., the (random) sum of observed outcomes on dataset $\mathcal{D}$, and $\gamma_{Sub} : \{0, 1, \ldots, K\} \to [0, 1]$ is*

$$\gamma_{Sub}(l) = \gamma_{Sub}(K - l) = \begin{cases} 1 - 2\sum_{j=\frac{m+1}{2}}^{m} \frac{\binom{l}{j}\binom{K-l}{m-j}}{\binom{K}{m}} & \text{if } m \text{ is odd} \\ 1 - 2\sum_{j=\frac{m}{2}+1}^{m} \frac{\binom{l}{j}\binom{K-l}{m-j}}{\binom{K}{m}} - \frac{\binom{l}{\frac{m}{2}}\binom{K-l}{\frac{m}{2}}}{\binom{K}{m}} & \text{if } m \text{ is even} \end{cases}$$

*then the majority of $m$ out of $K$ subsampled mechanisms without replacement and the output of our data-dependent RR algorithm have the same distribution.*

One thing special about subsampling is that when $m = 1$, it indeed results in the optimal error, which we show in Lemma 3.2 as follows. See a full proof in Appendix A.3. Note that when $m = 1$, subsampling outputs

a majority of 1 with probability exactly $\frac{1}{K} \sum_{i=1}^{K} p_i$. This lower bound only applies to the case when $m = 1$, since when $m > 1$, the probability of subsampling outputting a majority of 1 is not necessary $\frac{1}{K} \sum_{i=1}^{K} p_i$.

**Lemma 3.2** (Lower Bound on Error when $m = 1$). *Let $\mathcal{A}$ be an $(\epsilon, \delta)$-differentially private algorithm, where $\epsilon \in (0, \frac{1}{2})$ and $\delta \in [0, \frac{1}{2})$, that computes the majority of $K$ $(\epsilon, \delta)$-differentially private mechanisms $M_1, \ldots, M_K$, where $M_i : \mathcal{D} \to \{0, 1\}$ on dataset $\mathcal{D}$ and $\Pr[M_i(\mathcal{D}) = 1] = p_i, \forall i \in [K]$. Then, the error $\mathcal{E}(\mathcal{A}) \geq |\Pr[g(\mathcal{S}) = 1] - \frac{1}{K} \sum_{i=1}^{K} p_i|$, where $g(\mathcal{S})$ is the probability of the true majority output being 1 as defined in Definition 1.1.*

---

**Algorithm 1** DaRRM$(\cdot)$: Data-dependent Randomized Response Majority

---

1: Input: $K$ $(\epsilon, \Delta)$-DP mechanisms $\{M_i\}_{i=1}^{K}$, noise function $\gamma : \{0, 1\}^{K+1} \to [0, 1]$ (in our specific setting $\gamma : \{0, 1, \ldots, K\} \to [0, 1]$), dataset $\mathcal{D}$, privacy allowance $1 \leq m \leq K$, failure probability $\delta \geq \Delta \geq 0$
2: Output: $(m\epsilon, \delta)$-DP majority vote of $\{M_i\}_{i=1}^{K}$
3: $\mathcal{S} = \{S_1, .., S_K\}$, where $S_i \sim M_i(\mathcal{D})$
4: $\mathcal{L} = \sum_{i=1}^{K} S_i$
5: Set probability $p_\gamma \leftarrow \gamma(\mathcal{S})$ (in our setting $p_\gamma \leftarrow \gamma(\mathcal{L})$)
6: Flip the $p_\gamma$- biased coin
7: **if** Head (with probability $p_\gamma$) **then**
8:     Output $\mathbb{I}\{\frac{1}{K}\mathcal{L} \geq \frac{1}{2}\}$
9: **else**
10:     Output 0/1 with equal probability
11: **end if**

---

**Data-dependent Randomized Response (DaRRM).** Does subsampling give optimal utility when $m > 1$? Inspired by the connection between RR and subsampling, we propose Data-dependent Randomized Response Majority (DaRRM) in Algorithm 1, to study optimizing privacy-utility tradeoffs in private majority ensembling. In particular, DaRRM has a *non-constant* success probability $p_\gamma$ that is set by a parameterized noise function $\gamma$, which in turn depends on the set of observed outcomes $\mathcal{S} = \{S_1, \ldots, S_K\}$. In fact, we can show that DaRRM is general: any *reasonable* algorithm $\mathcal{A}$, name one whose output is at least as good as a random guess, can be captured by the DaRRM framework in Lemma 3.3 (see a full proof in Appendix A.4). We denote DaRRM instantiated with a specific noise function $\gamma$ by DaRRM$_\gamma$.

**Lemma 3.3** (Generality of DaRRM). *Let $\mathcal{A}$ be any randomized algorithm to compute the majority function $g$ on $\mathcal{S}$ such that for all $\mathcal{S}$, $\Pr[\mathcal{A}(\mathcal{S}) = g(\mathcal{S})] \geq 1/2$ (i.e. $\mathcal{A}$ is at least as good as a random guess). Then, there exists a a general function $\gamma : \{0, 1\}^{K+1} \to [0, 1]$ such that if one sets $p_\gamma$ by $\gamma(\mathcal{S})$ in DaRRM, the output distribution of DaRRM$_\gamma$ is the same as the output distribution of $\mathcal{A}$.*

**Designing the $\gamma$ Function.**    With the DaRRM framework, we ask: how to design a good $\gamma$ function that maximizes the utility? First, we introduce two characteristics of $\gamma$ that do not affect the utility, while simplifying the analysis and the empirical optimization:

   (a) **A function of the sum of observed samples**: Since the observed samples set $\mathcal{S}$ is a permutation-invariant set, a sufficient statistic that captures the full state of $\mathcal{S}$ is $\mathcal{L} = \sum_{i=1}^{K} S_i$, the sum of observed outcomes. This allows us to reduce $\gamma(\mathcal{S}) = \gamma(\mathcal{L})$. *Hence, in the rest of the paper, we focus on $\gamma : \{0, 1, \ldots, K\} \to [0, 1]$.*

   (b) **Symmetric around $\frac{K}{2}$**: If $\gamma$ is asymmetric, we can symmetrize by reflecting one region about $\frac{K}{2}$ and achieve better or equal expected utility, where the utility is summed over symmetric distributions of $p_i$.

Note that $\gamma_{Sub}$ satisfies both characteristics. Now, recall $\mathcal{L}(\mathcal{D})$ and $\mathcal{L}(\mathcal{D}')$ are the sum of observed outcomes on adjacent datasets $\mathcal{D}$ and $\mathcal{D}'$. Also, recall $p_i = \Pr[M_i(\mathcal{D}) = 1]$ and $p_i' = \Pr[M_i(\mathcal{D}') = 1]$ are the output probabilities of the mechanism $M_i$ on $\mathcal{D}, \mathcal{D}'$. To design a good noise function $\gamma$ in DaRRM, we start by deriving conditions for a $\gamma$ function such that DaRRM$_\gamma$ is $(m\epsilon, \delta)$-differentially private in Lemma 3.4 (see a full proof in Appendix A.5).

**Lemma 3.4** ($\gamma$ privacy condition). *Consider using DaRRM (Algorithm 1) to solve Problem 1.1, let $\alpha_l = \Pr[\mathcal{L}(\mathcal{D}) = l]$ and $\alpha'_l = \Pr[\mathcal{L}(\mathcal{D}') = l]$, where $\mathcal{D}$ and $\mathcal{D}'$ are adjacent datasets and $l \in \{0, \ldots, K\}$. For a noise function $\gamma : \{0, 1, \ldots, K\} \rightarrow [0, 1]$ such that $\gamma(l) = \gamma(K - l), \forall l$, DaRRM$_\gamma$ is $(m\epsilon, \delta)$-differentially private if and only if for all $\alpha_l, \alpha'_l$, the following holds,*

$$f(p_1, \ldots, p_K, p'_1, \ldots, p'_K; \gamma) \leq e^{m\epsilon} - 1 + 2\delta \tag{1}$$

*where $f$ is called the **privacy cost objective** and*

$$f(p_1, \ldots, p_K, p'_1, \ldots, p'_K; \gamma) := \sum_{l=0}^{\frac{K-1}{2}} (e^{m\epsilon}\alpha'_l - \alpha_l) \cdot \gamma(l) + \sum_{l=\frac{K+1}{2}}^{K} (\alpha_l - e^{m\epsilon}\alpha'_l) \cdot \gamma(l)$$

# 4 Provable Privacy Amplification

We theoretically demonstrate that privacy is provably amplified under improved design of $\gamma$ in our DaRRM framework. Specifically, we show when the mechanisms are i.i.d. and $\delta = 0$, we gain privacy amplification by a factor of 2 compared to the naïve subsampling baseline by carefully designing $\gamma$.

**Theorem 4.1** (Provable Privacy Amplification by 2). *Consider using DaRRM (Algorithm 1) to solve Problem 1.1, with i.i.d. mechanisms $\{M_i\}_{i=1}^{K}$, i.e., $p_i = p$, $p'_i = p'$, $\forall i \in [K]$, the privacy allowance $m \in [K]$ and $\delta = \Delta = 0$. Let the noise function $\gamma : \{0, 1, \ldots, K\} \rightarrow [0, 1]$ be that: if $m \geq \frac{K+1}{2}$, $\gamma(l) = 1$ and if $m \leq \frac{K-1}{2}$,*

$$\gamma(l) = \begin{cases} 1 - 2h(l) & \forall l \leq \frac{K-1}{2} \\ 2h(l) - 1 & \forall l \geq \frac{K+1}{2} \end{cases}$$

*where $h(l) = \sum_{i=m}^{2m-1} \frac{\binom{l}{i}\binom{K-l}{2m-1-i}}{\binom{K}{2m-1}}$, then DaRRM$_\gamma$ is $m\epsilon$-differentially private.*

**Interpretation.** First, when $m \leq \frac{K-1}{2}$ is small, the $\gamma(l)$ in Theorem 4.1 corresponds to outputting the majority based on subsampling $2m - 1$ outcomes, from Lemma 3.1. However, the subsampling baseline, whose privacy loss is reasoned through simple composition, would have indicated that one can only output the majority based on $m$ outcomes, therefore implying a 2x privacy gain. When $m \geq \frac{K+1}{2}$, the above theorem indicates that we can set a constant $\gamma = 1$, which implies we are optimally outputting the true majority with no noise while still surprisingly ensuring $m\epsilon$ privacy.

**Intuition.** This 2x privacy gain is intuitively possible because the majority is only dependent on half of the mechanisms' outputs, therefore the privacy leakage is also halved. To see this, we start by analyzing the privacy cost objective in Eq. 31, where with a careful analysis of its gradient, we show that the maximum indeed occurs $(p^*, p'^*) = (0, 0)$ when $\gamma$ satisfies certain conditions. Now, when $(p^*, p'^*) \rightarrow 0$, note that the probability ratio of outputting 1 with $2m-1$ outcomes is approximately $e^{m\epsilon}$, where dependence on $m$ follows because the probability of outputting 1 is dominated by the probability that exactly $m$ mechanisms output 1. To rigorize this, we derive sufficient conditions for $\gamma$ functions that satisfy $\max_{(p,p')} f(p, p'; \gamma) = f(0, 0; \gamma) \leq e^{m\epsilon} - 1$ as indicated by Lemma 3.4, to ensure DaRRM to be $m\epsilon$-differentially private and a more detailed overview and the full proof can be found in Appendix B.

# 5 Optimizing the Noise Function $\gamma$ in DaRRM

Theoretically designing $\gamma$ and extending privacy amplification results to the $\delta > 0$ case is difficult and it is likely that our crafted $\gamma$ is far from optimal. On the other hand, one can optimize for such $\gamma^*$ that maximizes the utility but this involves solving a "Semi-infinite Programming" problem, due to the infinitely many privacy constraints, which are the constraints in the optimization problem necessary to ensure DaRRM with the optimized $\gamma$ satisfy a given privacy loss. Solving a "Semi-infinite Programming" problem in general is non-tractable, but we show that in our specific setting this is in fact tractable, proposing a novel learning

approach based on DaRRM that can optimize the noise function to maximize the utility. To the best of our knowledge, such optimization, presented as follows, is the first of its kind:

$$\min_{\gamma \in [0,1]^{K+1}} \mathbb{E}_{p_1, p_2, \ldots, p_K \sim \mathcal{T}}[\mathcal{E}(\text{DaRRM}_\gamma)] \tag{2}$$

$$\text{s.t.} \max_{\{(p_i, p_i') \in \mathcal{F}_i\}_{i=1}^K} f(p_1, \ldots, p_K, p_1', \ldots, p_K'; \gamma) \leq e^{m\epsilon} - 1 + 2\delta \tag{3}$$

$$\gamma(l) = \gamma(K - l), \forall l \in \{0, 1, \ldots, K\}$$

where $f$ is the privacy cost objective as defined in Lemma 3.4, $\mathcal{F}_i$ is the feasible region where $(p_i, p_i')$ lies due to each mechanism $M_i$ being $\epsilon$-differentially private. Observe that since $\gamma$ is symmetric around $\frac{K}{2}$, we only need to optimize $\frac{K+1}{2}$ variables instead of $K + 1$ variables. $\mathcal{T}$ is the distribution from which $p_1, \ldots, p_K$ are drawn. We want to stress that no prior knowledge about the dataset or the amount of consensus among the private mechanisms is required to use our optimization framework. When there is no prior knowledge about $p_1, \ldots, p_K$, $\mathcal{T}$ is set to be the uniform distribution for maximizing the expected utility. Note the above optimization problem also enables the flexibility of incorporating prior knowledge about the mechanisms by choosing a prior distribution $\mathcal{T}$ to further improve the utility.

**Optimizing Over All Algorithms.** We want to stress that by solving the above optimization problem, we are indeed optimizing over *all* algorithms for maximal utility, since we show in Lemma 3.3 DaRRM that captures *all reasonable* algorithms computing a private majority.

**Linear Optimization Objective.** Perhaps surprisingly, it turns out that optimizing for $\gamma^*$ is a Linear Programming (LP) problem! Indeed, after expanding the optimization objective in Eq. 2 by the utility definition (see Definition 2.4), optimizing the above objective is essentially same as optimizing:

$$\min_{\gamma \in [0,1]^{K+1}} -\frac{1}{2} \sum_{l=\frac{K+1}{2}}^{K} \mathbb{E}_{p_1, p_2, \ldots, p_K \sim \mathcal{T}} [(\alpha_l - \alpha_{K-l})] \cdot \gamma(l)$$

where $\alpha_l = \Pr[\mathcal{L}(\mathcal{D}) = l], \forall l \in \{0, 1, \ldots, K\}$ and observe $\mathcal{L}(\mathcal{D}) \sim \text{PoissonBinomial}(p_1, \ldots, p_K)$. The above objective is linear in $\gamma$. See a full derivation in Appendix C.1.

Although taking the expectation over $p_1, \ldots, p_K$ involves integrating over $K$ variables and this can be computationally expensive, we discuss how to formulate a computationally efficient approximation of the objective in Appendix C.2, which we later use in the experiments. Note that the objective only for maximizing the utility and hence approximating the objective does not affect the privacy guarantee.

**Reducing Infinitely Many Constraints to A Polynomial Set.** The constraints in the optimization problem (Eq. 3) is what makes sure the output of DaRRM$_\gamma$ is $m\epsilon$-differentially private. We thus call them *the privacy constraints*. Note that the privacy constraints are linear in $\gamma$.

Though it appears we need to solve for infinitely many such privacy constraints since $p_i$'s and $p_i'$'s are continuous, we show that through a structural understanding of DaRRM, we can reduce the number of privacy constraints from infinitely many to exponentially many, and further to a polynomial set. First, we observe the privacy cost objective $f$ is linear in each independent pair of $(p_i, p_i')$ fixing all $(p_j, p_j')$, $\forall j \neq i$, and hence finding the worst case probabilities in $(p_i, p_i')$ given any $\gamma$, $(p_i^*, p_i'^*) = \arg\max_{(p_i, p_i')} f(p_1, \ldots, p_K, p_1', \ldots, p_K'; \gamma)$ is a linear programming (LP) problem. Furthermore, since $p_i$ and $p_i'$ are the probability of outputting 1 from the $i$-th $(\epsilon, \Delta)$-differentially private mechanism $M_i$ on adjacent datasets, by definition, they are close and lie in a feasible region $\mathcal{F}_i$, which we show has 8 corners if $\delta > 0$ (and only 4 corners if $\delta = 0$). This implies $(p_i^*, p_i'^*)$ only happens at one of the corners of $\mathcal{F}_i$, and hence the number of constraints reduces to $K^8$ (and $K^4$ if $\delta = 0$). Second, observe that $\alpha_l$ and $\alpha_l'$ in the privacy cost objective $f$ are the pmf of two Poisson Binomial distributions at $l \in \{0, \ldots, K\}$. Notice that the Poisson Binomial is invariant under the permutation of its parameters, i.e. PoissonBinomial$(p_1, \ldots, p_K)$ has the same distribution as PoissonBinomial$(\pi(p_1, \ldots, p_K))$, under some permutation $\pi$. Based on this observation, we show the number of constraints can be further reduced to $O(K^7)$ if $\delta > 0$ (and $O(K^3)$ if $\delta = 0$). We formalize the two-step reduction of the number of privacy constraints in Lemma 5.1 as follows. See a full proof in Appendix C.3. [1]

---

[1] **Practical Limitation.** Although the number of constraints is polynomial in $K$ and optimizing $\gamma$ in DaRRM is an LP, $O(K^7)$ can still make the number of constraints intractably large when $K$ is large. In practice, we observe with the Gurobi

**Lemma 5.1.** *Consider using DaRRM (Algorithm 1) to solve Problem 1.1 and let $f$ be the privacy cost objective as defined in Lemma 3.4. Given an arbitrary noise function $\gamma$, let the worst case probabilities be $(p_1^*, \ldots, p_K^*, p_1'^*, \ldots, p_K'^*) = \arg\max_{\{(p_i, p_i')\}_{i=1}^K} f(p_1, \ldots, p_K, p_1', \ldots, p_K'; \gamma)$.*

$$(p_1^*, \ldots, p_K^*, p_1'^*, \ldots, p_K'^*) = \underset{\{(p_i, p_i')\}_{i=1}^K}{\arg\max} \, f(p_1, \ldots, p_K, p_1', \ldots, p_K'; \gamma)$$

*Then, each pair $(p_i^*, p_i'^*), \forall i \in [K]$ satisfies*

$$(p_i^*, p_i'^*) \in \{(0,0), (1,1), (0,\Delta), (\Delta, 0), (1-\Delta, 1),$$
$$(1, 1-\Delta), (\frac{e^\epsilon + \Delta}{e^\epsilon + 1}, \frac{1-\Delta}{e^\epsilon + 1}), (\frac{1-\Delta}{e^\epsilon + 1}, \frac{e^\epsilon + \Delta}{e^\epsilon + 1})\}$$

*Furthermore, when $\delta > 0$, there exists a finite vector set $\mathcal{P}$ of size $O(K^7)$ such that if $\beta = \max_{\{(p_i, p_i')\}_{i=1}^K \in \mathcal{P}} f(p_1, \ldots, p_K, p_1', \ldots, p_K'; \gamma)$, then $f(p_1^*, \ldots, p_K^*, p_1'^*, \ldots, p_K'^*; \gamma) \leq \beta$. When $\delta = 0$, the size of $\mathcal{P}$ can be reduced to $O(K^3)$.*

## 6 Experiments

We empirically solve[2] the above optimization problem (Eq. 2) using the `Gurobi`[3] solver and first present the shape of the optimized $\gamma$ function, which we call $\gamma_{opt}$, and its utility in Section 6.1. Then, we demonstrate the compelling effectiveness of DaRRM with an optimized $\gamma$ function, i.e., DaRRM$_{\gamma_{opt}}$, in ensembling labels for private prediction from private teachers through the application of semi-supervised knowledge transfer for private image classification in Section 6.2.

### 6.1 Optimized $\gamma$ in Simulations

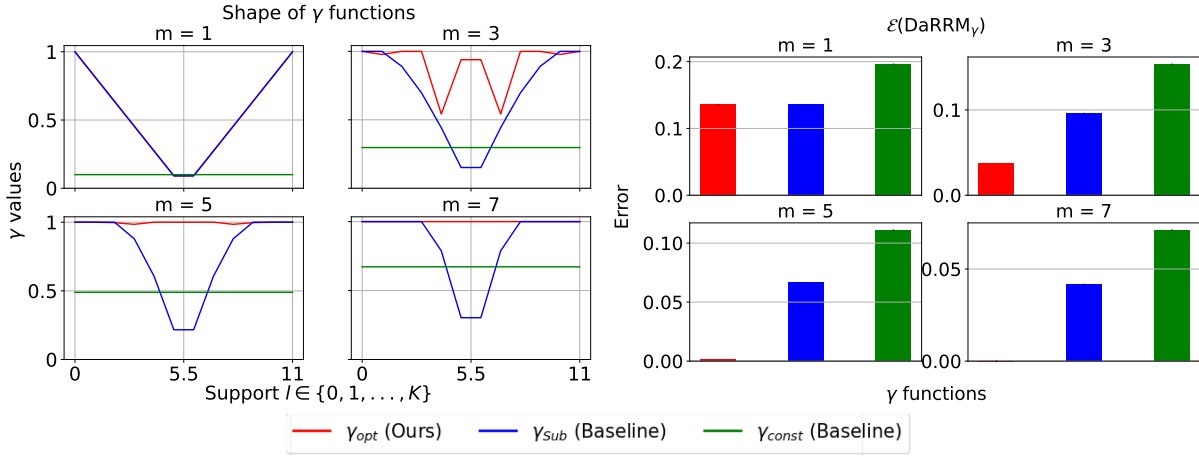

Figure 2: Plots of the shape and $\mathcal{E}(\text{DaRRM}_\gamma)$ of different $\gamma$ functions: the optimized $\gamma_{opt}$, and the baselines $\gamma_{Sub}$ (corresponding to subsampling) and $\gamma_{const}$ (corresponding to RR). Here, $K = 11, m \in \{1, 3, 5, 7\}, \epsilon = 0.1, \Delta = 10^{-5}$ and $\delta = 1 - (1-\Delta)^m \approx m\Delta$.

We compare the shape and the error $\mathcal{E}(\text{DaRRM}_\gamma)$ of different $\gamma$ functions: an optimized $\gamma_{opt}$ and the subsampling $\gamma_{Sub}$ as in Lemma 3.1[4]. We also compare against $p_{const}$ in the classical baseline RR (see

---

optimizer, one can optimize $\gamma$ for $K \leq 41$ on a laptop if $\delta > 0$. But if $\delta = 0$, since the number of privacy constraints is $O(K^3)$, one can optimize for $K$ over 100.

[2]All code for the experiments can be found at `https://anonymous.4open.science/r/OptimizedPrivateMajority-CF50`

[3]`https://www.gurobi.com/`

[4]Note the subsampling mechanism from Section 4, which enjoys a privacy amplification by a factor of 2, only applies to pure differential privacy settings (i.e., when $\Delta = \delta = 0$). However, we focus on the more general approximate differential privacy

Section A.1) and $\mathcal{E}(\mathsf{RR})$. Here, $p_{const}$ can be viewed as a constant noise function $\gamma_{const}(l) = p_{const}, \forall l \in \{0, 1, \ldots, K\}$; and $\mathcal{E}(\mathsf{RR})$ is the same as $\mathcal{E}(\mathsf{DaRRM}_{\gamma_{const}})$.

We present the results with $K = 11, \epsilon = 0.1, \Delta = 10^{-5}$ and $m \in \{1, 3, 5, 7\}$. We assume there is no prior knowledge about the mechanisms $\{M_i\}_{i=1}^{K}$, and set the prior distribution from which $p_i$'s are drawn, $\mathcal{T}$, to be the uniform distribution, in the optimization objective (Eq. 2) searching for $\gamma_{opt}$. To ensure a fair comparison against the subsampling baseline, we set $\delta$ to be the one by $m$-fold general composition (see Theorem 2.3), which in this case, is $\delta = 1 - (1 - \Delta)^m \approx m\Delta$. We plot each $\gamma$ functions over the support $\{0, 1, \ldots, K\}$ and the corresponding error of each algorithm in Figure 2.

**Discussion.** In summary, at $m = 1$, the optimized noise function $\gamma_{opt}$ overlaps with $\gamma_{sub}$ which corresponds to the subsampling baseline. This agrees with our lower bound on the error in Lemma 3.2, which implies that at $m = 1$, subsampling indeed gives the optimal error. When $m > 1$, the optimized noise function $\gamma_{opt}$ has the highest probability of outputting the true majority over the support than the $\gamma$ functions corresponding to the baselines. This implies $\mathsf{DaRRM}_{\gamma_{opt}}$ has the lowest error (and hence, highest utility), which is verified on the bottom set of plots. More results on comparing the $\mathsf{DaRRM}_{\gamma_{opt}}$ optimized under the uniform $\mathcal{T}$ against the baselines by general composition (Theorem 2.3) and in pure differential privacy settings (i.e., $\Delta = \delta = 0$) for large $K$ and $m$ can be found in Appendix D.1.1 and D.1.2. Furthermore, we include results optimizing $\gamma$ using a non-uniform $\mathcal{T}$ prior in Appendix D.1.3.

## 6.2 Private Semi-Supervised Knowledge Transfer

| Dataset | MNIST | | | Dataset | Fashion-MNIST | | |
|---|---|---|---|---|---|---|---|
| | GNMax | DaRRM$_{\gamma_{Sub}}$ | DaRRM$_{\gamma_{opt}}$ | | GNMax | DaRRM$_{\gamma_{Sub}}$ | DaRRM$_{\gamma_{opt}}$ |
| # Queries | (Baseline) | (Baseline) | (Ours) | # Queries | (Baseline) | (Baseline) | (Ours) |
| $Q = 20$ | 0.63 (0.09) | 0.76 (0.09) | **0.79 (0.09)** | $Q = 20$ | 0.65 (0.11) | 0.90 (0.07) | **0.96 (0.03)** |
| $Q = 50$ | 0.66 (0.06) | 0.75 (0.06) | **0.79 (0.05)** | $Q = 50$ | 0.59 (0.06) | 0.94 (0.03) | **0.96 (0.02)** |
| $Q = 100$ | 0.64 (0.04) | 0.76 (0.04) | **0.80 (0.04)** | $Q = 100$ | 0.64 (0.04) | 0.93 (0.02) | **0.96 (0.02)** |

Table 1: Accuracy of the predicted labels of $Q$ query samples on datasets MNIST (on the left) and Fashion-MNIST (on the right). We report the mean and one std. in parentheses over 10 random draws of the query samples from the test dataset. Note each prediction on the query sample is $(\epsilon_{query}, \delta_{query})$ -differentially private. With the same per query privacy loss (and hence the same total privacy loss over $Q$ samples), $\mathsf{DaRRM}_{\gamma_{opt}}$ achieves the highest accuracy compared to the other two baselines.

**Semi-supervised Knowledge Transfer.** We apply our $\mathsf{DaRRM}$ framework in the application of semi-supervised knowledge transfer for private image classification. We follow a similar setup as in PATE Papernot et al. (2017; 2018), where one trains $K$ teachers, each on a subset of a sensitive dataset, and at the inference time, queries the teachers for the majority of their votes, i.e., the predicted labels, of a test sample. Each time the teachers are queried, there is a privacy loss, and we focus on this private prediction subroutine in this section. To limit the total privacy loss over all queries, the student model is also trained on a public dataset without labels. The student model queries the labels of a small portion of the samples in this dataset from the teachers and is then trained using semi-supervised learning algorithms on both the labeled and unlabeled samples from the public dataset.

**Baselines.** We want the privacy loss per query of a test sample to the teachers to be $(\epsilon_{query}, \delta_{query})$. This can be achieved via two ways: 1) Train $K$ non-private teachers, add Gaussian noise to the number of predicted labels from the teachers in each output class, and output the majority of the noisy votes. This is exactly the $\mathsf{GNMax}$ algorithm from PATE Papernot et al. (2018). 2) Train $K$ $(\epsilon, \Delta)$-differentially private teachers and output the majority of the teachers' votes by adding a smaller amount of noise. This can be computed using $\mathsf{DaRRM}$ with an appropriate noise function $\gamma$. We compare the performance of $\mathsf{GNMax}$ and $\mathsf{DaRRM}$ with two $\gamma$ functions: $\gamma_{opt}$ (i.e., the optimized $\gamma$), and $\gamma_{Sub}$ (i.e., the subsampling baseline). The overall privacy loss over $Q$ queries to the teachers can be computed by general composition (Theorem 2.3).

---

settings (with $\Delta > 0$) in the experiments, and hence, the subsampling baseline we consider throughout this section is the basic version without privacy amplification. To see how the subsampling mechanism from Section 4 with privacy amplification compares against the other algorithms, please refer to Appendix D.1.2.

**Experiment Setup.** We use samples from two randomly chosen classes — class 5 and 8 — from the `MNIST` and `Fashion-MNIST` datasets to form our training and testing datasets. Our `MNIST` has a total of 11272 training samples and 1866 testing samples; our `Fashion-MNIST` has 10000 training samples and 2000 testing samples. We train $K = 11$ teachers on equally divided subsets of the training datasets. Each teacher is a CNN model. The non-private and private teachers are trained using `SGD` and `DP-SGD` Abadi et al. (2016), respectively, for 5 epochs. *DaRRM Setup:* The Gaussian noise in DP-SGD has zero mean and std. $\sigma_{dpsgd} = 12$; the gradient norm clipping threshold is $C = 1$. This results in each private teacher, trained on `MNIST` and `Fashion-MNIST`, being $(\epsilon, \Delta) = (0.0892, 10^{-4})$ and $(0.0852, 10^{-4})$-differentially private, respectively, after 5 epochs. We set the privacy allowance $m = 3^5$ and the privacy loss per query is then computed using general composition under $m$-fold, which give the same privacy loss in the high privacy regime, resulting in $(\epsilon_{query}, \delta_{query}) = (0.2676, 0.0003)$ on `MNIST` and $(0.2556, 0.0003)$ on `Fashion-MNIST`. *GNMax Setup:* We now compute the std. $\sigma$ of the Gaussian noise used by `GNMax` to achieve a per-query privacy loss of $(m\epsilon, m\Delta)$, as in the `DaRRM` setup. We optimize $\sigma$ according to the Renyi differential privacy loss bound of Gaussian noise. Although Papernot et al. (2018) gives a potentially tighter data-dependent privacy loss bound for majority ensembling *non-private* teachers, we found when $K$ and the number of output classes are small as in our case, even if all teachers agree on a single output class, the condition of the data-dependent bound is not satisfied. Hence, we only use the privacy loss bound of Gaussian noise here to set $\sigma$ in `GNMax`. See Appendix D.2.1 for more details, including the $\sigma$ values and other parameters. Finally, the per sample privacy loss and the total privacy loss over $Q$ queries, which is computed by advanced composition, are reported in Table 9.

The testing dataset is treated as the public dataset on which one trains a student model. Papernot et al. (2018) empirically shows querying $Q = 1\%N$ samples from a public dataset of size $N$ suffices to train a student model with a good performance. Therefore, we pick $Q \in \{20, 50, 100\}$. We repeat the selection of $Q$ samples 10 times and report the mean test accuracy with one std. in parentheses in Table 1. The $Q$ queries serve as the labeled samples in training the student model. The higher the accuracy of the labels from the queries, the better the final performance of the student model. We skip the actual training of the student model using semi-supervised learning algorithms here.

| Dataset | # Queries | Privacy loss per query $(\epsilon_{query}, \delta_{query})$ | Total privacy loss over $Q$ queries $(\epsilon_{total}, \delta_{total})$ |
|---|---|---|---|
| `MNIST` | $Q = 20$ | $(0.2676, 0.0003)$ | $(5.352, 0.006)$ |
| | $Q = 50$ | | $(9.901, 0.015)$ |
| | $Q = 100$ | | $(15.044, 0.030)$ |
| `Fashion MNIST` | $Q = 20$ | $(0.2556, 0.0003)$ | $(5.112, 0.006)$ |
| | $Q = 50$ | | $(9.382, 0.015)$ |
| | $Q = 100$ | | $(14.219, 0.030)$ |

Table 2: The privacy loss per query to the teachers and the total privacy loss over $Q$ queries. Note the total privacy loss is computed by general composition (see Theorem 2.3), where we set $\delta' = 0.0001$.

**Discussion.** Table 1 shows $\mathsf{DaRRM}_{\gamma_{opt}}$ achieves the highest accuracy (i.e., utility) compared to the two baselines on both datasets. First, comparing to $\mathsf{DaRRM}_{\gamma_{Sub}}$, we verify that subsampling does not achieve a tight privacy-utility tradeoff, and we can optimize the noise function $\gamma$ in `DaRRM` to maximize the utility given a target privacy loss. Second, comparing to `GNMax`, the result shows there are regimes where ensembling private teachers gives a higher utility than directly ensembling non-private teachers, assuming the outputs in both settings have the same privacy loss. Intuitively, this is because ensembling private teachers adds fine-grained noise during both training the teachers and aggregation of teachers' votes, while ensembling non-private teachers adds a coarser amount of noise only to the teachers' outputs. This further motivates

---

[5] Here, we present results with privacy allowance $m = 3$ because we think this is a more interesting case. $m = 1$ is less interesting, since one cannot get improvement compared to the subsampling baseline. $m$ close to a $\frac{K}{2} \approx 5$ is also less interesting, as this case seems too easy for our proposed method (the optimized $\gamma$ function is very close to 1, meaning very little noise needs to be added in this case). Hence, we pick $m = 3$, which is a case when improvement is possible, and is also potentially challenging for our optimization framework. This is also realistic as most applications would only want to tolerate a constant privacy overhead. See more results with different privacy allowance $m$'s in this setting in Appendix D.2.2.

private prediction from private teachers and the practical usage of DaRRM, in addition to the need of aggregating private teachers in federated learning settings with an honest-but-curious server.

## 7 Conclusion

In computing a private majority from $K$ private mechanisms, we propose the DaRRM framework, which is provably general, with a customizable $\gamma$ function. We show a privacy amplification by a factor of 2 in the i.i.d. mechanisms and a pure differential privacy setting. For the general setting, we propose an tractable optimization algorithm that maximizes utility while ensuring privacy guarantees. Furthermore, we demonstrate the empirical effectiveness of DaRRM with an optimized $\gamma$. We hope that this work inspires more research on the intersection of privacy frameworks and optimization.

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

# A Details of Section 3

## A.1 Randomized Response with Constant Probability $p_{const}$

---

**Algorithm 2** Randomized Response Majority (RR)

---

1: Input: $K$ $(\epsilon, \Delta)$-DP mechanisms $\{M_i\}_{i=1}^K$, noise function $\gamma : \{0, \ldots, K\} \to [0, 1]$, dataset $\mathcal{D}$, privacy allowance $1 \le m \le K$, failure probability $\delta \ge \Delta \ge 0$
2: Output: $(m\epsilon, \delta)$-DP majority vote of $\{M_i\}_{i=1}^K$
3: Compute a *constant* probability $p_{const} \in [0, 1]$
4: Flip the $p_{const}$- biased coin
5: **if** Head (with probability $p_{const}$) **then**
6: $\quad$ $\mathcal{S} = \{S_1, .., S_k\}$, where $S_i \sim M_i(\mathcal{D})$
7: $\quad$ $\mathcal{L} = \sum_{i=1}^K S_i$
8: $\quad$ Output $\mathbb{I}\{\frac{1}{K}\mathcal{L} \ge \frac{1}{2}\}$
9: **else**
10: $\quad$ Output 0/1 with equal probability
11: **end if**

---

We show the magnitude of $p_{const}$ in RR (Algorithm 2) to solve Problem 1.1, such that the output is $(m\epsilon, \delta)$-DP, in Lemma A.1.

**Lemma A.1.** *Consider using RR (Algorithm 2) to solve Problem 1.1. Let the majority of $K$ $(\epsilon, \Delta)$-differentially private mechanisms be $(\tau\epsilon, \lambda)$-differentially private, where $\tau \in [1, K]$ and $\lambda \in [0, 1)$ are computed by simple composition (Theorem 2.2) or general composition (Theorem 2.3). If*

$$p_{const} \le \frac{e^{m\epsilon} - 1 + 2\delta}{\frac{2(e^{\tau\epsilon} - e^{m\epsilon} + (1 + e^{m\epsilon})\lambda)}{e^{\tau\epsilon} + 1} + e^{m\epsilon} - 1} \tag{4}$$

*then RR is $(m\epsilon, \delta)$-differentially private.*

*Proof of Lemma A.1.* Let $x \in \{0, 1\}$ denote the output of RR. Let $q_x = \Pr[\mathcal{L}(\mathcal{D}) = x]$ and $q'_x = \Pr[\mathcal{L}(\mathcal{D}') = x]$, where $\mathcal{L}(\mathcal{D}) = \sum_{i=1}^K M_i(\mathcal{D})$, $\mathcal{L}(\mathcal{D}') = \sum_{i=1}^K M_i(\mathcal{D}')$ and $\mathcal{D}, \mathcal{D}'$ are adjacent datasets. Recall each mechanism $M_i$ is $(\epsilon, \Delta)$-differentially private, and the majority of the outputs of $\{M_i\}_{i=1}^K$ is $(\tau\epsilon, \lambda)$-differentially private. When $\Delta = 0$, using simple composition, $\tau = K$ and $\lambda = 0$. When $\Delta > 0$, using general composition $\tau \approx \sqrt{K}$ and $\lambda \approx K\Delta$. By definition of differential privacy (Definition 2.1), all of the following four constraints on $q_x, q'_x$ apply:

$$q_x \le e^{\tau\epsilon} q'_x + \lambda, \quad \text{and} \quad 1 - q'_x \le e^{\tau\epsilon}(1 - q_x) + \lambda$$
$$q'_x \le e^{\tau\epsilon} q_x + \lambda, \quad \text{and} \quad 1 - q_x \le e^{\tau\epsilon}(1 - q'_x) + \lambda$$

To ensure RR is $(m\epsilon, \delta)$-differentially private, $p_{const}$ needs to be such that for all possible $q_x, q'_x \in [0, 1]$,

$$\Pr[\mathsf{RR}(\mathcal{D}) = x] \le e^{m\epsilon} \Pr[\mathsf{RR}(\mathcal{D}') = x] + \delta \tag{5}$$

$$p_{const} \cdot q_x + \frac{1}{2}(1 - p_{const}) \le e^{m\epsilon}(p_{const} \cdot q'_x + \frac{1}{2}(1 - p_{const})) + \delta \tag{6}$$

$$(q_x - e^{m\epsilon} q'_x + \frac{1}{2}e^{m\epsilon} - \frac{1}{2}) \cdot p_{const} \le \frac{1}{2}e^{m\epsilon} - \frac{1}{2} + \delta \tag{7}$$

Let $h(q_x, q'_x) := q_x - e^{m\epsilon} q'_x + \frac{1}{2}e^{m\epsilon} - \frac{1}{2}$. The above inequality of $p_{const}$ (Eq. 7) needs to hold for worst case output probabilities $q_x^*, q_x'^*$ that cause the maximum privacy loss. That is, $p_{const}$ needs to satisfy

$$p_{const} \cdot max_{q_x, q'_x} h(q_x, q'_x) \le \frac{1}{2}e^{m\epsilon} - \frac{1}{2} + \delta \tag{8}$$

To find the worst case output probabilities, we solve the following Linear Programming (LP) problem:

$$\text{Objective:} \qquad \max_{q_x, q'_x} \quad h(q_x, q'_x) := q_x - e^{m\epsilon} q'_x + \frac{1}{2} e^{m\epsilon} - \frac{1}{2} \qquad (9)$$

$$\text{Subject to:} \qquad 0 \le q_x \le 1, 0 \le q'_x \le 1 \qquad (10)$$

$$q_x \le e^{\tau\epsilon} q'_x + \lambda, 1 - q'_x \le e^{\tau\epsilon}(1 - q_x) + \lambda \qquad (11)$$

$$q'_x \le e^{\tau\epsilon} q_x + \lambda, 1 - q_x \le e^{\tau\epsilon}(1 - q'_x) + \lambda \qquad (12)$$

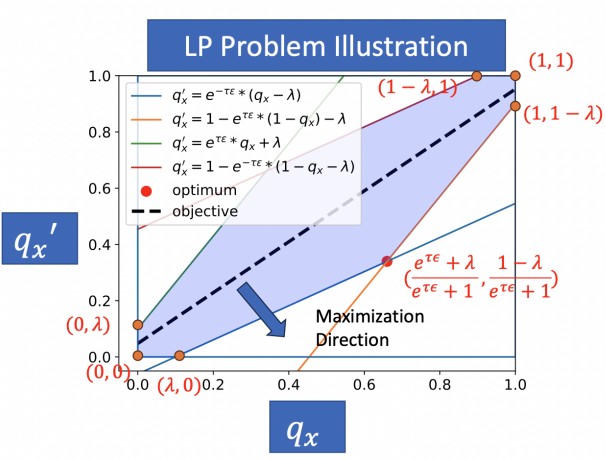

Figure 3: A visualization of the above LP problem.

The optimum of any LP problem is at the corners of the feasible region, which is bounded by the optimization constraints. We plot the feasible region $\mathcal{F}$ and the objective of the above LP problem in Figure 3. Here, $(q_x^*, q_x'^*) = \arg\max_{q_x, q'_x} h(q_x, q'_x) \in \{(0, 0), (1, 1), (0, \lambda), (\lambda, 0), (1 - \lambda, 1), (1, 1 - \lambda), (\frac{1-\lambda}{e^{\tau\epsilon}+1}, \frac{e^{\tau\epsilon}+\lambda}{e^{\tau\epsilon}+1}), (\frac{e^{\tau\epsilon}+\lambda}{e^{\tau\epsilon}+1}, \frac{1-\lambda}{e^{\tau\epsilon}+1})\}$. The optimum of the LP problem – that is, the worse case probabilities $q_x^*, q_x'^*$ – is,

$$q_x^* = \frac{e^{\tau\epsilon} + \lambda}{e^{\tau\epsilon} + 1}, \quad q_x'^* = \frac{1 - \lambda}{e^{\tau\epsilon} + 1} \qquad (13)$$

By Eq. 8,

$$p_{const} \cdot \left( \frac{e^{\tau\epsilon} + \lambda}{e^{\tau\epsilon} + 1} - e^{m\epsilon} \frac{1 - \lambda}{e^{\tau\epsilon} + 1} + \frac{1}{2} e^{m\epsilon} - \frac{1}{2} \right) \le \frac{1}{2}(e^{m\epsilon} - 1) + \delta \qquad (14)$$

$$p_{const} \cdot \left( \frac{e^{\tau\epsilon} - e^{m\epsilon} + (1 + e^{m\epsilon})\lambda}{e^{\tau\epsilon} + 1} + \frac{1}{2}(e^{m\epsilon} - 1) \right) \le \frac{1}{2}(e^{m\epsilon} - 1) + \delta \qquad (15)$$

$$p_{const} \le \frac{e^{m\epsilon} - 1 + 2\delta}{\frac{2(e^{\tau\epsilon} - e^{m\epsilon} + (1 + e^{m\epsilon})\lambda)}{e^{\tau\epsilon} + 1} + e^{m\epsilon} - 1} \qquad (16)$$

For small $m, \epsilon, K$, using the approximation $e^y \approx 1 + y$ and that $\tau\epsilon < 2$,

$$p_{const} \approx \frac{m\epsilon + 2\delta}{\frac{2(\tau\epsilon - m\epsilon + (2 + m\epsilon)\lambda)}{\tau\epsilon + 2} + m\epsilon} \approx \frac{m\epsilon + 2\delta}{\tau\epsilon + (2 + m\epsilon)\lambda} \qquad (17)$$

In the pure differential privacy setting, $\delta = 0, \lambda = 0, \tau = K$, and so $p_{const} \approx \frac{m}{K}$; and in the approximate differential privacy setting, $\lambda \approx 0, \delta \approx 0, \tau \approx \sqrt{K}$, and so $p_{const} \approx \frac{m}{\sqrt{K}}$. $\qquad \square$

---

**Algorithm 3** Subsampling Majority (SubMaj)

---

1: Input: $K$ $(\epsilon, \Delta)$-DP mechanisms $\{M_i\}_{i=1}^K$, noise function $\gamma : \{0, \ldots, K\} \to [0, 1]$, dataset $\mathcal{D}$, privacy allowance $1 \leq m \leq K$, failure probability $\delta \geq \Delta \geq 0$
2: Output: $(m\epsilon, \delta)$-DP majority vote of $\{M_i\}_{i=1}^K$
3: $\mathcal{S} = \{S_1, .., S_k\}$, where $S_i \sim M_i(\mathcal{D})$
4: $\mathcal{J}_m \leftarrow m$ indices chosen uniformly at random from $[K]$ without replacement
5: $\widehat{\mathcal{L}} = \sum_{j \in \mathcal{J}} S_j$
6: Output $\mathbb{I}\{\frac{1}{m}\widehat{\mathcal{L}} \geq \frac{1}{2}\}$

---

## A.2 Proof of Lemma 3.1

**Lemma A.2** (Restatement of Lemma 3.1). *Consider Problem 1.1, with the privacy allowance $m \in [K]$. Consider the data-dependent algorithm that computes $\mathcal{L}(\mathcal{D})$ and then applies RR with probability $p_\gamma$. If $p_\gamma = \gamma_{Sub}(l)$, where $l \in \{0, 1, \ldots, K\}$ is the value of $\mathcal{L}(\mathcal{D})$, i.e., the (random) sum of observed outcomes on dataset $\mathcal{D}$, and $\gamma_{Sub} : \{0, 1, \ldots, K\} \to [0, 1]$ is*

$$\gamma_{Sub}(l) = \gamma_{Sub}(K - l)$$

$$= \begin{cases} 1 - 2\sum_{j=\frac{m+1}{2}}^m \frac{\binom{l}{j}\binom{K-l}{m-j}}{\binom{K}{m}} & \text{if } m \text{ is odd} \\ 1 - 2\sum_{j=\frac{m}{2}+1}^m \frac{\binom{l}{j}\binom{K-l}{m-j}}{\binom{K}{m}} - \frac{\binom{l}{\frac{m}{2}}\binom{K-l}{\frac{m}{2}}}{\binom{K}{m}} & \text{if } m \text{ is even} \end{cases}$$

*then the majority of $m$ out of $K$ subsampled mechanisms without replacement and the output of our data-dependent RR algorithm have the same distribution.*

*Proof of Lemma 3.1.* Let $\mathcal{L} = \sum_{i=1}^K S_i$ be the sum of observed outcomes from $K$ mechanisms. Following Algorithm 3, $\mathcal{J}_m$ denotes the $m$ indices chosen uniformly at random from $[K]$ without replacement. Conditioned on $\mathcal{L}$, notice the output of SubMaj follows a hypergeometric distribution. The output probability of SubMaj is

$$\Pr[\mathsf{SubMaj}(\mathcal{D}) = 1] = \sum_{l=0}^K \Pr[\mathsf{SubMaj}(\mathcal{D}) = 1 \mid \mathcal{L} = l] \cdot \Pr[\mathcal{L} = l] \tag{18}$$

$$= \sum_{l=0}^K \Pr[\sum_{j \in \mathcal{J}_m} S_j \geq \frac{m}{2} \mid \mathcal{L} = l] \cdot \Pr[\mathcal{L} = l] \tag{19}$$

$$= \begin{cases} \sum_{l=0}^K (\sum_{j=\frac{m+1}{2}}^m \frac{\binom{l}{j}\binom{K-l}{m-j}}{\binom{K}{m}}) \cdot \Pr[\mathcal{L} = l] & \text{if } m \text{ is odd} \\ \sum_{l=0}^K (\sum_{j=\frac{m}{2}+1}^m \frac{\binom{l}{j}\binom{K-l}{m-j}}{\binom{K}{m}} + \frac{1}{2}\frac{\binom{l}{\frac{m}{2}}\binom{K-l}{\frac{m}{2}}}{\binom{K}{m}}) \cdot \Pr[\mathcal{L} = l] & \text{if } m \text{ is even} \end{cases} \tag{20}$$

Consider an arbitrary noise function $\gamma_{Sub} : \{0, 1, \ldots, K\} \to [0, 1]$. Let $\mathsf{RR\text{-}d}(\mathcal{D})$ denote the output of the data-dependent RR-d on dataset $\mathcal{D}$, where RR-d has the *non-constant* probability set by $\gamma_{Sub}$. The output probability of RR is,

$$\Pr[\mathsf{RR\text{-}d}(\mathcal{D}) = 1] = \sum_{l=0}^K \Pr[\mathsf{RR\text{-}d}(\mathcal{D}) = 1 \mid \mathcal{L} = l] \cdot \Pr[\mathcal{L} = l] \tag{21}$$

$$= \sum_{l=0}^K (\gamma_{Sub}(l) \cdot \mathbb{I}\{l \geq \frac{K+1}{2}\} + \frac{1}{2}(1 - \gamma_{Sub}(l))) \cdot \Pr[\mathcal{L} = l] \tag{22}$$

We want $\Pr[\mathsf{RR\text{-}d}(\mathcal{D}) = 1] = \Pr[\mathsf{Submaj}(\mathcal{D}) = 1]$.

If $m$ is odd, for any $l \leq \frac{K-1}{2}$, this is

$$\frac{1}{2}(1 - \gamma_{Sub}(l)) = \sum_{j=\frac{m+1}{2}}^{m} \frac{\binom{l}{j}\binom{K-l}{m-j}}{\binom{K}{m}}$$

$$\Rightarrow \gamma_{Sub}(l) = 1 - 2 \sum_{j=\frac{m+1}{2}}^{m} \frac{\binom{l}{j}\binom{K-l}{m-j}}{\binom{K}{m}} \tag{23}$$

and for any $l \geq \frac{K+1}{2}$, this is

$$\frac{1}{2} + \frac{1}{2}\gamma_{Sub}(l) = \sum_{j=\frac{m+1}{2}}^{m} \frac{\binom{l}{j}\binom{K-l}{m-j}}{\binom{K}{m}}$$

$$\Rightarrow \gamma_{Sub}(l) = 2 \sum_{j=\frac{m+1}{2}}^{m} \frac{\binom{l}{j}\binom{K-l}{m-j}}{\binom{K}{m}} - 1 \tag{24}$$

Similarly, if $m$ is even, for any $l \leq \frac{K-1}{2}$, this is

$$\frac{1}{2}(1 - \gamma_{Sub}(l)) = \sum_{j=\frac{m}{2}+1}^{m} \frac{\binom{l}{j}\binom{K-l}{m-j}}{\binom{K}{m}} + \frac{1}{2}\frac{\binom{l}{\frac{m}{2}}\binom{K-l}{\frac{m}{2}}}{\binom{K}{m}}$$

$$\Rightarrow \gamma_{Sub}(l) = 1 - 2 \sum_{j=\frac{m}{2}+1}^{m} \frac{\binom{l}{j}\binom{K-l}{m-j}}{\binom{K}{m}} - \frac{\binom{l}{\frac{m}{2}}\binom{K-l}{\frac{m}{2}}}{\binom{K}{m}} \tag{25}$$

and for any $l \geq \frac{K+1}{2}$, this is

$$\frac{1}{2} + \frac{1}{2}\gamma_{Sub}(l) = \sum_{j=\frac{m}{2}+1}^{m} \frac{\binom{l}{j}\binom{K-l}{m-j}}{\binom{K}{m}} + \frac{1}{2}\frac{\binom{l}{\frac{m}{2}}\binom{K-l}{\frac{m}{2}}}{\binom{K}{m}}$$

$$\Rightarrow \gamma_{Sub}(l) = 2 \sum_{j=\frac{m}{2}+1}^{m} \frac{\binom{l}{j}\binom{K-l}{m-j}}{\binom{K}{m}} + \frac{\binom{l}{\frac{m}{2}}\binom{K-l}{\frac{m}{2}}}{\binom{K}{m}} - 1 \tag{26}$$

Next, we show the above $\gamma_{Sub}$ is indeed symmetric around $\frac{K}{2}$. For any $l \leq \frac{K-1}{2}$, there is $K - l \geq \frac{K+1}{2}$. If $m$ is odd,

$$\gamma_{Sub}(K - l) = 2 \sum_{j=\frac{m+1}{2}}^{m} \frac{\binom{K-l}{j}\binom{l}{m-j}}{\binom{K}{m}} - 1 = 2\Big(1 - \sum_{j=1}^{\frac{m-1}{2}} \frac{\binom{K-l}{j}\binom{l}{m-j}}{\binom{K}{m}}\Big) - 1$$

$$= 1 - 2 \sum_{j=1}^{\frac{m-1}{2}} \frac{\binom{K-l}{j}\binom{l}{m-j}}{\binom{K}{m}} = 1 - 2 \sum_{j=\frac{m+1}{2}}^{m} \frac{\binom{l}{j}\binom{K-l}{m-j}}{\binom{K}{m}}$$

$$= \gamma_{Sub}(l) \tag{27}$$

Similarly, if $m$ is even,

$$\gamma_{Sub}(K - l) = 2 \sum_{j=\frac{m}{2}+1}^{m} \frac{\binom{K-l}{j}\binom{l}{m-j}}{\binom{K}{m}} + \frac{\binom{l}{\frac{m}{2}}\binom{K-l}{\frac{m}{2}}}{\binom{K}{m}} - 1 = 2\Big(1 - \sum_{j=1}^{\frac{m}{2}-1} \frac{\binom{K-l}{j}\binom{l}{m-j}}{\binom{K}{m}} - \frac{1}{2}\frac{\binom{l}{\frac{m}{2}}\binom{K-l}{\frac{m}{2}}}{\binom{K}{m}}\Big) - 1$$

$$= 1 - 2 \sum_{j=1}^{\frac{m}{2}-1} \frac{\binom{K-l}{j}\binom{l}{m-j}}{\binom{K}{m}} - \frac{\binom{l}{\frac{m}{2}}\binom{K-l}{\frac{m}{2}}}{\binom{K}{m}} = 1 - 2 \sum_{j=\frac{m}{2}+1}^{m} \frac{\binom{l}{j}\binom{K-l}{m-j}}{\binom{K}{m}} - \frac{\binom{l}{\frac{m}{2}}\binom{K-l}{\frac{m}{2}}}{\binom{K}{m}}$$

$$= \gamma_{Sub}(l) \tag{28}$$

Now, combining Eq. 23, Eq. 24 and Eq. 27, if $m$ is odd, setting $\gamma_{Sub}$ as

$$\gamma_{Sub}(l) = \gamma_{Sub}(K - l) = 1 - 2 \sum_{j=\frac{m+1}{2}}^{m} \frac{\binom{l}{j}\binom{K-l}{m-j}}{\binom{K}{m}} \tag{29}$$

makes RR-d have the same output distribution as SubMaj.

Similarly, combining Eq. 25, Eq. 26 and Eq. 28, if $m$ is even, setting $\gamma_{Sub}$ as

$$\gamma_{Sub}(l) = \gamma_{Sub}(K - l) = 1 - 2 \sum_{j=\frac{m}{2}+1}^{m} \frac{\binom{l}{j}\binom{K-l}{m-j}}{\binom{K}{m}} - \frac{\binom{l}{\frac{m}{2}}\binom{K-l}{\frac{m}{2}}}{\binom{K}{m}} \tag{30}$$

makes RR-d have the same output distribution as SubMaj.

$\square$

### A.3    Proof of Lemma 3.2

**Lemma A.3** (Restatement of Lemma 3.2). *Let $\mathcal{A}$ be an $(\epsilon, \delta)$-differentially private algorithm, where $\epsilon \in (0, \frac{1}{2})$ and $\delta \in [0, \frac{1}{2})$, that computes the majority of $K$ $(\epsilon, \delta)$-differentially private mechanisms $M_1, \ldots, M_K$, where $M_i : \mathcal{D} \to \{0, 1\}$ on dataset $\mathcal{D}$ and $\Pr[M_i(\mathcal{D}) = 1] = p_i, \forall i \in [K]$. Then, the error $\mathcal{E}(\mathcal{A}) \geq |\Pr[g(\mathcal{S}) = 1] - \frac{1}{K}\sum_{i=1}^{K} p_i|$, where $g(\mathcal{S})$ is the probability of the true majority output being 1 as defined in Definition 1.1.*

*Proof.* Consider the setting where $M_i$'s are i.i.d., i.e., $\Pr[M_i(\mathcal{D}) = 1] = p, \forall i \in [K]$ for some $p \in [0, 1]$ on any dataset $\mathcal{D}$. Then, it suffices to show $\mathcal{E}(\mathcal{A}) \geq |\Pr[g(\mathcal{S})] = 1 - p|$, because a lower bound in this special case would indicate a lower bound for the more general case, where $p_i$'s can be different.

Construct a dataset $\mathcal{D}_0$ and $K$ mechanisms $\{M_i\}_{i=1}^K$ such that $\Pr[\mathcal{M}_i(\mathcal{D}_0) = 1] = \Pr[\mathcal{M}_i(\mathcal{D}_0) = 0] = \frac{1}{2}$ and without loss of generality, we may assume $\Pr[\mathcal{A}(\mathcal{D}_0) = 1] \leq \frac{1}{2}$.

Next, we construct a sequence of datasets $\mathcal{D}_1, \mathcal{D}_2, \ldots, \mathcal{D}_L$, such that $\mathcal{D}_j$ and $\mathcal{D}_{j+1}$ are neighboring datasets tha t differ in one entry, for all $j \in [L-1]$, and $\Pr[M_i(\mathcal{D}_j) = 1] = \frac{1}{2}e^{j\epsilon} + \sum_{l=0}^{j-1} e^{l\epsilon}\delta, \forall i \in [K], \forall j \in [L]$. Choose $L \in \mathbb{N}$ such that $\frac{1}{2}e^{L\epsilon} + \sum_{l=0}^{L-1} e^{l\epsilon}\delta = p$, for some $1 \geq p > \frac{1}{2}$.

Now, by definition of differential privacy,

$$\Pr[\mathcal{A}(\mathcal{D}_1) = 1] \leq e^{\epsilon}\Pr[\mathcal{A}(\mathcal{D}_0) = 1] + \delta$$
$$\Pr[\mathcal{A}(\mathcal{D}_2) = 1] \leq e^{\epsilon}\Pr[\mathcal{A}(\mathcal{D}_1) = 1] + \delta \leq e^{2\epsilon}\Pr[\mathcal{A}(\mathcal{D}_0) = 1] + e^{\epsilon}\delta + \delta$$
$$\cdots$$
$$\Pr[\mathcal{A}(\mathcal{D}_L) = 1] \leq e^{L\epsilon}\Pr[\mathcal{A}(\mathcal{D}_0) = 1] + \sum_{l=0}^{L-1} e^{\epsilon l}\delta \leq e^{L\epsilon}\frac{1}{2} + \sum_{l=0}^{L-1} e^{\epsilon l}\delta = p$$

Since the probability of true majority being 1 on dataset $\mathcal{D}_L$ is $\Pr[g(\mathcal{S}) = 1] \geq p > \frac{1}{2}$, there is

$$\mathcal{E}(\mathcal{A}) = |\Pr[g(\mathcal{S}) = 1] - \Pr[\mathcal{A}(\mathcal{D}_L) = 1]| \geq \Pr[g(\mathcal{S}) = 1] - p$$

$\square$

## A.4 Proof of Lemma 3.3

**Lemma A.4** (Restatement of Lemma 3.3)**.** *Let $\mathcal{A}$ be any randomized algorithm to compute the majority function $g$ on $\mathcal{S}$ such that for all $\mathcal{S}$, $\Pr[\mathcal{A}(\mathcal{S}) = g(\mathcal{S})] \geq 1/2$ (i.e. $\mathcal{A}$ is at least as good as a random guess). Then, there exists a a general function $\gamma : \{0,1\}^{K+1} \to [0,1]$ such that if one sets $p_\gamma$ by $\gamma(\mathcal{S})$ in DaRRM, the output distribution of DaRRM$_\gamma$ is the same as the output distribution of $\mathcal{A}$.*

*Proof of Lemma 3.3.* For some $\mathcal{D}$ and conditioned on $\mathcal{S}$, we see that by definition $\Pr[\text{DaRRM}_\gamma(\mathcal{S}) = g(\mathcal{S})] = \gamma(\mathcal{S}) + (1/2)(1 - \gamma(\mathcal{S}))$. We want to set $\gamma$ such that $\Pr[\text{DaRRM}_\gamma(\mathcal{S}) = g(\mathcal{S})] = \Pr[\mathcal{A}(\mathcal{S}) = g(\mathcal{S})]$. Therefore, we set $\gamma(\mathcal{S}) = 2\Pr[\mathcal{A}(\mathcal{S}) = g(\mathcal{S})] - 1$.

Lastly, we need to justify that $\gamma \in [0,1]$. Clearly, $\gamma(\mathcal{S}) \leq 2 - 1 \leq 1$ since $\Pr[\mathcal{A}(\mathcal{S}) = g(\mathcal{S})] \leq 1$. Note that the non-negativity follows from assumption. $\qquad\square$

## A.5 Proof of Lemma 3.4

**Lemma A.5** (Restatement of Lemma 3.4)**.** *Consider using DaRRM (Algorithm 1) to solve Problem 1.1, let $\alpha_l = \Pr[\mathcal{L}(\mathcal{D}) = l]$ and $\alpha'_l = \Pr[\mathcal{L}(\mathcal{D}') = l]$, where $\mathcal{D}$ and $\mathcal{D}'$ are adjacent datasets and $l \in \{0, \dots, K\}$. For a noise function $\gamma : \{0, 1, \dots, K\} \to [0,1]$ such that $\gamma(l) = \gamma(K - l), \forall l$, DaRRM$_\gamma$ is $(m\epsilon, \delta)$-differentially private if and only if for all $\alpha_l, \alpha'_l$, the following holds,*

$$f(p_1, \dots, p_K, p'_1, \dots, p'_K; \gamma) \leq e^{m\epsilon} - 1 + 2\delta \tag{31}$$

*where $f$ is called the **privacy cost objective** and*

$$f(p_1, \dots, p_K, p'_1, \dots, p'_K; \gamma) := \sum_{l=0}^{\frac{K-1}{2}} (e^{m\epsilon} \alpha'_l - \alpha_l) \cdot \gamma(l) + \sum_{l=\frac{K+1}{2}}^{K} (\alpha_l - e^{m\epsilon} \alpha'_l) \cdot \gamma(l)$$

*Proof of Lemma 3.4.* By the definition of differential privacy (Definition 2.1),

$$\text{DaRRM}_\gamma \text{ is } (m\epsilon, \delta)\text{-differentially private}$$
$$\iff \Pr[\text{DaRRM}_\gamma(\mathcal{D}) = 1] \leq e^{m\epsilon} \Pr[\text{DaRRM}_\gamma(\mathcal{D}') = 1] + \delta,$$
$$\text{and } \Pr[\text{DaRRM}_\gamma(\mathcal{D}) = 0] \leq e^{m\epsilon} \Pr[\text{DaRRM}_\gamma(\mathcal{D}') = 0] + \delta, \quad \forall \text{ adjacent datasets } \mathcal{D}, \mathcal{D}' \tag{32}$$

Let random variables $\mathcal{L}(\mathcal{D}) = \sum_{i=1}^{K} S(\mathcal{D})$ and $\mathcal{L}(\mathcal{D}') = \sum_{i=1}^{K} S(\mathcal{D}')$ be the sum of observed outcomes on adjacent datasets $\mathcal{D}$ and $\mathcal{D}'$, based on which one sets $p_\gamma$ in DaRRM. Let $\alpha_l = \Pr[\mathcal{L}(\mathcal{D}) = l]$ and $\alpha'_l = \Pr[\mathcal{L}(\mathcal{D}') = l], \forall l \in \{0, 1, \dots, K\}$.

Consider the output being 1.

$$\Pr[\text{DaRRM}_\gamma(\mathcal{D}) = 1] \leq e^{m\epsilon} \Pr[\text{DaRRM}_\gamma(\mathcal{D}') = 1] + \delta \tag{33}$$

$$\iff \sum_{l=0}^{K} \Pr[\text{DaRRM}_\gamma(\mathcal{D}) = 1 \mid \mathcal{L}(\mathcal{D}) = l] \cdot \Pr[\mathcal{L}(\mathcal{D}) = l] \tag{34}$$

$$\leq e^{m\epsilon} \Big( \sum_{l=0}^{K} \Pr[\text{DaRRM}_\gamma(\mathcal{D}') = 1 \mid \mathcal{L}(\mathcal{D}') = l] \cdot \Pr[\mathcal{L}(\mathcal{D}') = l] \Big) + \delta$$

$$\iff \sum_{l=0}^{K} \Big( \gamma(l) \cdot \mathbb{I}\{l \geq \frac{K}{2}\} + \frac{1}{2}(1 - \gamma(l)) \Big) \cdot \Pr[\mathcal{L}(\mathcal{D}) = l] \tag{35}$$

$$\leq e^{m\epsilon} \Big( \sum_{l=0}^{K} \Big( \gamma(l) \cdot \mathbb{I}\{l \geq \frac{K}{2}\} + \frac{1}{2}(1 - \gamma(l))\} \Big) \cdot \Pr[\mathcal{L}(\mathcal{D}') = l] \Big) + \delta$$

$$\Longleftrightarrow \sum_{l=\frac{K+1}{2}}^{K} \Big(\gamma(l) + \frac{1}{2}(1-\gamma(l))\Big) \cdot \Pr[\mathcal{L}(\mathcal{D}) = l] + \sum_{l=0}^{\frac{K-1}{2}} \frac{1}{2}(1-\gamma(l)) \cdot \Pr[\mathcal{L}(\mathcal{D}) = l] \tag{36}$$

$$\leq e^{m\epsilon} \Big( \sum_{l=\frac{K+1}{2}}^{K} \Big(\gamma(l) + \frac{1}{2}(1-\gamma(l))\Big) \cdot \Pr[\mathcal{L}(\mathcal{D}) = l]\Big) + e^{m\epsilon} \Big( \sum_{l=0}^{\frac{K-1}{2}} \frac{1}{2}(1-\gamma(l)) \cdot \Pr[\mathcal{L}(\mathcal{D}') = l]\Big) + \delta$$

$$\Longleftrightarrow \sum_{l=\frac{K+1}{2}}^{K} \frac{1}{2}\gamma(l)\alpha_l - \sum_{l=0}^{\frac{K-1}{2}} \frac{1}{2}\gamma(l)\alpha_l + \frac{1}{2} \tag{37}$$

$$\leq e^{m\epsilon} \sum_{l=\frac{K+1}{2}}^{K} \frac{1}{2}\gamma(l)\alpha_l' - e^{m\epsilon} \sum_{l=0}^{\frac{K-1}{2}} \frac{1}{2}\gamma(l)\alpha_l' + \frac{1}{2}e^{m\epsilon} + \delta$$

$$\Longleftrightarrow \sum_{l=\frac{K+1}{2}}^{K} (\alpha_l - e^{m\epsilon}\alpha_l')\gamma(l) - \sum_{l=0}^{\frac{K-1}{2}} (\alpha_l - e^{m\epsilon}\alpha_l')\gamma(l) \leq e^{m\epsilon} - 1 + 2\delta \tag{38}$$

Similarly, consider the output being 0.

$$\Pr[\mathsf{DaRRM}_\gamma(\mathcal{D}) = 0] \leq e^{m\epsilon} \Pr[\mathsf{DaRRM}_\gamma(\mathcal{D}') = 0] + \delta \tag{39}$$

$$\Longleftrightarrow \sum_{l=0}^{K} \Pr[\mathsf{DaRRM}_\gamma(\mathcal{D}) = 0 \mid \mathcal{L}(\mathcal{D}) = l] \cdot \Pr[\mathcal{L}(\mathcal{D}) = l] \tag{40}$$

$$\leq e^{m\epsilon} \Big( \sum_{l=0}^{K} \Pr[\mathsf{DaRRM}_\gamma(\mathcal{D}') = 0 \mid \mathcal{L}(\mathcal{D}') = l] \cdot \Pr[\mathcal{L}(\mathcal{D}') = l]\Big) + \delta$$

$$\Longleftrightarrow \sum_{l=0}^{K} \Big(\gamma(l) \cdot \mathbb{I}\{l < \frac{K}{2}\} + \frac{1}{2}(1-\gamma(l))\Big) \cdot \Pr[\mathcal{L}(\mathcal{D}) = l] \tag{41}$$

$$\leq e^{m\epsilon} \Big( \sum_{l=0}^{K} \gamma(l) \cdot \mathbb{I}\{l < \frac{K}{2}\} + \frac{1}{2}(1-\gamma(l))\Big) \cdot \Pr[\mathcal{L}(\mathcal{D}') = l] + \delta$$

$$\Longleftrightarrow \sum_{l=0}^{\frac{K-1}{2}} \Big(\gamma(l) + \frac{1}{2}(1-\gamma(l))\Big) \cdot \Pr[\mathcal{L}(\mathcal{D}) = l] + \sum_{l=\frac{K+1}{2}}^{K} \frac{1}{2}(1-\gamma(l)) \cdot \Pr[\mathcal{L}(\mathcal{D}) = l] \tag{42}$$

$$\leq e^{m\epsilon} \Big( \sum_{l=0}^{\frac{K-1}{2}} \Big(\gamma(l) + \frac{1}{2}(1-\gamma(l))\Big) \cdot \Pr[\mathcal{L}(\mathcal{D}') = l] + \sum_{l=\frac{K+1}{2}}^{K} \frac{1}{2}(1-\gamma(l)) \cdot \Pr[\mathcal{L}(\mathcal{D}') = l]\Big) + \delta$$

$$\Longleftrightarrow \sum_{l=0}^{\frac{K-1}{2}} \frac{1}{2}\gamma(l)\alpha_l - \sum_{l=\frac{K+1}{2}}^{K} \frac{1}{2}\gamma(l)\alpha_l + \frac{1}{2} \tag{43}$$

$$\leq e^{m\epsilon} \sum_{l=0}^{\frac{K-1}{2}} \frac{1}{2}\gamma(l)\alpha_l' - e^{m\epsilon} \sum_{l=\frac{K+1}{2}}^{K} \frac{1}{2}\gamma(l)\alpha_l' + \frac{1}{2}e^{m\epsilon} + \delta$$

$$\Longleftrightarrow \sum_{l=0}^{\frac{K-1}{2}} (\alpha_l - e^{m\epsilon}\alpha_l')\gamma(l) - \sum_{l=\frac{K+1}{2}}^{K} (\alpha_l - e^{m\epsilon}\alpha_l')\gamma(l) \leq e^{m\epsilon} - 1 + 2\delta \tag{44}$$

Therefore, plugging Eq. 38 and Eq. 44 into Eq. 32,

$$\text{DaRRM}_\gamma \text{ is } (m\epsilon, \delta)\text{-differentially private}$$

$$\iff \sum_{l=\frac{K+1}{2}}^{K} (\alpha_l - e^{m\epsilon}\alpha_l')\gamma(l) - \sum_{l=0}^{\frac{K-1}{2}} (\alpha_l - e^{m\epsilon}\alpha_l')\gamma(l) \leq e^{m\epsilon} - 1 + 2\delta \tag{45}$$

$$\text{and} \sum_{l=0}^{\frac{K-1}{2}} (\alpha_l - e^{m\epsilon}\alpha_l')\gamma(l) - \sum_{l=\frac{K+1}{2}}^{K} (\alpha_l - e^{m\epsilon}\alpha_l')\gamma(l) \leq e^{m\epsilon} - 1 + 2\delta \tag{46}$$

where $\alpha_l = \Pr[\mathcal{L}(\mathcal{D}) = l]$ and $\alpha_l' = \Pr[\mathcal{L}(\mathcal{D}') = l]$, $\forall l \in \{0, 1, \dots, K\}$ and $\mathcal{D}, \mathcal{D}'$ are any adjacent datasets.

Next, we show if $\gamma$ is symmetric around $\frac{K}{2}$, i.e., $\gamma(l) = \gamma(K - l)$, satisfying either one of Eq. 45 or Eq. 46 implies satisfying the other one. Following Eq. 45,

$$\sum_{l=\frac{K+1}{2}}^{K} (\alpha_l - e^{m\epsilon}\alpha_l')\gamma(l) - \sum_{l=0}^{\frac{K-1}{2}} (\alpha_l - e^{m\epsilon}\alpha_l')\gamma(l) \leq e^{m\epsilon} - 1 + 2\delta \tag{47}$$

$$\iff \sum_{l=0}^{\frac{K-1}{2}} (\alpha_{K-l} - e^{m\epsilon}\alpha_{K-l}') \cdot \gamma(K-l) - \sum_{l=\frac{K-1}{2}}^{K} (\alpha_{K-l} - e^{m\epsilon}\alpha_{K-l}') \cdot \gamma(K-l) \leq e^{m\epsilon} - 1 + 2\delta \tag{48}$$

$$\iff \sum_{l=0}^{\frac{K-1}{2}} (\alpha_{K-l} - e^{m\epsilon}\alpha_{K-l}') \cdot \gamma(l) - \sum_{l=\frac{K-1}{2}}^{K} (\alpha_{K-l} - e^{m\epsilon}\alpha_{K-l}') \cdot \gamma(l) \leq e^{m\epsilon} - 1 + 2\delta \tag{49}$$

Since $\gamma(l) = \gamma(K - l)$

For analysis purpose, we rewrite Eq. 46 as

$$\sum_{l=0}^{\frac{K-1}{2}} (\widetilde{\alpha}_l - e^{m\epsilon}\widetilde{\alpha}_l') \cdot \gamma(l) - \sum_{l=\frac{K-1}{2}}^{K} (\widetilde{\alpha}_l - e^{m\epsilon}\widetilde{\alpha}_l') \cdot \gamma(l) \leq e^{m\epsilon} - 1 + 2\delta \tag{50}$$

and proceed by showing Eq. 49 $\iff$ Eq. 50.

Recall $p_i = \Pr[M_i(\mathcal{D}) = 1]$ and $p_i' = \Pr[M_i(\mathcal{D}') = 1]$. Observe $\mathcal{L}(\mathcal{D}) \sim \text{PoissonBinomial}(\{p_i\}_{i=1}^{K})$ and $\mathcal{L}(\mathcal{D}') \sim \text{PoissonBinomial}(\{p_i'\}_{i=1}^{K})$. Let $F_l = \{\mathcal{A} : |\mathcal{A}| = l, \mathcal{A} \subseteq [K]\}$, for any $l \in \{0, \dots, K\}$, denote the set of all subsets of $l$ integers that can be selected from $[K]$. Let $\mathcal{A}^c = [K] \setminus \mathcal{A}$ be $\mathcal{A}$'s complement set. Notice $F_{K-l} = \{\mathcal{A}^c : \mathcal{A} \in F_l\}$.

Since $\alpha$ denotes the pmf of the Poisson Binomial distribution at $l$, it follows that

$$\alpha_l = \Pr[\mathcal{L}(\mathcal{D}) = l] = \sum_{\mathcal{A} \in F_l} \Pi_{i \in \mathcal{A}} p_i \Pi_{j \in \mathcal{A}^c} (1 - p_j) \tag{51}$$

Consider $\beta_i = 1 - p_i, \forall i \in [K]$ and a new random variable $\mathcal{L}^\beta \sim \text{PoissonBinomial}(\{\beta_i\}_{i=1}^{K})$, and let $\widetilde{\alpha}_l = \Pr[\mathcal{L}^\beta = 1]$. Observe that

$$\widetilde{\alpha}_l' = \Pr[\mathcal{L}^\beta = l] = \sum_{\mathcal{A} \in F_l} \Pi_{j \in \mathcal{A}} \beta_i \Pi_{i \in \mathcal{A}^c} (1 - \beta_i) = \sum_{\mathcal{A} \in F_l} \Pi_{j \in \mathcal{A}} (1 - p_j) \Pi_{i \in \mathcal{A}^c} p_i$$

$$= \sum_{\mathcal{A}^c \in F_{K-l}} \Pi_{j \in \mathcal{A}} (1 - p_i) \Pi_{i \in \mathcal{A}^c} p_i = \sum_{\mathcal{A} \in F_{K-l}} \Pi_{i \in \mathcal{A}} p_i \Pi_{j \in \mathcal{A}^c} (1 - p_i)$$

$$= \alpha_{K-l} \tag{52}$$

Similarly, consider $\beta_i' = 1 - p_i', \forall i \in [K]$ and a new random variable $\mathcal{L}'^{\beta} \sim \text{PoissonBinomial}(\beta_i'\}_{i=1}^L)$, and let $\widetilde{\alpha}_l' = \Pr[\mathcal{L}'^{\beta} = 1]$. Then, $\widetilde{\alpha}_l' = \alpha'_{K-l}$.

Since Eq. 49 holds for all possible $\alpha_{K-l}$, $\alpha'_{K-l}$, Eq. 50 then holds for all $\widetilde{\alpha}_l, \widetilde{\alpha}_l'$ in the $K$-simplex, and so Eq. 50 follows by relabeling $\alpha_{K-l}$ as $\widetilde{\alpha}_l$ and $\alpha'_{K-l}$ as $\widetilde{\alpha}_l'$.

The above implies Eq. 45 $\iff$ Eq. 46. Therefore,

$$\mathsf{DaRRM}_\gamma \text{ is } (m\epsilon, \delta)\text{-differentially private}$$

$$\iff \underbrace{\sum_{l=\frac{K+1}{2}}^{K} (\alpha_l - e^{m\epsilon}\alpha_l')\gamma(l) - \sum_{l=0}^{\frac{K-1}{2}} (\alpha_l - e^{m\epsilon}\alpha_l')\gamma(l)}_{:= f(p_1,\dots,p_K,p_1',\dots,p_K';\gamma)} \leq e^{m\epsilon} - 1 + 2\delta \tag{53}$$

$\square$

## B  Details of Section 4: Provable Privacy Amplification

In this section, we consider Problem 1.1 in the pure differential privacy and i.i.d. mechanisms setting. That is, $\delta = \Delta = 0$ and $p = p_i = \Pr[M_i(\mathcal{D}) = 1], p' = p'_i = \Pr[M_i(\mathcal{D}') = 1], \forall i \in [K]$. Our goal is to search for a good noise function $\gamma$ such that: 1) $\mathsf{DaRRM}_\gamma$ is $m\epsilon$-DP, and 2) $\mathsf{DaRRM}_\gamma$ achieves higher utility than that of the baselines (see Section 3) under a fixed privacy loss. Our main finding of such a $\gamma$ function is presented in Theorem 4.1, which states given a privacy allowance $m \in [K]$, one can indeed output the majority of $2m - 1$ subsampled mechanisms, instead of just $m$ as indicated by simple composition. Later, we formally verify in Lemma B.11, Section B.3 that taking the majority of more mechanisms strictly increases the utility.

To start, by Lemma 3.4, for any noise function $\gamma$, $\gamma$ satisfying goal 1) is equivalent to satisfying

$$f(p, p'; \gamma) \leq e^\epsilon - 1 \tag{54}$$

where $f(p, p'; \gamma) = \sum_{l=0}^{\frac{K-1}{2}} (e^{m\epsilon} \alpha'_l - \alpha_l) \cdot \gamma(l) + \sum_{l=\frac{K+1}{2}}^{K} (\alpha_l - e^{m\epsilon} \alpha'_l) \cdot \gamma(l)$ refers to the privacy cost objective (see Lemma 3.4) in the i.i.d. mechanisms setting, and recall $\alpha_l = \Pr[\mathcal{L}(\mathcal{D}) = l]$ and $\alpha'_l = \Pr[\mathcal{L}(\mathcal{D}') = l]$, $\forall l \in \{0, 1, \ldots, K\}$. Notice in this setting, $\mathcal{L}(\mathcal{D}) \sim \text{Binomial}(p)$, and $\mathcal{L}(\mathcal{D}') \sim \text{Binomial}(p')$.

**Monotonicity Assumption.**  For analysis, we restrict our search for a $\gamma$ function with good utility to the class with a mild monotonicity assumption: $\gamma(l) \geq \gamma(l+1), \forall l \leq \frac{K-1}{2}$ and $\gamma(l) \leq \gamma(l+1), \forall l \geq \frac{K+1}{2}$. This matches our intuition that as $\mathcal{L}(\mathcal{D}) = \sum_{i=1}^{K} S_i$, i.e., the number of mechanisms outputting 1, approaches 0 or $K$, there is a clearer majority and so not much noise is needed to ensure privacy, which implies a larger value of $\gamma$.

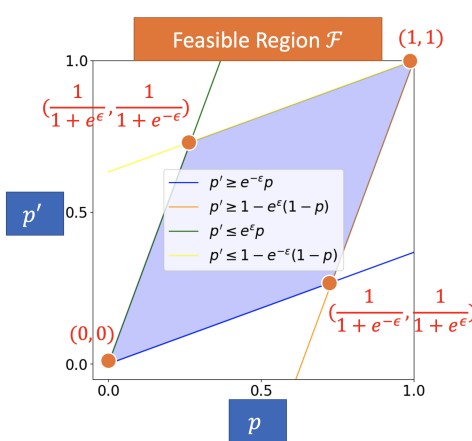

Figure 4: The feasible region $\mathcal{F}$ is plotted as the blue area. The four boundaries are implied by $p, p'$ satisfying $\epsilon$-differential privacy.

**Roadmap of Proof of Theorem 4.1.**  Since $\gamma$ needs to enable Eq. 54 to be satisfied for all $p, p' \in [0, 1]$, we begin by showing characteristics of **the worst case probabilities**, i.e., $(p^*, p'^*) = \arg\max_{(p,p')} f(p, p'; \gamma)$, given any $\gamma : \{0, 1, \ldots, K\} \to [0, 1]$ that is symmetric around $\frac{K}{2}$ and that satisfies the above monotonicity assumption, in Lemma B.1, Section B.1. We call $(p^*, p'^*)$ the worst case probabilities, since they incur the largest privacy loss. Later in Section B.2, we present the main proof of Theorem 4.1, where we focus on searching for a good $\gamma$ that enables $f(p^*, p'^*; \gamma) \leq e^\epsilon - 1$, based on the characteristics of $(p^*, p'^*)$ in Lemma B.1, to ensure $\mathsf{DaRRM}_\gamma$ is $m\epsilon$-differentially private.

### B.1  Characterizing the Worst Case Probabilities

First, note $(p, p')$ are close to each other and lie in a feasible region $\mathcal{F}$, due to each mechanism $M_i$ being $\epsilon$-differentially private; and so does $(p^*, p'^*)$. The feasible region, as illustrated in Figure 4, is bounded by (a) $p' \leq e^\epsilon p$ (b) $p \leq e^\epsilon p'$ (c) $1 - p' \leq e^\epsilon(1 - p)$, and (d) $1 - p \leq e^\epsilon(1 - p')$, where the four boundaries are derived from the definition of differential privacy. Therefore, we only need to search for $(p^*, p'^*) = \arg\max_{(p,p')\in\mathcal{F}} f(p, p'; \gamma)$.

Next, we show that given $\gamma$ satisfying certain conditions, $(p^*, p'^*)$ can only be on two of the four boundaries of $\mathcal{F}$ in Lemma B.1 — that is, either $p^* = e^\epsilon p'$, i.e., on the blue line in Figure 4, or $1 - p'^* = e^\epsilon(1 - p^*)$, i.e., on the orange line in Figure 4.

**Lemma B.1** (Characteristics of worst case probabilities)**.** *For any noise function $\gamma : \{0, 1, \ldots, K\} \to [0, 1]$ that is 1) symmetric around $\frac{K}{2}$, 2) satisfies the monotonicity assumption, and 3) $\gamma(\frac{K-1}{2}) > 0$ and $\gamma(\frac{K+1}{2}) > 0$, the worst case probabilities given $\gamma$, $(p^*, p'^*) = \arg\max_{(p,p')\in\mathcal{F}} f(p, p'; \gamma)$, must satisfy one of the following two equalities:*

$$p^* = e^\epsilon p'^*, \qquad\qquad \forall p^* \in [0, \frac{1}{e^{-\epsilon} + 1}], p'^* \in [0, \frac{1}{1 + e^\epsilon}]$$

$$or \quad 1 - p'^* = e^\epsilon(1 - p^*), \qquad\qquad \forall p^* \in [\frac{1}{1 + e^{-\epsilon}}, 1], p'^* \in [\frac{1}{1 + e^\epsilon}, 1]$$

To show Lemma B.1, we first show in Lemma B.2 that the search of $(p^*, p'^*)$ can be refined to one of the four boundaries of $\mathcal{F}$, via a careful gradient analysis of $f(p, p'; \gamma)$ in $\mathcal{F}$, and then show in Lemma B.3 that the search of $(p^*, p'^*)$ can be further refined to two of the four boundaries, due to symmetry of $p, p'$. Lemma B.1 directly follows from the two.

**Lemma B.2.** *For any noise function* $\gamma : \{0, 1, \ldots, K\} \to [0, 1]$ *that is 1) symmetric around* $\frac{K}{2}$*, 2) satisfies the monotonicity assumption, and 3)* $\gamma(\frac{K-1}{2}) > 0$ *and* $\gamma(\frac{K+1}{2}) > 0$*, the worst case probabilities given* $\gamma$*,* $(p^*, p'^*) = \arg\max_{(p,p') \in \mathcal{F}} f(p, p'; \gamma)$*, must satisfy one of the following four equalities:*

$$p'^* = e^\epsilon p^*, \qquad\qquad \forall p^* \in [0, \frac{1}{1 + e^\epsilon}], p'^* \in [0, \frac{1}{1 + e^{-\epsilon}}]$$

$$p^* = e^\epsilon p'^*, \qquad\qquad \forall p^* \in [0, \frac{1}{e^{-\epsilon} + 1}], p'^* \in [0, \frac{1}{1 + e^\epsilon}]$$

$$1 - p^* = e^\epsilon(1 - p'^*), \qquad\qquad \forall p^* \in [\frac{1}{1 + e^\epsilon}, 1], p'^* \in [\frac{1}{1 + e^{-\epsilon}}, 1]$$

$$1 - p'^* = e^\epsilon(1 - p^*), \qquad\qquad \forall p^* \in [\frac{1}{1 + e^{-\epsilon}}, 1], p'^* \in [\frac{1}{1 + e^\epsilon}, 1]$$

*Proof of Lemma B.2.* Recall the privacy cost objective (as defined in Lemma 3.4) is now

$$f(p, p'; \gamma) = \sum_{l=0}^{\frac{K-1}{2}} (e^{m\epsilon}\alpha'_l - \alpha_l) \cdot \gamma(l) + \sum_{l=\frac{K+1}{2}}^{K} (\alpha_l - e^{m\epsilon}\alpha'_l) \cdot \gamma(l)$$

where $\alpha_l = \Pr[\mathcal{L}(\mathcal{D}) = l]$ and $\alpha'_l = \Pr[\mathcal{L}(\mathcal{D}') = l]$, $\forall l \in \{0, 1, \ldots, K\}$. Since $\mathcal{L}(\mathcal{D}) \sim \text{Binomial}(p)$ and $\mathcal{L}(\mathcal{D}') \sim \text{Binomial}(p')$ in the i.i.d. mechanisms setting, and using the pmf of the Binomial distribution, $f$ can be written as

$$f(p, p'; \gamma) = \sum_{l=0}^{\frac{K-1}{2}} (e^{m\epsilon}\binom{K}{l}p'^l(1-p')^{K-l} - \binom{K}{l}p^l(1-p)^{K-l}) \cdot \gamma(l) + \sum_{l=\frac{K+1}{2}}^{K} (\binom{K}{l}p^l(1-p)^{K-l} - e^{m\epsilon}\binom{K}{l}p'^l(1-p')^{K-l})$$

The gradients w.r.t. $p$ and $p'$ are

$$\nabla_p f(p, p'; \gamma) = \underbrace{\sum_{l=0}^{\frac{K-1}{2}} -\binom{K}{l}\gamma(l) \cdot (lp^{l-1}(1-p)^{K-l} - p^l(K-l)(1-p)^{K-l-1})}_{:=A} \qquad (55)$$

$$+ \underbrace{\sum_{l=\frac{K+1}{2}}^{K} \binom{K}{l}\gamma(l) \cdot (lp^{l-1}(1-p)^{K-l} - p^l(K-l)(1-p)^{K-l-1})}_{:=B}$$

and

$$\nabla_{p'} f(p, p'; \gamma) = \sum_{l=0}^{\frac{K-1}{2}} e^{m\epsilon}\binom{K}{l}\gamma(l) \cdot (lp'^{l-1}(1-p')^{K-l} - p'^l(K-l)(1-p')^{K-l-1}) \qquad (56)$$

$$+ \sum_{l=\frac{K+1}{2}}^{K} -e^{m\epsilon}\binom{K}{l}\gamma(l) \cdot (lp'^{l-1}(1-p')^{K-l} - p'^l(K-l)(1-p')^{K-l-1})$$

We show in the following $\forall p \in (0,1)$, $\nabla_p f(p, p'; \gamma) > 0$ and $\nabla_{p'} f(p, p'; \gamma) < 0$. This implies there is no local maximum inside $\mathcal{F}$, and so $(p^*, p'^*) = \arg\max_{p,p'} f(p, p'; \gamma)$ must be on one of the four boundaries of $\mathcal{F}$. Also, if $p = 0$, then $p' = 0$, and $(0,0)$ is a corner point at the intersection of two boundaries. Similarly, if $p = 1$, then $p' = 1$, and $(1,1)$ is also a corner point. This concludes $\forall p \in [0,1]$, $(p^*, p'^*) = \arg\max_{p,p'} f(p, p'; \gamma)$ must be on one of the four boundaries of $\mathcal{F}$.

To show $\nabla_p f(p, p'; \gamma) > 0$ for $p \in (0,1)$, we write $\nabla_p f(p, p'; \gamma) = A + B$ as in Eq. 55, and show that $A > 0$ and $B > 0$.

To show $A > 0$, first note

$$A := \sum_{l=0}^{\frac{K-1}{2}} \gamma(l) \binom{K}{l} \cdot (p^l (K-l)(1-p)^{K-l-1} - l p^{l-1}(1-p)^{K-l}) > 0 \tag{57}$$

$$\iff \sum_{l=0}^{\frac{K-1}{2}} \gamma(l) \binom{K}{l} \cdot p^l (K-l)(1-p)^{K-l-1} > \sum_{l=0}^{\frac{K-1}{2}} \gamma(l) \binom{K}{l} \cdot l p^{l-1}(1-p)^{K-l}) \tag{58}$$

$$\iff \sum_{l=0}^{\frac{K-1}{2}} \gamma(l) \binom{K-1}{l} \frac{K}{K-l} \cdot p^l (K-l)(1-p)^{K-l-1} > \sum_{l=1}^{\frac{K-1}{2}} \gamma(l) \binom{K-1}{l-1} \frac{K}{l} \cdot l p^{l-1}(1-p)^{K-l} \tag{59}$$

$$\iff K \sum_{l=0}^{\frac{K-1}{2}} \gamma(l) \binom{K-1}{l} p^l (1-p)^{K-l-1} > K \sum_{l=1}^{\frac{K-1}{2}} \gamma(l) \binom{K-1}{l-1} p^{l-1}(1-p)^{K-l} \tag{60}$$

$$\iff \sum_{l=0}^{\frac{K-1}{2}} \gamma(l) \binom{K-1}{l} p^l (1-p)^{K-l-1} > \sum_{l=0}^{\frac{K-1}{2}-1} \gamma(l+1) \binom{K-1}{l} p^l (1-p)^{K-l-1} \tag{61}$$

Since $\forall l \leq \frac{K-1}{2}$, $\gamma(l) \geq \gamma(l+1)$ and $p \in (0,1)$, there is for $l \in \{0, \ldots, \frac{K-1}{2} - 1\}$,

$$\gamma(l) \binom{K-1}{l} p^l (1-p)^{K-l-1} \geq \gamma(l+1) \binom{K-1}{l} p^l (1-p)^{K-l-1} \tag{62}$$

Furthermore, since $\gamma(\frac{K-1}{2}) > 0$ and $p \in (0,1)$,

$$\gamma(\frac{K-1}{2}) \binom{K-1}{\frac{K-1}{2}} p^{\frac{K-1}{2}} (1-p)^{\frac{K-1}{2}} > 0 \tag{63}$$

Eq. 62 and Eq. 63 combined implies

$$\gamma(\frac{K-1}{2}) \binom{K-1}{\frac{K-1}{2}} p^{\frac{K-1}{2}} (1-p)^{\frac{K-1}{2}} + \sum_{l=0}^{\frac{K-1}{2}-1} \gamma(l) \binom{K-1}{l} p^l (1-p)^{K-l-1} > \sum_{l=0}^{\frac{K-1}{2}-1} \gamma(l+1) \binom{K-1}{l} p^l (1-p)^{K-l-1} \tag{64}$$

and hence, Eq. 61 holds. This further implies $A > 0$.

Next, to show $B > 0$, note that

$$B := \sum_{l=\frac{K+1}{2}}^{K} \binom{K}{l} \gamma(l) \cdot (l p^{l-1}(1-p)^{K-l} - p^l (K-l)(1-p)^{K-l-1}) > 0 \tag{65}$$

$$\iff \sum_{l=\frac{K+1}{2}}^{K} \binom{K}{l} \gamma(l) \cdot l p^{l-1}(1-p)^{K-l} > \sum_{l=\frac{K+1}{2}}^{K} \binom{K}{l} p^l (K-l)(1-p)^{K-l-1} \tag{66}$$

$$\iff \sum_{l=\frac{K+1}{2}}^{K} \gamma(l) \binom{K-1}{l-1} \frac{K}{l} \cdot l p^{l-1} (1-p)^{K-l} \tag{67}$$

$$> \sum_{l=\frac{K+1}{2}}^{K-1} \gamma(l) \binom{K-1}{l} \frac{K}{K-l} \cdot p^l (K-l)(1-p)^{K-l-1}$$

$$\iff K \sum_{l=\frac{K+1}{2}}^{K} \gamma(l) \binom{K-1}{l-1} \cdot p^{l-1} (1-p)^{K-l} \tag{68}$$

$$> K \sum_{l=\frac{K+1}{2}}^{K-1} \gamma(l) \binom{K-1}{l} \cdot p^l (1-p)^{K-l-1}$$

$$\iff \sum_{l=\frac{K+1}{2}}^{K} \gamma(l) \binom{K-1}{l-1} \cdot p^{l-1} (1-p)^{K-l} > \sum_{l=\frac{K+1}{2}+1}^{K} \gamma(l-1) \binom{K-1}{l-1} \cdot p^{l-1} (1-p)^{K-l} \tag{69}$$

Since $\forall l \geq \frac{K+1}{2}$, $\gamma(l) \geq \gamma(l-1)$ and $p \in (0,1)$, there is for $l \in \{\frac{K+1}{2}+1, \ldots, K\}$,

$$\gamma(l) \binom{K-1}{l-1} p^{l-1} (1-p)^{K-l} \geq \gamma(l-1) \binom{K-1}{l-1} p^{l-1} (1-p)^{K-l} \tag{70}$$

Furthermore, since $\gamma(\frac{K+1}{2}) > 0$ and $p \in (0,1)$,

$$\gamma(\frac{K+1}{2}) \binom{K-1}{\frac{K-1}{2}} p^{\frac{K-1}{2}} (1-p)^{\frac{K-1}{2}} > 0 \tag{71}$$

Eq. 70 and Eq. 71 combined implies

$$\gamma(\frac{K+1}{2}) \binom{K-1}{\frac{K-1}{2}} p^{\frac{K-1}{2}} (1-p)^{\frac{K-1}{2}} + \sum_{l=\frac{K+1}{2}+1}^{K} \gamma(l) \binom{K-1}{l-1} \cdot p^{l-1} (1-p)^{K-l} > \sum_{l=\frac{K+1}{2}+1}^{K} \gamma(l-1) \binom{K-1}{l-1} \cdot p^{l-1} (1-p)^{K-l} \tag{72}$$

and hence Eq. 69 holds. This further implies $B > 0$.

Following Eq.55, for $p \in (0,1)$ and $\gamma$ satisfying the three assumptions,

$$\nabla_p f(p, p'; \gamma) = A + B > 0 \tag{73}$$

Following similar techniques, one can show for $p \in (0,1)$ and $\gamma$ satisfying the three conditions,

$$\nabla_{p'} f(p, p'; \gamma) < 0 \tag{74}$$

This implies there is no local minima or local maxima inside the feasible region $\mathcal{F}$. Also recall $(p, p') \in \{(0,0), (1,1)\}$ are two special cases where $(p, p')$ is at the intersection of two boundaries. Hence, we conclude the worst case probability $(p^*, p'^*) = \arg\max_{p,p' \in \mathcal{F}} f(p, p'; \gamma)$ is on one of the four boundaries of $\mathcal{F}$ — that is, $(p^*, p'^*)$ satisfy one of the following:

$$p'^* = e^\epsilon p^*, \qquad \forall p \in [0, \frac{1}{1+e^\epsilon}], p' \in [0, \frac{1}{1+e^{-\epsilon}}]$$

$$p^* = e^\epsilon p'^*, \qquad \forall p \in [0, \frac{1}{e^{-\epsilon}+1}], p' \in [0, \frac{1}{1+e^\epsilon}]$$

$$1 - p^* = e^\epsilon (1 - p'^*), \qquad \forall p \in [\frac{1}{1+e^\epsilon}, 1], p' \in [\frac{1}{1+e^{-\epsilon}}, 1]$$

$$1 - p'^* = e^\epsilon (1 - p^*), \qquad \forall p \in [\frac{1}{1+e^{-\epsilon}}, 1], p' \in [\frac{1}{1+e^\epsilon}, 1]$$

$\square$

**Lemma B.3.** *For any noise function $\gamma : \{0, 1, \ldots, K\} \to [0, 1]$ function that is 1) symmetric around $\frac{K}{2}$ and 2) satisfies the monotonicity assumption, the privacy cost objective $f(p, p'; \gamma)$ is maximized when $p \geq p'$.*

*Proof of Lemma B.3.* Following Eq. 33 and Eq. 38 in the proof of Lemma 3.4, and that $\delta = 0$,

$$\Pr[\mathsf{DaRRM}_\gamma(\mathcal{D}) = 1] \leq e^{m\epsilon} \Pr[\mathsf{DaRRM}_\gamma(\mathcal{D}') = 1] \tag{75}$$

$$\iff \underbrace{\sum_{l=\frac{K+1}{2}}^{K} (\alpha_l - e^{m\epsilon}\alpha_l')\gamma(l) - \sum_{l=0}^{\frac{K-1}{2}} (\alpha_l - e^{m\epsilon}\alpha_l')\gamma(l) \leq e^{m\epsilon} - 1}_{=f(p,p';\gamma)} \tag{76}$$

where $\alpha_l = \Pr[\mathcal{L}(\mathcal{D}) = l]$ and $\alpha_l' = \Pr[\mathcal{L}(\mathcal{D}') = l]$, $\forall l \in \{0, 1, \ldots, K\}$. This implies

$$f(p, p'; \gamma) = \frac{\Pr[\mathsf{DaRRM}_\gamma(\mathcal{D}) = 1]}{\Pr[\mathsf{DaRRM}_\gamma(\mathcal{D}') = 1]} - 1 \tag{77}$$

Hence, $f(p, p'; \gamma)$ is maximized when $\Pr[\mathsf{DaRRM}_\gamma(\mathcal{D}) = 1] \geq \Pr[\mathsf{DaRRM}_\gamma(\mathcal{D}') = 1]$.

$$\Pr[\mathsf{DaRRM}_\gamma(\mathcal{D}) = 1] = \sum_{l=0}^{K} \Pr[\mathsf{DaRRM}_\gamma(\mathcal{D}) = 1 \mid \mathcal{L}(\mathcal{D}) = 1] \cdot \Pr[\mathcal{L}(\mathcal{D}) = l] \tag{78}$$

$$= \sum_{l=0}^{K} \left( \gamma(l) \cdot \mathbb{I}\{l \geq \frac{K}{2}\} + \frac{1}{2}(1 - \gamma(l)) \right) \cdot \Pr[\mathcal{L}(\mathcal{D}) = l] \tag{79}$$

$$= \sum_{l=0}^{\frac{K-1}{2}} \frac{1}{2}(1 - \gamma(l)) \cdot \alpha_l + \sum_{l=\frac{K+1}{2}}^{K} \left( \gamma(l) + \frac{1}{2}(1 - \gamma(l)) \right) \cdot \alpha_l \tag{80}$$

$$= \frac{1}{2} \sum_{l=\frac{K+1}{2}}^{K} \gamma(l) \binom{K}{l} p^l (1-p)^{K-l} - \frac{1}{2} \sum_{l=0}^{\frac{K-1}{2}} \gamma(l) \binom{K}{l} p^l (1-p)^{K-l-1} + \frac{1}{2} \tag{81}$$

where the last line follows from the observation that in the i.i.d. mechanisms setting, $\mathcal{L}(\mathcal{D}) \sim \text{Binomial}(p)$ and $\alpha_l$ is hence the pmf of the Binomial distribution at $l$.

Similarly,

$$\Pr[\mathsf{DaRRM}_\gamma(\mathcal{D}') = 1] = \frac{1}{2} \sum_{l=\frac{K+1}{2}}^{K} \gamma(l) \binom{K}{l} p'^l (1-p')^{K-l} - \frac{1}{2} \sum_{l=0}^{\frac{K-1}{2}} \gamma(l) \binom{K}{l} p'^l (1-p')^{K-l-1} + \frac{1}{2} \tag{82}$$

Now define the objective

$$h(\beta) = \frac{1}{2} \sum_{l=\frac{K+1}{2}}^{K} \gamma(l) \binom{K}{l} \beta^l (1-\beta)^{K-l} - \frac{1}{2} \sum_{l=0}^{\frac{K-1}{2}} \gamma(l) \binom{K}{l} \beta^l (1-\beta)^{K-l-1} + \frac{1}{2} \tag{83}$$

for $\beta \in [0, 1]$ and it follows that $\Pr[\mathsf{DaRRM}_\gamma(\mathcal{D}) = 1] = h(p)$ and $\Pr[\mathsf{DaRRM}_\gamma(\mathcal{D}') = 1] = h(p')$. We now analyze the monotonicity of $h(\beta)$ in $\beta$.

For ease of presentation, define $g(l) := \begin{cases} -\frac{1}{2}\gamma(l) & \forall l \leq \frac{K}{2} \\ \frac{1}{2}\gamma(l) & \forall l \geq \frac{K}{2} \end{cases}$. Since $\gamma(l) \geq \gamma(l+1), \forall l \leq \frac{K}{2}$ and $\gamma(l+1) \geq \gamma(l), \forall l \geq \frac{K}{2}$, there is $g(l+1) \geq g(l), \forall l \in \{0, \ldots, K\}$. And replacing $\gamma(l)$ with $g(l)$ in Eq. 83,

$$h(\beta) = \sum_{l=0}^{K} g(l) \binom{K}{l} \beta^l (1-\beta)^{K-l} \tag{84}$$

$$\nabla_\beta h(\beta) = \sum_{l=0}^{K} g(l) \binom{K}{l} \left( l\beta^{l-1}(1-\beta)^{K-l} - (K-l)\beta^l(1-\beta)^{K-l-1} \right) \tag{85}$$

$$= \sum_{l=1}^{K} g(l) \binom{K-1}{l-1} \frac{K}{l} l\beta^{l-1}(1-\beta)^{K-l} - \sum_{l=0}^{K-1} \binom{K-1}{l} \frac{K}{K-l}(K-l)\beta^l(1-\beta)^{K-l-1} \tag{86}$$

$$= K \sum_{l=1}^{K} \binom{K-1}{l-1} \beta^{l-1}(1-\beta)^{K-l} - K \sum_{l=0}^{K-1} \binom{K-1}{l} \beta^l(1-\beta)^{K-l-1} \tag{87}$$

$$= K \sum_{l=0}^{K-1} g(l+1) \binom{K-1}{l} \beta^l(1-\beta)^{K-l-1} - K \sum_{l=0}^{K-1} g(l) \binom{K-1}{l} \beta^l(1-\beta)^{K-l-1} \tag{88}$$

$$= K \sum_{l=0}^{K-1} \left( g(l+1) - g(l) \right) \binom{K-1}{l} \beta^l(1-\beta)^{K-l-1} \tag{89}$$

Since $g(l+1) \geq g(l)$ and $\binom{K-1}{l}\beta^l(1-\beta)^{K-l-1} \geq 0$, $\nabla_\beta h(\beta) \geq 0$. This implies $h(\beta)$ is monotonically non-decreasing in $\beta$ and hence,

$$\Pr[\mathsf{DaRRM}_\gamma(\mathcal{D}) = 1] \geq \Pr[\mathsf{DaRRM}_\gamma(\mathcal{D}') = 1] \iff p \geq p' \tag{90}$$

Therefore, $f(p, p'; \gamma)$ is maximzied when $p \geq p'$. $\qquad\square$

## B.2 Proof of Privacy Amplification (Theorem 4.1)

**Theorem B.4** (Restatement of Theorem 4.1). *Consider using DaRRM (Algorithm 1) to solve Problem 1.1, with i.i.d. mechanisms $\{M_i\}_{i=1}^{K}$, i.e., $p_i = p$, $p_i' = p'$, $\forall i \in [K]$, the privacy allowance $m \in [K]$ and $\delta = \Delta = 0$. Let the noise function $\gamma : \{0, 1, \dots, K\} \to [0, 1]$ be that: if $m \geq \frac{K+1}{2}$,*

$$\gamma(l) = 1$$

*and if $m \leq \frac{K-1}{2}$,*

$$\gamma(l) = \begin{cases} 1 - 2h(l) & \forall l \leq \frac{K-1}{2} \\ 2h(l) - 1 & \forall l \geq \frac{K+1}{2} \end{cases}$$

*where $h(l) = \sum_{i=m}^{2m-1} \frac{\binom{l}{i}\binom{K-l}{2m-1-i}}{\binom{K}{2m-1}}$, then DaRRM$_\gamma$ is $m\epsilon$-differentially private.*

**Roadmap.** Theorem 4.1 consists of two parts: $\gamma$ under a large privacy allowance $m \geq \frac{K+1}{2}$ and $\gamma$ under a small privacy allowance $m \leq \frac{K-1}{2}$. We first show in Lemma B.5, Section B.2.1 that if $m \geq \frac{K+1}{2}$, setting $\gamma = 1$ suffices to ensure DaRRM$_\gamma$ to be $m\epsilon$-differentially private, and hence one can always output the true majority of $K$ mechanisms. In contrast, simple composition indicates only when $m = K$ can one output the true majority of $K$ mechanisms. Next, we show in Lemma B.10, Section B.2.2 that if $m \leq \frac{K-1}{2}$, one can set $\gamma$ to be $\gamma_{DSub}$, which corresponds to outputting the majority of $2m - 1$ subsampled mechanisms (and hence the name "Double Subsampling", or DSub). In contrast, simple compositon indicates one can only output the majority of $m$ subsampled mechanisms to make sure the output is $m\epsilon$-differentially private. **Theorem 4.1 follows directly from combining Lemma B.5 and Lemma B.10.**

### B.2.1 Privacy Amplification Under A Large Privacy Allowance $m \geq \frac{K+1}{2}$

The proof of Lemma B.5 is straightforward. We show that given the constant $\gamma_{max}(l) = 1$, if $m \geq \frac{K+1}{2}$, the worst case probabilities are $(p^*, p'^*) = \arg\max_{(p,p')\in\mathcal{F}} f(p, p'; \gamma_{max}) = (0, 0)$ and notice that $f(0, 0; \gamma_{max}) = e^{m\epsilon} - 1$, which satisfies the condition in Lemma 3.4. Hence, DaRRM$_{\gamma_{max}}$ is $m\epsilon$-differentially private.

**Lemma B.5** (Privacy amplification, $m \geq \frac{K+1}{2}$). *Consider using DaRRM (Algorithm 1) to solve Problem 1.1, with i.i.d. mechanisms $\{M_i\}_{i=1}^{K}$, i.e., $p_i = p$, $p_i' = p'$, $\forall i \in [K]$, the privacy allowance $m \geq \frac{K+1}{2}, m \in \mathbb{Z}$ and $\delta = \Delta = 0$. Let the noise function be the constant $\gamma_{max}(l) = 1, \forall l \in \{0, 1, \ldots, K\}$. Then, DaRRM$_{\gamma_{max}}$ is $m\epsilon$-differentially private.*

*Proof of Lemma B.5.* First, notice $\gamma_{max}(l) = 1, \forall l \in \{0, 1, \ldots, K\}$ is: 1) symmetric around $\frac{K}{2}$, 2) satisfies the monotonicity assumption, and 3) $\gamma_{max}(\frac{K-1}{2}) > 0$ and $\gamma_{max}(\frac{K+1}{2}) > 0$. Therefore, by Lemma B.1, the worst case probabilities given $\gamma_{max}$, i.e., $(p^*, p'^*) = \arg\max_{(p,p') \in \mathcal{F}} f(p, p'; \gamma_{max})$, are on one of the two boundaries of $\mathcal{F}$, satisfying

$$p^* = e^\epsilon p'^*, \qquad\qquad \forall p^* \in [0, \frac{1}{e^{-\epsilon}+1}], p'^* \in [0, \frac{1}{1+e^\epsilon}]$$

$$\text{or} \quad 1 - p'^* = e^\epsilon (1 - p^*), \qquad\qquad \forall p^* \in [\frac{1}{1+e^{-\epsilon}}, 1], p'^* \in [\frac{1}{1+e^\epsilon}, 1]$$

We now find the local maximums on the two possible boundaries, i.e.,

$$(p^*_{local}, p'^*_{local}) = \underset{(p,p'):p=e^\epsilon p', p \in [0, \frac{1}{e^{-\epsilon}+1}]}{\arg\max} f(p, p'; \gamma_{max})$$

and

$$(p^*_{local}, p'^*_{local}) = \underset{(p,p'):1-p'=e^\epsilon(1-p), p \in [\frac{1}{1+e^{-\epsilon}}, 1]}{\arg\max} f(p, p'; \gamma_{max})$$

separately.

**Part I: Local worst case probabilities on the boundary $p = e^\epsilon p'$.**

Plugging $p = e^\epsilon p'$ into the privacy cost objective $f(p, p'; \gamma_{max})$, one gets

$$f(p'; \gamma_{max}) = \sum_{l=0}^{\frac{K-1}{2}} (e^{m\epsilon} \binom{K}{l} p'^l (1-p')^{K-l} - \binom{K}{l} (e^\epsilon p')^l (1 - e^\epsilon p')^{K-l}) \tag{91}$$

$$+ \sum_{l=\frac{K+1}{2}}^{K} (\binom{K}{l} (e^\epsilon p')^l (1 - e^\epsilon p')^{K-l} - e^{m\epsilon} \binom{K}{l} p'^l (1-p')^{K-l})$$

The gradient w.r.t. $p'$ is

$$\nabla_{p'} f(p'; \gamma_{max}) = \sum_{l=0}^{\frac{K-1}{2}} \left( e^{m\epsilon} \binom{K}{l} (l p'^{l-1} (1-p')^{K-l} - p'^l (K-l)(1-p')^{K-l-1}) \right. \tag{92}$$

$$\left. - e^\epsilon \binom{K}{l} (l(e^\epsilon p')^{l-1}(1 - e^\epsilon p')^{K-l} - e^{\epsilon l} p'^l (K-l)(1 - e^\epsilon p')^{K-l-1}) \right)$$

$$+ \sum_{l=\frac{K+1}{2}}^{K} \left( e^\epsilon \binom{K}{l} (l(e^\epsilon p')^{l-1}(1 - e^\epsilon p')^{K-l} - e^{\epsilon l} p'^l (K-l)(1 - e^\epsilon p')^{K-l-1}) \right.$$

$$\left. - e^{m\epsilon} \binom{K}{l} (l p'^{l-1}(1-p')^{K-l} - p'^l (K-l)(1-p')^{K-l-1}) \right)$$

$$= -K \sum_{l=0}^{\frac{K-1}{2}} e^{m\epsilon} \binom{K-1}{l} p'^l (1-p')^{K-l-1} + K \sum_{l=\frac{K+1}{2}}^{K-1} e^{m\epsilon} \binom{K-1}{l} p'^l (1-p')^{K-l-1} \tag{93}$$

$$+ K \sum_{l=0}^{\frac{K-1}{2}} e^\epsilon \binom{K-1}{l} (\epsilon p')^\epsilon (1 - e^\epsilon p')^{K-l-1} - K \sum_{l=\frac{K+1}{2}}^{K-1} e^\epsilon \binom{K-1}{l} (e^\epsilon p')^l (1 - e^\epsilon p')^{K-l-1}$$

$$+ K \sum_{l=0}^{\frac{K-1}{2}-1} e^{m\epsilon} \binom{K-1}{l} p'^l (1-p')^{K-l-1} - K \sum_{l=\frac{K-1}{2}}^{K-1} e^{m\epsilon} \binom{K-1}{l} p'^l (1-p')^{K-l-1}$$

$$- K \sum_{l=0}^{\frac{K-1}{2}-1} e^{\epsilon} \binom{K-1}{l} (e^{\epsilon} p')^l (1-e^{\epsilon} p')^{K-l-1} + K \sum_{l=\frac{K-1}{2}}^{K-1} e^{\epsilon} \binom{K-1}{l} (e^{\epsilon} p')^l (1-e^{\epsilon} p')^{K-l-1}$$

$$= \underbrace{-2K \, e^{m\epsilon} \binom{K-1}{\frac{K-1}{2}} p'^{\frac{K-1}{2}} (1-p')^{\frac{K-1}{2}}}_{:=A} + \underbrace{2K \, e^{\epsilon} \binom{K-1}{\frac{K-1}{2}} (e^{\epsilon} p')^{\frac{K-1}{2}} (1-e^{\epsilon} p')^{\frac{K-1}{2}}}_{:=B} \qquad (94)$$

Notice that

$$\frac{A}{B} = \frac{e^{m\epsilon} \binom{K-1}{\frac{K-1}{2}} p'^{\frac{K-1}{2}} (1-p')^{\frac{K-1}{2}}}{e^{\epsilon} \binom{K-1}{\frac{K-1}{2}} (e^{\epsilon} p')^{\frac{K-1}{2}} (1-e^{\epsilon} p')^{\frac{K-1}{2}}} = \frac{e^{m\epsilon}}{e^{\frac{K+1}{2}\epsilon}} \cdot \left( \frac{1-p'}{1-e^{\epsilon} p'} \right)^{\frac{K-1}{2}} \qquad (95)$$

Since $\frac{1-p'}{1-e^{\epsilon} p'} \geq 1$ and $m \geq \frac{K+1}{2}$, $\frac{A}{B} \geq 1$. This implies $\nabla_{p'} f(p'; \gamma_{max}) \leq 0$. Hence, $f(p'; \gamma_{max})$ is monotonically non-increasing on the boundary, for $p' \in [0, \frac{1}{1+e^{\epsilon}}]$.

Therefore, $\arg\max_{p': p' \in [0, \frac{1}{1+e^{\epsilon}}]} f(p'; \gamma_{max}) = 0$. Since $p = e^{\epsilon} p'$, $p' = 0$ implies $p = 0$.

Hence,

$$(p^*_{local}, p'^*_{local}) = \arg\max_{(p,p'): p=e^{\epsilon} p', p \in [0, \frac{1}{e^{-\epsilon}+1}]} f(p, p'; \gamma_{max}) = (0, 0)$$

and

$$\max_{(p,p'): p=e^{\epsilon} p', p \in [0, \frac{1}{e^{-\epsilon}+1}]} f(p, p'; \gamma_{max}) = f(0, 0; \gamma_{max}) = e^{m\epsilon} - 1$$

**Part II: Local worst case probabilities on the boundary** $1 - p' = e^{\epsilon}(1 - p)$.

For simplicity, let $q = 1 - p$ and $q' = 1 - p'$. Note on this boundary $p \in [\frac{1}{1+e^{-\epsilon}}, 1]$ and $p' \in [\frac{1}{1+e^{\epsilon}}, 1]$, and hence, $q \in [0, \frac{1}{1+e^{\epsilon}}]$ and $q' \in [0, \frac{1}{1+e^{-\epsilon}}]$.

Plugging $q$ and $q'$ into the privacy cost objective $f(p, p'; \gamma_{max})$, one gets a new objective in $q, q'$ as

$$f(q, q'; \gamma_{max}) = \sum_{l=0}^{\frac{K-1}{2}} \left( e^{m\epsilon} \binom{K}{l} (1-q')^l q'^{K-l} - \binom{K}{l} (1-q)^l q^{K-l} \right) \cdot \gamma_{max}(l) \qquad (96)$$

$$+ \sum_{l=\frac{K+1}{2}}^{K} \left( \binom{K}{l} (1-q)^l q^{K-l} - e^{m\epsilon} \binom{K}{l} (1-q')^l q'^{K-l} \right) \cdot \gamma_{max}(l)$$

$$= \sum_{l=0}^{\frac{K-1}{2}} \left( e^{m\epsilon} \binom{K}{l} (1-q')^l q'^{K-l} - \binom{K}{l} (1-q)^l q^{K-l} \right) \qquad (97)$$

$$+ \sum_{l=\frac{K+1}{2}}^{K} \left( \binom{K}{l} (1-q)^l q^{K-l} - e^{m\epsilon} \binom{K}{l} (1-q')^l q'^{K-l} \right)$$

Since on this boundary, $1 - p' = e^{\epsilon}(1 - p)$, writing this in $q, q'$, this becomes $q' = e^{\epsilon} q$. Plugging $q' = e^{\epsilon} q$ into $f(q, q'; \gamma_{max})$, one gets

$$f(q; \gamma_{max}) = \sum_{l=0}^{\frac{K-1}{2}} \left( e^{m\epsilon} \binom{K}{l} (1-e^{\epsilon} q)^l (e^{\epsilon} q)^{K-l} - \binom{K}{l} (1-q)^l q^{K-l} \right) \qquad (98)$$

$$+ \sum_{l=\frac{K+1}{2}}^{K} \left( \binom{K}{l}(1-q)^l q^{K-l} - e^{m\epsilon}\binom{K}{l}(1-e^\epsilon q)^l (e^\epsilon q)^{K-l} \right)$$

The gradient w.r.t. $q$ is

$$\nabla_q f(q) = \sum_{l=0}^{\frac{K-1}{2}} \left( e^{m\epsilon}\binom{K}{l}\left((-e^\epsilon)l(1-e^\epsilon q)^{l-1}(e^\epsilon q)^{K-l} + e^\epsilon(K-l)(1-e^\epsilon q)^l(e^\epsilon q)^{K-l-1}\right)\right. \tag{99}$$
$$- \binom{K}{l}\left(-l(1-q)^{l-1}q^{K-l} + (K-l)(1-q)^l q^{K-l-1}\right)\Big)$$
$$+ \sum_{l=\frac{K+1}{2}}^{K}\left(\binom{K}{l}\left(-l(1-q)^{l-1}q^{K-l} + (K-l)(1-q)^l q^{K-l-1}\right)\right.$$
$$- e^{m\epsilon}\binom{K}{l}\left((-e^\epsilon)l(1-e^\epsilon q)^{l-1}(e^\epsilon q)^{K-l} + e^\epsilon(K-l)(1-e^\epsilon q)^l(e^\epsilon q)^{K-l-1}\right)\Big)$$
$$= -\sum_{l=1}^{\frac{K-1}{2}} e^{(m+1)\epsilon}\binom{K-1}{l-1}\frac{K}{l}l(1-e^\epsilon q)^{l-1}(e^\epsilon q)^{K-l} + \sum_{l=0}^{\frac{K-1}{2}} e^{(m+1)\epsilon}\binom{K-1}{l}\frac{K}{K-l}(K-l)(1-e^\epsilon q)^l(e^\epsilon q)^{K-l-1} \tag{100}$$
$$+ \sum_{l=1}^{\frac{K-1}{2}}\binom{K-1}{l-1}\frac{K}{l}l(1-q)^{l-1}q^{K-l} - \sum_{l=0}^{\frac{K-1}{2}}\binom{K-1}{l}\frac{K}{K-l}(K-l)(1-q)^l q^{K-l-1}$$
$$- \sum_{l=\frac{K+1}{2}}^{K}\binom{K-1}{l-1}\frac{K}{l}l(1-q)^{l-1}q^{K-l} + \sum_{l=\frac{K+1}{2}}^{K-1}\binom{K-1}{l}\frac{K}{K-l}(K-l)(1-q)^l q^{K-l-1}$$
$$+ \sum_{l=\frac{K+1}{2}}^{K} e^{(m+1)\epsilon}\binom{K-1}{l-1}\frac{K}{l}l(1-e^\epsilon q)^{l-1}(e^\epsilon q)^{K-l} - \sum_{l=\frac{K+1}{2}}^{K-1} e^{(m+1)\epsilon}\binom{K-1}{l}\frac{K}{K-l}(K-l)(1-e^\epsilon q)^l(e^\epsilon q)^{K-l-1}$$
$$= -K\sum_{l=1}^{\frac{K-1}{2}} e^{(m+1)\epsilon}\binom{K-1}{l-1}(1-e^\epsilon q)^{l-1}(e^\epsilon q)^{K-l} + K\sum_{l=0}^{\frac{K-1}{2}} e^{(m+1)\epsilon}\binom{K-1}{l}(1-e^\epsilon q)^l(e^\epsilon q)^{K-l-1} \tag{101}$$
$$+ K\sum_{l=1}^{\frac{K-1}{2}}\binom{K-1}{l-1}(1-q)^{l-1}q^{K-l} - K\sum_{l=0}^{\frac{K-1}{2}}\binom{K-1}{l}(1-q)^l q^{K-l-1}$$
$$- K\sum_{l=\frac{K+1}{2}}^{K}\binom{K-1}{l-1}(1-q)^{l-1}q^{K-l} + K\sum_{l=\frac{K+1}{2}}^{K-1}\binom{K-1}{l}(1-q)^l q^{K-l-1}$$
$$+ K\sum_{l=\frac{K+1}{2}}^{K} e^{(m+1)\epsilon}\binom{K-1}{l-1}(1-e^\epsilon q)^{l-1}(e^\epsilon q)^{K-l} - K\sum_{l=\frac{K+1}{2}}^{K-1} e^{(m+1)\epsilon}\binom{K-1}{l}(1-e^\epsilon q)^l(e^\epsilon q)^{K-l-1}$$
$$= 2Ke^{(m+1)\epsilon}\binom{K-1}{\frac{K-1}{2}}(1-e^\epsilon q)^{\frac{K-1}{2}}(e^\epsilon q)^{\frac{K-1}{2}} - 2K\binom{K-1}{\frac{K-1}{2}}(1-q)^{\frac{K-1}{2}}q^{\frac{K-1}{2}} \tag{102}$$

Recall $q \in [0, \frac{1}{1+e^\epsilon}]$ and so $(1-e^\epsilon q)(e^\epsilon q) \geq (1-q)q$. Furthermore, since $e^{(m+1)\epsilon} \geq 1$, there is $\nabla_q f(q) \geq 0$. This implies $f(q)$ is monotonically non-decreasing in $q$, and so the local maximum on this boundary is

$$(q^*_{local}, q'^*_{local}) = \underset{(q,q'):q'=e^\epsilon q, q\in[0,\frac{1}{1+e^\epsilon}]}{\arg\max} f(q, q'; \gamma_{max}) = \left(\frac{1}{1+e^\epsilon}, \frac{1}{1+e^{-\epsilon}}\right) \tag{103}$$

That is,

$$(p_{local}^*, p_{local}'^*) = \operatorname*{arg\,max}_{(p,p'):1-p'=e^\epsilon(1-p), p\in[\frac{1}{1+e^{-\epsilon}},1]} f(p,p';\gamma_{max}) = (1-q_{local}^*, 1-q_{local}'^*) = (\frac{1}{1+e^{-\epsilon}}, \frac{1}{1+e^\epsilon}) \tag{104}$$

**Part III: The global worst case probabilities.**

Notice that $(\frac{1}{1+e^{-\epsilon}}, \frac{1}{1+e^\epsilon})$, the maximum on the second boundary $1 - p' = e^\epsilon(1-p), \forall p \in [\frac{1}{1+e^{-\epsilon}}, 1]$, is indeed the minimum on the first boundary $p = e^\epsilon p', \forall p \in [0, \frac{1}{1+e^{-\epsilon}+1}]$.

Therefore, the global maximum given $\gamma_{max}$ is

$$(p^*, p'^*) = \operatorname*{arg\,max}_{(p,p')\in\mathcal{F}} f(p,p';\gamma_{max}) = \operatorname*{arg\,max}_{(p,p'):p=e^\epsilon p', p\in[0,\frac{1}{1+e^{-\epsilon}}]} f(p,p';\gamma_{max}) = (0,0) \tag{105}$$

and recall that $f(0,0;\gamma_{max}) = e^{m\epsilon} - 1$.

Hence, if $m \geq \frac{K+1}{2}$, by Lemma 3.4 $\mathsf{DaRRM}_{\gamma_{max}}$ is $m\epsilon$-differentially private.

$\square$

### B.2.2 Privacy Amplification Under A Small Privacy Allowance $m \leq \frac{K-1}{2}$

The proof of Lemma B.10 is slightly more involved. First, recall by Lemma 3.1, $\gamma_{Sub}$, the noise function that makes the output of $\mathsf{DaRRM}_{\gamma_{Sub}}$ and the subsampling baseline the same, is

$$\gamma_{Sub}(l) = \gamma_{Sub}(K=l)$$
$$= \begin{cases} 1 - 2\sum_{j=\frac{m+1}{2}}^m \frac{\binom{l}{j}\binom{K-l}{m-j}}{\binom{K}{m}} & \text{if } m \text{ is odd} \\ 1 - 2\sum_{j=\frac{m}{2}+1}^m \frac{\binom{l}{j}\binom{K-l}{m-j}}{\binom{K}{m}} - \frac{\binom{l}{\frac{m}{2}}\binom{K-l}{\frac{m}{2}}}{\binom{K}{m}} & \text{if } m \text{ is even} \end{cases}$$

for $l \in \{0, 1, \ldots, K\}$, suppose the privacy allowance $m \in \mathbb{Z}$.

If we define $h(l) := \begin{cases} \sum_{j=\frac{m+1}{2}}^m \frac{\binom{l}{j}\binom{K-l}{m-j}}{\binom{K}{m}} & \text{if } m \text{ is odd} \\ \sum_{j=\frac{m}{2}+1}^m \frac{\binom{l}{j}\binom{K-l}{m-j}}{\binom{K}{m}} - \frac{\binom{l}{\frac{m}{2}}\binom{K-l}{\frac{m}{2}}}{\binom{K}{m}} & \text{if } m \text{ is even} \end{cases}$, then $\gamma_{Sub}(l)$ can be written as $\gamma_{Sub}(l) = $

$\begin{cases} 1 - 2h(l) & \text{if } l \leq \frac{K-1}{2} \\ 2h(l) - 1 & \text{if } l \geq \frac{K+1}{2} \end{cases}$.

This can be generalized to a broader class of $\gamma$ functions — which we call the "symmetric form family" — as follows

**Definition B.6.** $\gamma : \{0, 1, \ldots, K\} \to [0, 1]$ *is a member of the "symmetric form family" if $\gamma$ follows*

$$\gamma(l) = \begin{cases} 1 - 2h(l) & \text{if } l \leq \frac{K-1}{2} \\ 2h(l) - 1 & \text{if } l \geq \frac{K+1}{2} \end{cases} \tag{106}$$

*where $h : \{0, 1, \ldots, K\} \to [0, 1]$ and*

$$h(l) + h(K-l) = 1, \quad h(l+1) \geq h(l), \quad \forall l \in \{0, 1, \ldots, K\}, \quad and \quad \gamma(\frac{K-1}{2}) > 0, \gamma(\frac{K+1}{2}) > 0$$

It is easy to verify any $\gamma$ function that belongs to the "symmetric form family" satisfies: 1) symmetric around $\frac{K}{2}$ and 2) the monotonicity assumption. Hence, Lemma B.1 can be invoked to find the worst case probabilities

given such $\gamma$, i.e., $(p^*, p'^*) = \arg\max_{(p,p')\in\mathcal{F}} f(p, p'; \gamma)$, which in turn gives us the guarantee of $\mathsf{DaRRM}_\gamma$ being $m\epsilon$-differentially private.

**Roadmap.** In this section, we restrict our search of a good $\gamma$ that maximizes the utility of $\mathsf{DaRRM}_\gamma$ to in the "symmetric form family". To show the main privacy amplification result under a small $m$ in Lemma B.10, Section B.2.4, we need a few building blocks, shown in Section B.2.3. We first show in Lemma B.7, Section B.2.3 two clean sufficient conditions that if a "symmetric form family" $\gamma$ satisfies, then $\mathsf{DaRRM}_\gamma$ is $m\epsilon$-differentially private, in terms of the expectation of the $\gamma$ function applied to Binomial random variables. The Binomial random variables appear in the lemma, because recall the sum of the observed outcomes on a dataset $\mathcal{D}$, $\mathcal{L}(\mathcal{D})$, follows a Binomial distribution in the i.i.d. mechanisms setting. Next, we show a recurrence relationship that connects the expectation of Binomial random variables to Hypergeometric random variables in Lemma B.9. This is needed because observe that for $\gamma$ functions that makes $\mathsf{DaRRM}_\gamma$ have the same output as the majority of subsampled mechanisms, the $h$ function is now a sum of pmfs of the Hypergeometric random variable.

Finally, the proof of the main result under a small $m$ (Lemma B.10) is presented in Section B.2.4, based on Lemma B.7 and Lemma B.9. We show in Lemma B.10 that $\gamma_{DSub}$, i.e., the $\gamma$ function that enables the output of $\mathsf{DaRRM}_{\gamma_{DSub}}$ and outputting the majority of $2m-1$ subsampled mechanisms to be the same, belongs to the "symmetric form family" and satisfies the sufficient conditions as stated in Lemma B.7, implying $\mathsf{DaRRM}_{\gamma_{DSub}}$ being $m\epsilon$-differentially private.

### B.2.3   Building Blocks

**Lemma B.7** (Privacy conditions of the "symmetric form family" functions)**.** *Let random variables $X \sim Binomial(K-1, p')$, $Y \sim Binomial(K-1, e^\epsilon p')$, $\hat{X} \sim Binomial(K-1, 1-e^\epsilon(1-p))$ and $\hat{Y} \sim Binomial(K-1, p)$. For a function $\gamma : \{0, 1, \dots, K\} \to [0,1]$ that belongs to the "symmetric form family" (Definition B.6), if $\gamma$ also satisfies both conditions as follows:*

$$e^{m\epsilon}\mathbb{E}_X[h(X+1) - h(X)] \geq e^\epsilon \mathbb{E}_Y[h(Y+1) - h(Y)], \quad \forall p' \in [0, \frac{1}{1+e^\epsilon}] \tag{107}$$

$$e^{(m+1)\epsilon}\mathbb{E}_{\hat{X}}[h(\hat{X}+1) - h(\hat{X})] \geq \mathbb{E}_{\hat{Y}}[h(\hat{Y}+1) - h(\hat{Y})], \quad \forall p \in [\frac{1}{1+e^{-\epsilon}}, 1] \tag{108}$$

*then Algorithm $\mathsf{DaRRM}_\gamma$ is $m\epsilon$-differentially private.*

*Proof of Lemma B.7.* Since $h(l+1) \geq h(l)$ on $l \in \{0, \dots, K\}$, $\gamma(l) \geq \gamma(l+1), \forall l \leq \frac{K}{2}$ and $\gamma(l+1) \geq \gamma(l), \forall l \geq \frac{K}{2}$. Furthermore, since $h(l) + h(K-l) = 1$, $\gamma(\frac{K-1}{2}) = 1 - 2h(\frac{K-1}{2}) = 1 - 2(1 - h(\frac{K+1}{2})) = 2h(\frac{K+1}{2}) - 1$. Hence, any $\gamma$ that belongs to the "symmetric form family" satisfies: 1) symmetric around $\frac{K}{2}$, 2) the monotonicity assumption, and 3) $\gamma(\frac{K-1}{2}) = \gamma(\frac{K+1}{2}) > 0$.

Therefore, by Lemma B.1, the worst case probabilities $(p^*, p'^*) = \arg\max_{(p,p')\in\mathcal{F}} f(p, p'; \gamma)$ are on one of the two boundaries of $\mathcal{F}$, satisfying

$$p^* = e^\epsilon p'^*, \qquad\qquad \forall p^* \in [0, \frac{1}{e^{-\epsilon}+1}], p'^* \in [0, \frac{1}{1+e^\epsilon}] \tag{109}$$

$$\text{or} \quad 1 - p'^* = e^\epsilon(1 - p^*), \qquad\qquad \forall p^* \in [\frac{1}{1+e^{-\epsilon}}, 1], p'^* \in [\frac{1}{1+e^\epsilon}, 1] \tag{110}$$

We now derive the sufficient conditions that if any $\gamma$ from the "symmetric form family" satisfy, then $\mathsf{DaRRM}_\gamma$ is $m\epsilon$-differentially private, from the two boundaries as in Eq. 109 and Eq. 110 separately.

**Part I: Deriving a sufficient condition from Eq. 109 for "symmetric form family" $\gamma$.**

Consider the boundary of $\mathcal{F}$, $p = e^\epsilon p'$, $\forall p \in [0, \frac{1}{1+e^{-\epsilon}}], p' \in [0, \frac{1}{1+e^\epsilon}]$.

Given any $\gamma$, plugging $p = e^\epsilon p'$ into the privacy cost objective $f(p, p'; \gamma)$, one gets

$$f(p'; \gamma) = \sum_{l=0}^{\frac{K-1}{2}} (e^{m\epsilon} \binom{K}{l} p'^l (1 - p')^{K-l} - \binom{K}{l} (e^\epsilon p')^l (1 - e^\epsilon p')^{K-l}) \cdot \gamma(l) \tag{111}$$

$$+ \sum_{l=\frac{K+1}{2}}^{K} (\binom{K}{l} (e^\epsilon p')^l (1 - e^\epsilon p')^{K-l} - e^{m\epsilon} \binom{K}{l} p'^l (1 - p')^{K-l}) \cdot \gamma(l)$$

The gradient w.r.t. $p'$ is

$$\frac{\nabla_{p'} f(p'); \gamma}{K} = e^{m\epsilon} \sum_{l=0}^{\frac{K-1}{2}-1} \binom{K-1}{l} p'^l (1 - p')^{K-l-1} \Big(\gamma(l+1) - \gamma(l)\Big) - 2e^{m\epsilon} \binom{K-1}{\frac{K-1}{2}} p'^{\frac{K-1}{2}} (1 - p')^{\frac{K-1}{2}} \gamma(\frac{K-1}{2})$$

$$\tag{112}$$

$$+ e^{m\epsilon} \sum_{l=\frac{K+1}{2}}^{K-1} \binom{K-1}{l} p'^l (1 - p')^{K-l-1} \Big(\gamma(l) - \gamma(l+1)\Big)$$

$$+ e^\epsilon \sum_{l=0}^{\frac{K-1}{2}-1} \binom{K-1}{l} (e^\epsilon p')^l (1 - e^\epsilon p')^{K-l-1} \Big(\gamma(l) - \gamma(l+1)\Big) + 2e^\epsilon \binom{K-1}{\frac{K-1}{2}} (e^\epsilon p')^{\frac{K-1}{2}} (1 - e^\epsilon p')^{\frac{K-1}{2}} \gamma(\frac{K-1}{2})$$

$$+ e^\epsilon \sum_{l=\frac{K+1}{2}}^{K-1} \binom{K-1}{l} (e^\epsilon p')^l (1 - e^\epsilon p')^{K-l-1} \Big(\gamma(l+1) - \gamma(l)\Big)$$

Consider $l \in \{0, 1, \ldots, K\}$ in the above Eq. 112. For any function $\gamma$ that belongs to the "symmetric form family",

1. If $l \leq \frac{K}{2}$, $\gamma(l) - \gamma(l+1) = (1 - 2h(l)) - (1 - 2h(l+1)) = 2h(l+1) - 2h(l)$

2. If $l \geq \frac{K}{2}$, $\gamma(l+1) - \gamma(l) = (2h(l+1) - 1) - (2h(l) - 1) = 2h(l+1) - 2h(l)$

3. Since $\gamma(\frac{K-1}{2}) = \gamma(\frac{K+1}{2})$,

$$2\gamma(\frac{K-1}{2}) = \Big(\gamma(\frac{K-1}{2}) + \gamma(\frac{K+1}{2})\Big) \tag{113}$$

$$= \Big(1 - 2h(\frac{K-1}{2}) + 2h(\frac{K+1}{2}) - 1\Big) \tag{114}$$

$$= 2h(\frac{K+1}{2}) - 2h(\frac{K-1}{2}) \tag{115}$$

Hence, following Eq. 112, the gradient, $\nabla_{p'} f(p'; \gamma)$, given a "symmetric form family" $\gamma$ can be written as

$$\frac{\nabla_{p'} f(p'; \gamma)}{K} = -e^{m\epsilon} \sum_{l=0}^{K-1} \binom{K-1}{l} p'^l (1 - p')^{K-l} \Big(2h(l+1) - 2h(l)\Big) \tag{116}$$

$$+ e^\epsilon \sum_{l=0}^{K-1} \binom{K-1}{l} (e^\epsilon p')^l (1 - e^\epsilon p')^{K-l-1} \Big(2h(l+1) - 2h(l)\Big) \tag{117}$$

$$= -2e^{m\epsilon} \mathbb{E}_X[h(X+1) - h(X)] + 2e^\epsilon \mathbb{E}_Y[h(Y+1) - h(Y)]$$

where $X \sim \text{Binomial}(K-1, p')$ and $Y \sim \text{Binomial}(K-1, e^\epsilon p')$. The above implies

$$\nabla_{p'} f(p'; \gamma) \leq 0 \iff e^\epsilon \mathbb{E}_Y[h(Y+1) - h(Y)] \leq e^{m\epsilon} \mathbb{E}_X[h(X+1) - h(X)] \tag{118}$$

If $\nabla_{p'} f(p'; \gamma) \leq 0$, then we know the local worst case probabilities on the boundary $p = e^\epsilon p', \forall p \in [0, \frac{1}{1+e^{-\epsilon}}]$ given any $\gamma$ is $(p^*_{local}, p'^*_{local}) = \arg\max_{(p,p'):p=e^\epsilon p', p \in [0, \frac{1}{1+e^{-\epsilon}}]} f(p, p'; \gamma) = (0, 0)$. Furthermore, recall the privacy cost objective given any $\gamma$ is

$$f(p, p'; \gamma)$$

$$= \sum_{l=0}^{\frac{K-1}{2}} (e^{m\epsilon} \alpha'_l - \alpha_l) \cdot \gamma(l) + \sum_{l=\frac{K+1}{2}}^{K} (\alpha_l - e^{m\epsilon} \alpha'_l) \cdot \gamma(l)$$

$$= \sum_{l=0}^{\frac{K-1}{2}} \left( e^{m\epsilon} \binom{K}{l} p'^l (1-p')^{K-l} - \binom{K}{l} p^l (1-p)^{K-l} \right) \cdot \gamma(l) + \sum_{l=\frac{K+1}{2}}^{K} \left( \binom{K}{l} p^l (1-p)^{K-l} - e^{m\epsilon} \binom{K}{l} p'^l (1-p')^{K-l} \right) \cdot \gamma(l)$$

and so for any $\gamma$,

$$f(0, 0; \gamma) = (e^{m\epsilon} - 1) \cdot \gamma(0) \leq e^{m\epsilon} - 1 \tag{119}$$

Also, notice the local minimum on this boundary is

$$(p_{min}, p'_{min}) = \underset{(p,p'):p=e^\epsilon p', p \in [0, \frac{1}{1+e^{-\epsilon}}]}{\arg\min} f(p, p'l; \gamma) = (\frac{1}{1+e^{-\epsilon}}, \frac{1}{1+e^\epsilon}) \tag{120}$$

**Part II: Deriving a sufficient condition from Eq. 110 for "symmetric form family" $\gamma$.**

Consider the boundary of $\mathcal{F}$, $1 - p' = e^\epsilon(1-p), \forall p \in [\frac{1}{1+e^{-\epsilon}}, 1], p' \in [\frac{1}{1+e^\epsilon}, 1]$. For simplicity, let $q = 1 - p \in [0, \frac{1}{1+e^\epsilon}]$ and $q' = 1 - p' \in [0, \frac{1}{1+e^{-\epsilon}}]$. Plugging $q' = e^\epsilon q$ into the privacy cost objective, one gets, given any $\gamma$,

$$f(q; \gamma) = \sum_{l=0}^{\frac{K-1}{2}} \left( e^{m\epsilon} \binom{K}{l} (1 - e^\epsilon q)^l (e^\epsilon q)^{K-l} - \binom{K}{l} (1-q)^l q^{K-l} \right) \cdot \gamma(l) \tag{121}$$

$$+ \sum_{l=\frac{K+1}{2}}^{K} \left( \binom{K}{l} (1-q)^l q^{K-l} - e^{m\epsilon} \binom{K}{l} (1 - e^\epsilon q)^l (e^\epsilon q)^{K-l} \right) \cdot \gamma(l)$$

The gradient w.r.t. $q$ is

$$\frac{\nabla_q f(q; \gamma)}{K} = \sum_{l=0}^{\frac{K-1}{2}-1} e^{(m+1)\epsilon} \binom{K-1}{l} (1 - e^\epsilon q)^l (e^\epsilon q)^{K-l-1} \cdot \left( \gamma(l) - \gamma(l+1) \right) \tag{122}$$

$$+ \sum_{l=\frac{K+1}{2}}^{K-1} \binom{K-1}{l} (1 - e^\epsilon q)^l (e^\epsilon q)^{K-l-1} \cdot \left( \gamma(l+1) - \gamma(l) \right) + 2e^{(m+1)\epsilon} \binom{K-1}{\frac{K-1}{2}} (1 - e^\epsilon q)^{\frac{K-1}{2}} (e^\epsilon q)^{\frac{K-1}{2}} \cdot \gamma(\frac{K-1}{2})$$

$$+ \sum_{l=0}^{\frac{K-1}{2}-1} \binom{K-1}{l} (1-q)^l q^{K-l-1} \cdot \left( \gamma(l+1) - \gamma(l) \right)$$

$$+ \sum_{l=\frac{K+1}{2}}^{K-1} (1-q)^l q^{K-l-1} \cdot \left( \gamma(l) - \gamma(l+1) \right) - 2 \binom{K-1}{\frac{K-1}{2}} (1-q)^{\frac{K-1}{2}} q^{\frac{K-1}{2}} \cdot \gamma(\frac{K-1}{2})$$

For any function $\gamma$ that belongs to the "symmetric form family", the gradient $\nabla_q f(q; \gamma)$ can be written as

$$\frac{\nabla_q f(q; \gamma)}{K} = e^{(m+1)\epsilon} \sum_{l=0}^{K-1} \binom{K-1}{l} (1 - e^\epsilon q)^l (e^\epsilon q)^{K-l-1} \cdot \left( 2h(l+1) - 2h(l) \right) \tag{123}$$

$$-\sum_{l=0}^{K}\binom{K-1}{l}(1-q)^l q^{K-l-1}\cdot\Big(2h(l+1)-2h(l)\Big)$$

$$=2e^{(m+1)\epsilon}\mathbb{E}_{\hat{X}}[h(\hat{X}+1)-h(\hat{X})]-2\mathbb{E}_{\hat{Y}}[h(\hat{Y}+1)-h(\hat{Y})] \tag{124}$$

where $\hat{X}\sim\text{Binomial}(K-1,1-e^\epsilon(1-p))$ and $\hat{Y}\sim\text{Binomial}(K-1,p)$. The above implies

$$\nabla_q f(q;\gamma)\geq 0 \iff e^{(m+1)\epsilon}\mathbb{E}_{\hat{X}}[h(\hat{X}+1)-h(\hat{X})]\geq\mathbb{E}_{\hat{Y}}[h(\hat{Y}+1)-h(\hat{Y})] \tag{125}$$

If $\nabla_q f(q;\gamma)\geq 0$, then since $q\in[0,\frac{1}{1+e^\epsilon}]$, we know that the local maximum given any $\gamma$ is $(q^*_{local},q'^*_{local})=\arg\max_{(q,q'):q'=e^\epsilon q,q\in[0,\frac{1}{1+e^\epsilon}]}f(q,q';\gamma)=(\frac{1}{1+e^\epsilon},\frac{1}{1+e^{-\epsilon}})$. That is,

$$(p^*_{local},p'^*_{local})=\underset{(p,p'):1-p'=e^\epsilon(1-p),p\in[\frac{1}{1+e^{-\epsilon}},1]}{\arg\max}f(p,p';\gamma)=(1-q^*_{local},1-q'^*_{local})=(\frac{1}{1+e^{-\epsilon}},\frac{1}{1+e^\epsilon})$$

Notice by Eq. 120, the above $(\frac{1}{1+e^{-\epsilon}},\frac{1}{1+e^\epsilon})$ is the local minimum on the first boundary $p=e^\epsilon p',\forall p\in[0,\frac{1}{1+e^{-\epsilon}}]$. Therefore, given an arbitrary $\gamma$ function, if it satisfies both of the following:

1. On the boundary $p=e^\epsilon p',\forall p\in[0,\frac{1}{1+e^{-\epsilon}}]$, $\nabla_{p'}f(p';\gamma)\leq 0$

2. On the boundary $1-p'=e^\epsilon(1-p),\forall p\in[\frac{1}{1+e^{-\epsilon}},1]$, $\nabla_{q'}f(q';\gamma)\geq 0$ where $q'=1-p'$

then the global worst case probabilities given this $\gamma$ is $(p^*,p'^*)=\arg\max_{(p,p')\in\mathcal{F}}f(p,p';\gamma)=(0,0)$. Furthermore, since by Eq. 119, $f(0,0;\gamma)\leq e^{m\epsilon}-1$ for any $\gamma$, this implies $\mathsf{DaRRM}_\gamma$ is $m\epsilon$-differentially private by Lemma 3.4.

Now, if $\gamma$ belongs to the "symmetric form family", by Eq. 118 and Eq. 125, the sufficient conditions for $\gamma$ that enables $\mathsf{DaRRM}_\gamma$ to be $m\epsilon$-differentially private are hence

$$e^\epsilon\mathbb{E}_Y[h(Y+1)-h(Y)]\leq e^{m\epsilon}\mathbb{E}_X[h(X+1)-h(X)],\quad\forall p'\in[0,\frac{1}{1+e^\epsilon}]$$

$$\text{and}\quad e^{(m+1)\epsilon}\mathbb{E}_{\hat{X}}[h(\hat{X}+1)-h(\hat{X})]\geq\mathbb{E}_{\hat{Y}}[h(\hat{Y}+1)-h(\hat{Y})],\quad\forall p\in[\frac{1}{1+e^{-\epsilon}},1]$$

where $X\sim\text{Binomial}(K-1,p')$, $Y\sim\text{Binomial}(K-1,e^\epsilon p')$, $\hat{X}\sim\text{Binomial}(K-1,1-e^\epsilon(1-p))$ and $\hat{Y}\sim\text{Binomial}(K-1,p)$.

$\qquad\square$

**Lemma B.8** (Binomial Expectation Recurrence Relationship (Theorem 2.1 of Zhang et al. (2019))). *Let $X_{(K-1)}\sim Binomial(K-1,p)$ and $X_{(K)}\sim Binomial(K,p)$. Let $g(x)$ be a function with $-\infty<\mathbb{E}[g(X_{(K-1)})]<\infty$ and $-\infty<g(-1)<\infty$, then*

$$Kp\mathbb{E}_{X_{(K-1)}}[g(X_{(K-1)})]=\mathbb{E}_{X_{(K)}}[X_{(K)}g(X_{(K)}-1)] \tag{126}$$

**Lemma B.9.** *Given $i,m,K\in\mathbb{Z}$, $K\geq 1$, $0\leq i\leq m\leq K$, let $X_{(K)}\sim Binomial(K,p)$ for some $p\in[0,1]$, there is*

$$\frac{1}{\binom{K}{m}}\mathbb{E}_{X_{(K)}}\left[\binom{X}{i}\binom{K-X}{m-i}\right]=\binom{m}{i}p^i(1-p)^{m-i} \tag{127}$$

*Proof of Lemma B.9.* We show the above statement in Eq. 127 by induction on $K$ and $m$.

Base Case: $K=1$.

1. If $m=0$, then $i=0$. $\frac{1}{\binom{1}{0}}\mathbb{E}_{X_{(1)}}[\binom{X}{0}\binom{1-X}{0}]=\mathbb{E}_{X_{(1)}}[1]=1$, and $\binom{0}{0}p^0(1-p)^0=1$.

2. If $m = 1$,

    (a) $i = 0$, $\frac{1}{\binom{1}{1}}\mathbb{E}_{X_{(1)}}[\binom{X}{0}\binom{1-X}{1}] = \mathbb{E}_{X_{(1)}}[1 - X] = 1 - p$, and $\binom{1}{0}p^0(1-p)^1 = 1 - p$

    (b) $i = 1$, $\frac{1}{\binom{1}{1}}\mathbb{E}_{X_{(1)}}[\binom{X}{1}\binom{1-X}{0}] = \mathbb{E}_{X_{(1)}}[X] = p$, and $\binom{1}{1}p^1(1-p)^0 = p$.

Hence, Eq. 127 holds for the base case.

Induction Hypothesis: Suppose the statement holds for some $K \geq 1$ and $0 \leq i \leq m \leq K$. Consider $1 \leq i \leq m \leq K + 1$,

$$\frac{1}{\binom{K+1}{m}}\mathbb{E}_{X_{(K+1)}}\left[\binom{X}{i}\binom{K+1-X}{m-i}\right] \tag{128}$$

$$= \frac{1}{\binom{K+1}{m}}\mathbb{E}_{X_{(K+1)}}\left[\frac{X!}{i!(X-i)!}\frac{(K+1-X)!}{(m-i)!(K+1-X-(m-i))!}\right] \tag{129}$$

$$= \frac{1}{\binom{K+1}{m}i!(m-i)!}\mathbb{E}_{X_{(K+1)}}\left[X\frac{(X-1)!}{((X-1)-(i-1))!}\frac{(K-(X-1))!}{(K-(X-1)-((m-1)-(i-1)))!}\right] \tag{130}$$

$$= \frac{1}{\binom{K+1}{m}i!(m-i)!}\mathbb{E}_{X_{(K)}}\left[\frac{X!}{(X-(i-1))!}\frac{(K-X)!}{(K-X-((m-1)-(i-1)))!}\right] \tag{131}$$

(By Lemma B.8)

$$= \frac{(i-1)!(m-i)!}{\binom{K+1}{m}i!(m-i)!}\mathbb{E}_{X_{(K)}}\left[\binom{X}{i-1}\binom{K-X}{(m-1)-(i-1)}\right] \tag{132}$$

$$= \frac{(i-1)!}{\binom{K+1}{m}i!}(K+1)p\binom{K}{m-1}\binom{m-1}{i-1}p^{i-1}(1-p)^{m-i} \tag{133}$$

(By Induction Hypothesis)

$$= \frac{m!(K+1-m)!}{(K+1)!i}\frac{K!}{(m-1)!(K-m+1)!}\frac{(m-1)!}{(i-1)!(m-i)!}(K+1)p^i(1-p)^{m-i} \tag{134}$$

$$= \frac{m!}{i!(m-i)!}p^i(1-p)^{m-i} = \binom{m}{i}p^i(1-p)^{m-i} \tag{135}$$

Now we consider the edge cases when $0 = i \leq m$.

If $i = 0$ and $m = 0$,

$$\frac{1}{\binom{K+1}{0}}\mathbb{E}_{X_{(K+1)}}[\binom{X}{0}\binom{K+1-X}{0}] = 1 \cdot \mathbb{E}_{X_{(K+1)}}[1] = 1 = \binom{0}{0}p^0(1-p)^0 \tag{136}$$

If $i = 0$ and $m > 0$,

$$\frac{1}{\binom{K+1}{m}}\mathbb{E}_{X_{(K+1)}}[\binom{K+1-X}{m}] \tag{137}$$

$$= \frac{1}{\binom{K+1}{m}}\sum_{x=0}^{K+1}\binom{K+1-x}{m}\binom{K+1}{x}p^x(1-p)^{K+1-x} \tag{138}$$

$$= \frac{1}{\binom{K+1}{m}}\sum_{x=0}^{K+1}\binom{K+1-x}{m}\left(\binom{K}{x}+\binom{K}{x-1}\mathbb{I}\{x\geq 1\}\right)p^x(1-p)^{K+1-x} \tag{139}$$

$$= \frac{1}{\binom{K+1}{m}}\sum_{x=0}^{K}\binom{K+1-x}{m}\binom{K}{x}p^x(1-p)^{K+1-x} + \frac{1}{\binom{K+1}{m}}\sum_{x=1}^{K+1}\binom{K+1-x}{m}\binom{K}{x-1}p^x(1-p)^{K+1-x} \tag{140}$$

(Since when $x = K + 1$ and $m > 0$, $\binom{K+1-x}{m} = 0$)

$$= \frac{1}{\binom{K+1}{m}} \Big( \sum_{x=0}^{K} \binom{K-x}{m} \binom{K}{x} p^x (1-p)^{K+1-x} + \sum_{x=0}^{K} \binom{K-x}{m-1} \binom{K}{x} p^x (1-p)^{K+1-x} \Big) \tag{141}$$

$$+ \frac{1}{\binom{K+1}{m}} \sum_{x=0}^{K} \binom{K-x}{m} \binom{K}{x} p^{x+1} (1-p)^{K-x}$$

$$\text{(Since } \binom{K+1-x}{m} = \binom{K-x}{m} + \binom{K-x}{m-1} )$$

$$= \frac{1}{\binom{K+1}{m}} \Big( (1-p) \mathbb{E}_{X_{(K)}} [\binom{K-X}{m}] + (1-p) \mathbb{E}_{X_{(k)}} [\binom{K-X}{m-1}] \Big) + \frac{1}{\binom{K+1}{m}} p \mathbb{E}_{X_{(K)}} [\binom{K-X}{m}] \tag{142}$$

$$= \frac{1}{\binom{K+1}{m}} \Big( \mathbb{E}_{X_{(K)}} [\binom{K-X}{m}] + (1-p) \mathbb{E}_{X_{(K)}} [\binom{K-X}{m-1}] \Big) \tag{143}$$

$$= \frac{1}{\binom{K+1}{m}} \Big( \binom{K}{m} (1-p)^m + (1-p) \binom{K}{m-1} (1-p)^{m-1} \Big) \tag{144}$$

$$\text{(By Induction Hypothesis)} \tag{145}$$

$$= \frac{1}{\binom{K+1}{m}} \binom{K+1}{m} (1-p)^m \tag{146}$$

$$= (1-p)^m \tag{147}$$

Hence, Eq. 127 holds for all $K \geq 1$ and $0 \leq i \leq m \leq K$.

$\square$

### B.2.4 Main Result: Privacy Amplification Under a Small $m$

**Lemma B.10** (Privacy amplification, $m \leq \frac{K-1}{2}$). *Consider using DaRRM (Algorithm 1) to solve Problem 1.1, with i.i.d. mechanisms $\{M_i\}_{i=1}^{K}$, $p_i = p$, $p_i' = p'$, $\forall i \in [K]$, the privacy allowance $1 \leq m \leq \frac{K-1}{2}, m \in \mathbb{Z}$ and $\delta = \Delta = 0$. Let the noise function be that*

$$\gamma_{DSub}(l) = \begin{cases} 1 - 2h(l) & \forall l \in \{0, 1, \ldots, \frac{K-1}{2}\} \\ 2h(l) - 1 & \forall l \in \{\frac{K+1}{2}, \ldots, K\} \end{cases} \tag{148}$$

*where $h : \{0, 1, \ldots, K\} \to [0, 1]$ and $h(l) = \sum_{i=m}^{2m-1} \frac{\binom{l}{i} \binom{K-l}{2m-1-i}}{\binom{K}{2m-1}}$, $\forall l \in \{0, 1, \ldots, K\}$, then Algorithm DaRRM$_{\gamma_{DSub}}$ is $m\epsilon$-differentially private.*

*Proof of Lemma B.10.* First, note $\gamma_{DSub}$ belongs to the "symmetric form family". We show $\gamma_{DSub}$ satisfies the two sufficient conditions in Lemma B.7 and hence by Lemma B.7, DaRRM$_{\gamma_{DSub}}$ is $m\epsilon$-differentially private. Specifically, we consider $h(l) = \sum_{i=m}^{2m-1} \frac{\binom{l}{i} \binom{K-l}{2m-1-i}}{\binom{K}{2m-1}}$, $\forall l \in \{0, 1, \ldots, K\}$ and $1 \leq m \leq K$.

Two show the first condition is satisfied, let $X_{(K-1)} \sim \text{Binomial}(K-1, p)$ and $Y_{(K-1)} \sim \text{Binomial}(K-1, e^\epsilon p)$, and consider $p \in [0, \frac{1}{1+e^\epsilon}]$.

$$\mathbb{E}_{X_{(K-1)}}[h(X+1)] = \frac{1}{\binom{K}{2m-1}} \sum_{i=m}^{2m-1} \mathbb{E}_{X_{(K-1)}} [\binom{X+1}{i} \binom{K-X-1}{2m-1-i}] \tag{149}$$

$$= \frac{1}{\binom{K}{2m-1}} \sum_{i=m}^{2m-1} \mathbb{E}_{X_{(K-1)}} [\binom{X}{i} \binom{K-X-1}{2m-1-i} + \binom{X}{i-1} \binom{K-X-1}{2m-1-i}] \tag{150}$$

$$\text{(Since } \binom{X+1}{i} = \binom{X}{i} + \binom{X}{i-1} \mathbb{I}\{i \geq 1\})$$

$$= \frac{1}{\binom{K}{2m-1}} \sum_{i=m}^{2m-1} \left( \mathbb{E}_{X_{(K-1)}}\left[ \binom{X}{i}\binom{K-1-X}{2m-1-i} \right] + \mathbb{E}_{X_{(K-1)}}\left[ \binom{X}{i-1}\binom{K-1-X}{(2m-2)-(i-1)} \right] \right) \tag{151}$$

$$= \frac{1}{\binom{K}{2m-1}} \sum_{i=m}^{2m-1} \left( \binom{K-1}{2m-1}\binom{2m-1}{i}p^i(1-p)^{2m-1-i} + \binom{K-1}{2m-2}\binom{2m-2}{i-1}p^{i-1}(1-p)^{2m-1-i} \right) \tag{152}$$

(By Lemma B.9)

$$\mathbb{E}_{X_{(K-1)}}[h(X)] = \frac{1}{\binom{K}{2m-1}} \sum_{i=m}^{2m-1} \mathbb{E}_{X_{(K-1)}}\left[ \binom{X}{i}\binom{K-X}{2m-1-i} \right] \tag{153}$$

$$\left(\text{Since } \binom{K-X}{2m-1-i} = \binom{K-1-X}{2m-1-i} + \binom{K-1-X}{2m-2-i}\right)$$

$$= \frac{1}{\binom{K}{2m-1}} \sum_{i=m}^{2m-1} \left( \mathbb{E}_{X_{(K-1)}}\left[ \binom{X}{i}\binom{K-1-X}{2m-1-i} \right] + \mathbb{E}_{X_{(K-1)}}\left[ \binom{X}{i}\binom{K-1-X}{2m-2-i} \right]\mathbb{I}\{i \le 2m-2\} \right) \tag{154}$$

$$= \frac{1}{\binom{K}{2m-1}} \sum_{i=m}^{2m-1} \left( \binom{K-1}{2m-1}\binom{2m-1}{i}p^i(1-p)^{2m-1-i} + \binom{K-1}{2m-2}\binom{2m-2}{i}p^i(1-p)^{2m-2-i}\mathbb{I}\{i \le 2m-2\} \right) \tag{155}$$

(By Lemma B.9)

Hence, following Eq. 155 and Eq. 152,

$$\mathbb{E}_{X_{(K-1)}}[h(X+1) - h(X)] \tag{156}$$

$$= \frac{1}{\binom{K}{2m-1}} \left( \sum_{i=m}^{2m-1} \binom{K-1}{2m-2}\binom{2m-2}{i-1}p^{i-1}(1-p)^{2m-1-i} - \sum_{i=m}^{2m-2} \binom{K-1}{2m-2}\binom{2m-2}{i}p^i(1-p)^{2m-2-i} \right) \tag{157}$$

$$= \frac{1}{\binom{K}{2m-1}} \left( \sum_{i=m-1}^{2m-2} \binom{K-1}{2m-2}\binom{2m-2}{i}p^i(1-p)^{2m-2-i} - \sum_{i=m}^{2m-2} \binom{K-1}{2m-2}\binom{2m-2}{i}p^i(1-p)^{2m-2-i} \right) \tag{158}$$

$$= \frac{2m-1}{K} \binom{2m-2}{m-1}p^{m-1}(1-p)^{m-1} \tag{159}$$

Similarly,

$$\mathbb{E}_{Y_{(K-1)}}[h(Y+1) - h(Y)] = \frac{2m-1}{K} \binom{2m-2}{m-1}(e^\epsilon p)^{m-1}(1 - e^\epsilon p)^{m-1} \tag{160}$$

Since $p \in [0, \frac{1}{1+e^\epsilon}]$, there is $p(1-p) \ge e^{-\epsilon}e^\epsilon p(1 - e^\epsilon p)$. Hence,

$$e^{(m-1)\epsilon}\mathbb{E}_{X_{(K-1)}}[h(X+1) - h(X)] = \frac{2m-1}{K} \binom{2m-2}{m-1}e^{(m-1)\epsilon}p^{m-1}(1-p)^{m-1} \tag{161}$$

$$\ge \frac{2m-1}{K} \binom{2m-2}{m-1}e^{(m-1)\epsilon}(e^{-\epsilon}e^\epsilon p(1 - e^\epsilon p))^{m-1} \tag{162}$$

$$= \frac{2m-1}{K} \binom{2m-2}{m-1}(e^\epsilon p)^{m-1}(1 - e^\epsilon p)^{m-1} \tag{163}$$

$$= \mathbb{E}_{Y_{(K-1)}}[h(Y+1) - h(Y)] \tag{164}$$

implying

$$e^{m\epsilon}\mathbb{E}_{X_{(K-1)}}[h(X+1) - h(X)] \geq e^{\epsilon}\mathbb{E}_{Y_{(K-1)}}[h(Y+1) - h(Y)] \tag{165}$$

and the first condition is satisfied.

To show the second condition is satisfied, let $\hat{X}_{(K-1)} \sim \text{Binom}(K-1, 1-e^{\epsilon}(1-p))$ and $\hat{Y}_{(K-1)} \sim \text{Binom}(K-1, p)$, and consider $p \in [\frac{1}{1+e^{-\epsilon}}, 1)$.

$$\mathbb{E}_{\hat{X}_{(K-1)}}[h(\hat{X}+1)] = \frac{1}{\binom{K}{2m-1}} \sum_{i=m}^{2m-1} \left( \mathbb{E}_{\hat{X}_{(K-1)}}[\binom{\hat{X}}{i}\binom{K-1-\hat{X}}{2m-1-i}] + \mathbb{E}_{\hat{X}_{(K-1)}}[\binom{\hat{X}}{i-1}\binom{K-1-\hat{X}}{(2m-2)-(i-1)}] \right) \tag{166}$$

$$= \frac{1}{\binom{K}{2m-1}} \sum_{i=m}^{2m-1} \left( \binom{K-1}{2m-1}\binom{2m-1}{i}(1-e^{\epsilon}(1-p))^i(e^{\epsilon}(1-p))^{2m-1-i} \right. \tag{167}$$
$$\left. + \binom{K-1}{2m-2}\binom{2m-2}{i-1}(1-e^{\epsilon}(1-p))^{i-1}(e^{\epsilon}(1-p))^{2m-1-i} \right)$$
By Lemma B.9

and

$$\mathbb{E}_{\hat{X}_{(K-1)}}[h(\hat{X})] = \frac{1}{\binom{K}{2m-1}} \sum_{i=m}^{2m-1} \left( \mathbb{E}_{\hat{X}_{(K-1)}}[\binom{\hat{X}}{i}\binom{K-1-\hat{X}}{2m-1-i}] + \mathbb{E}_{\hat{X}_{(K-1)}}[\binom{\hat{X}}{i}\binom{K-1-\hat{X}}{2m-2-i}]\mathbb{I}\{i \leq 2m-2\} \right) \tag{168}$$

$$= \frac{1}{\binom{K}{2m-1}} \sum_{i=m}^{2m-1} \left( \binom{K-1}{2m-1}\binom{2m-1}{i}(1-e^{\epsilon}(1-p))^i(e^{\epsilon}(1-p))^{2m-1-i} \right. \tag{169}$$
$$\left. + \binom{K-1}{2m-2}\binom{2m-2}{i}(1-e^{\epsilon}(1-p))^i(e^{\epsilon}(1-p))^{2m-2-i}\mathbb{I}\{i \leq 2m-2\} \right)$$
By Lemma B.9

Hence, following Eq. 167 and Eq. 169,

$$\mathbb{E}_{\hat{X}_{(K-1)}}[h(\hat{X}+1) - h(\hat{X})] \tag{170}$$

$$= \frac{1}{\binom{K}{2m-1}} \left( \sum_{i=m}^{2m-1} \binom{K-1}{2m-2}\binom{2m-2}{i-1}(1-e^{\epsilon}(1-p))^{i-1}(e^{\epsilon}(1-p))^{2m-1-i} \right. \tag{171}$$
$$\left. - \sum_{i=m}^{2m-2} \binom{K-1}{2m-2}\binom{2m-2}{i}(1-e^{\epsilon}(1-p))^i(e^{\epsilon}(1-p))^{2m-2-i} \right)$$

$$= \frac{1}{\binom{K}{2m-1}} \left( \sum_{i=m-1}^{2m-2} \binom{K-1}{2m-2}\binom{2m-2}{i}(1-e^{\epsilon}(1-p))^i(e^{\epsilon}(1-p))^{2m-2-i} \right. \tag{172}$$
$$\left. - \sum_{i=m}^{2m-2} \binom{K-1}{2m-2}\binom{2m-2}{i}(1-e^{\epsilon}(1-p))^i(e^{\epsilon}(1-p))^{2m-2-i} \right)$$

$$= \frac{2m-1}{K}\binom{2m-2}{m-1}(1-e^{\epsilon}(1-p))^{m-1}(e^{\epsilon}(1-p))^{m-1} \tag{173}$$

Similarly,

$$\mathbb{E}_{\hat{Y}_{(K-1)}}[h(\hat{Y}+1) - h(\hat{Y})] = \frac{2m-1}{K}\binom{2m-2}{m-1}p^{m-1}(1-p)^{m-1} \tag{174}$$

Hence,

$$e^{(m+1)\epsilon}\mathbb{E}_{\hat{X}_{(K-1)}}[h(\hat{X}+1)-h(\hat{X})] = e^{(m+1)\epsilon}\frac{2m-1}{K}\binom{2m-2}{m-1}(1-e^\epsilon(1-p))^{m-1}(e^\epsilon(1-p))^{m-1} \quad (175)$$

$$\geq \frac{2m-1}{K}\binom{2m-2}{m-1}(1-e^\epsilon(1-p))^{m-1}e^{(m-1)\epsilon}(1-p)^{m-1} \quad (176)$$

$$= \frac{2m-1}{K}\binom{2m-2}{m-1}(e^\epsilon - e^{2\epsilon}(1-p))^{m-1}(1-p)^{m-1} \quad (177)$$

Note that

$$e^\epsilon - e^{2\epsilon}(1-p) = e^\epsilon - e^{2\epsilon} + e^{2\epsilon}p \geq p \quad (178)$$

$$\iff (e^\epsilon+1)(e^\epsilon-1)p \geq e^\epsilon(e^\epsilon-1) \quad (179)$$

$$\iff p \geq \frac{e^\epsilon}{e^\epsilon+1} = \frac{1}{1+e^{-\epsilon}} \quad (180)$$

and the condition needs to hold for $p \in [\frac{1}{1+e^{-\epsilon}}, 1]$.

Therefore, following Eq. 177,

$$e^{(m+1)\epsilon}\mathbb{E}_{\hat{X}_{(K-1)}}[h(\hat{X}+1)-h(\hat{X})] \geq \frac{2m-1}{K}\binom{2m-2}{m-1}p^{m-1}(1-p)^{m-1} \quad (181)$$

$$= \mathbb{E}_{\hat{Y}_{(K-1)}}[h(\hat{Y}+1)-h(\hat{Y})] \quad (182)$$

implying the second condition is satisfied.

Therefore, by Lemma B.7, $\mathsf{DaRRM}_{\gamma_{DSub}}$ is $m\epsilon$-differentially private.

$\square$

## B.3 Comparing the Utility of Subsampling Approaches

Intuitively, if we subsample $2m-1$ mechanisms, the utility is higher than that of the naïve subsampling approach which outputs the majority based on only $m$ mechanisms. To complete the story, we formally compare the utility of outputting the majority of $2m-1$ subsampled mechanisms (Theorem 4.1) and outputting the majority of $m$ subsampled mechanisms (simple composition, Theorem 2.2) in the i.i.d. mechanisms and pure differential privacy setting, fixing the output privacy loss to be $m\epsilon$.

**Lemma B.11.** *Consider Problem 1.1 with i.i.d. mechanisms $\{M_i\}_{i=1}^K$, i.e., $p = p_i = \Pr[M_i(\mathcal{D}) = 1], p' = p_i' = \Pr[M_i(\mathcal{D}') = 1], \forall i \in [K]$. Let $\gamma_1 : \{0, 1, \dots, K\} \to [0, 1], \gamma_2 : \{0, 1, \dots, K\} \to [0, 1]$ be two functions that are both symmetric around $\frac{K}{2}$. If $1 \geq \gamma_1(l) \geq \gamma_2(l) \geq 0, \forall l \in \{0, \dots, K\}$, then $\mathcal{E}(\mathsf{DaRRM}_{\gamma_1}) \leq \mathcal{E}(\mathsf{DaRRM}_{\gamma_2})$.*

*Proof.* Recall $\mathcal{S} = \{S_1, \dots, S_K\}$, where $S_i \sim M_i(\mathcal{D})$, is the set of observed outcomes from the mechanisms $\{M_i\}_{i=1}^K$. By Definition 2.4, for any $\gamma$ that is symmetric around $\frac{K}{2}$, the error of $\mathsf{DaRRM}_\gamma$ is

$$\mathcal{E}(\mathsf{DaRRM}_\gamma) = \left| \Pr[\mathsf{DaRRM}_\gamma(\mathcal{D}) = 1] - \Pr[g(\mathcal{S}) = 1] \right| \quad (183)$$

$$= \left| \sum_{l=\frac{K+1}{2}}^K \left(\gamma(l) + \frac{1}{2}(1-\gamma(l))\right) \cdot \alpha_l + \sum_{l=0}^{\frac{K-1}{2}} \frac{1}{2}(1-\gamma(l)) \cdot \alpha_l - \sum_{l=\frac{K+1}{2}}^K \alpha_l \right| \quad (184)$$

$$= \left| \sum_{l=\frac{K+1}{2}}^K \left(\frac{1}{2}\gamma(l) - \frac{1}{2}\right) \cdot \alpha_l + \sum_{l=0}^{\frac{K-1}{2}} \left(\frac{1}{2} - \frac{1}{2}\gamma(l)\right) \cdot \alpha_l \right| \quad (185)$$

$$= \left| \frac{1}{2} \sum_{l=\frac{K+1}{2}}^K (1-\gamma(l)) \cdot (\alpha_l - \alpha_{K-l}) \right| \quad (186)$$

where $\alpha_l = \binom{K}{l} p^l (1-p)^{K-l}$, $\forall l \in \{0, 1, \ldots, K\}$ and recall $p = \Pr[M_i(\mathcal{D}) = 1]$, $\forall i \in [K]$.

For any $l \geq \frac{K+1}{2}$,

1. If $p = 0$ or $p = 1$, $\alpha_l = \alpha_{K-l}$.

2. Otherwise, for $p \in (0, 1)$,

    (a) If $p \geq \frac{1}{2}$,

$$\frac{\alpha_l}{\alpha_{K-l}} = \frac{p^l (1-p)^{K-l}}{p^{K-l}(1-p)^l} = p^{2l-K}(1-p)^{K-2l} = (\underbrace{\frac{p}{1-p}}_{\geq 1})^{\overbrace{2l-K}^{\geq 0}} \geq 1, \quad \Rightarrow \alpha_l \geq \alpha_{K-l} \qquad (187)$$

    (b) If $p < \frac{1}{2}$,

$$\frac{\alpha_l}{\alpha_{K-l}} = (\underbrace{\frac{p}{1-p}}_{\leq 1})^{\overbrace{2l-K}^{\geq 0}} \leq 1, \quad \Rightarrow \alpha_l \leq \alpha_{K-l} \qquad (188)$$

Hence, if $p \geq \frac{1}{2}$, then $\alpha_l \geq \alpha_{K-l}, \forall l \geq \frac{K+1}{2}$. Since $\gamma_1(l) \geq \gamma_2(l), \forall l \in \{0, \ldots, K\}$, $1 - \gamma_1(l) \leq 1 - \gamma_2(l)$, and so

$$\mathcal{E}(\mathsf{DaRRM}_{\gamma_1}) = \sum_{l=\frac{K+1}{2}}^{K} \frac{1}{2}(1 - \gamma_1(l)) \cdot (\alpha_l - \alpha_{K-l}) \leq \sum_{l=\frac{K+1}{2}}^{K} \frac{1}{2}(1 - \gamma_2(l)) \cdot (\alpha_l - \alpha_{K-l}) = \mathcal{E}(\mathsf{DaRRM}_{\gamma_2}) \quad (189)$$

Similarly, if $p < \frac{1}{2}$, then $\alpha_l \leq \alpha_{K-l}, \forall l \geq \frac{K+1}{2}$ and

$$\mathcal{E}(\mathsf{DaRRM}_{\gamma_1}) = \sum_{l=\frac{K+1}{2}}^{K} \frac{1}{2}(1 - \gamma_1(l)) \cdot (\alpha_{K-l} - \alpha_l) \leq \sum_{l=\frac{K+1}{2}}^{K} \frac{1}{2}(1 - \gamma_2(l)) \cdot (\alpha_{K-l} - \alpha_l) = \mathcal{E}(\mathsf{DaRRM}_{\gamma_2})) \quad (190)$$

Therefore,

$$\mathcal{E}(\mathsf{DaRRM}_{\gamma_1}) \leq \mathcal{E}(\mathsf{DaRRM}_{\gamma_2}) \qquad (191)$$

$\square$

Since $\gamma_{DSub}(l) \geq \gamma_{Sub}(l)$, $\forall l \in \{0, 1, \ldots, K\}$, by Lemma B.11, $\mathcal{E}(\mathsf{DaRRM}_{\gamma_{DSub}}) \leq \mathcal{E}(\mathsf{DaRRM}_{\gamma_{Sub}})$ — that is, outputting $2m - 1$ mechanisms has a higher utility than outputting $m$ mechanisms.

# C  Details of Section 5: Optimizing the Noise Function $\gamma$ in DaRRM

## C.1  Deriving the Optimization Objective

For any $\gamma$ function that is symmetric around $\frac{K}{2}$, we can write the optimization objective as

$$\mathbb{E}_{p_1,p_2,\dots,p_K \sim \mathcal{T}}[\mathcal{E}(\mathsf{DaRRM}_\gamma)] \tag{192}$$

$$= \mathbb{E}_{p_1,p_2,\dots,p_K \sim \mathcal{T}}[|\Pr[\mathsf{DaRRM}_\gamma(\mathcal{D}) = 1] - \Pr[g(\mathcal{S}) = 1]|] \tag{193}$$

$$= \mathbb{E}_{p_1,p_2,\dots,p_K \sim \mathcal{T}}\left[ \left| \sum_{l=\frac{K+1}{2}}^{K} \left( \alpha_l \cdot (\gamma(l) + \frac{1}{2}(1 - \gamma(l))) - \alpha_l \right) + \sum_{l=0}^{\frac{K-1}{2}} \alpha_l \cdot \frac{1}{2}(1 - \gamma(l)) \right| \right] \tag{194}$$

$$= \mathbb{E}_{p_1,p_2,\dots,p_K \sim \mathcal{T}}\left[ \left| \sum_{l=0}^{\frac{K-1}{2}} \alpha_l(\frac{1}{2}\gamma(l) - \frac{1}{2}) + \sum_{l=\frac{K+1}{2}}^{K} \alpha_l(\frac{1}{2} - \frac{1}{2}\gamma(l)) \right| \right] \tag{195}$$

The above follows by conditioning on $\mathcal{L} = l \in \{0, 1, \dots, K\}$, i.e. the sum of observed outcomes in $\mathcal{S}$

$$= \mathbb{E}_{p_1,p_2,\dots,p_K \sim \mathcal{T}}\left[ \left| \frac{1}{2} \sum_{l=\frac{K+1}{2}}^{K} (\alpha_l - \alpha_{K-l})(1 - \gamma(l)) \right| \right] \tag{196}$$

The above follows by symmetry of $\gamma$

Furthermore, notice the objective is symmetric around 0, and can be written as

$$\mathbb{E}_{p_1,p_2,\dots,p_K \sim \mathcal{T}}\left[ \frac{1}{2} \sum_{l=\frac{K+1}{2}}^{K} (\alpha_l - \alpha_{K-l})(1 - \gamma(l)) \right] \tag{197}$$

$$= \frac{1}{2} \mathbb{E}_{p_1,p_2,\dots,p_K \sim \mathcal{T}}\left[ \sum_{l=\frac{K+1}{2}}^{K} \left( (\alpha_l - \alpha_{K-l}) - (\alpha_l - \alpha_{K-l})\gamma(l) \right) \right] \tag{198}$$

$$= \underbrace{\frac{1}{2} \mathbb{E}_{p_1,p_2,\dots,p_K \sim \mathcal{T}}\left[ \sum_{l=\frac{K+1}{2}}^{K} (\alpha_l - \alpha_{K-l}) \right]}_{:=A} \underbrace{- \frac{1}{2} \mathbb{E}_{p_1,p_2,\dots,p_K \sim \mathcal{T}}\left[ \sum_{l=\frac{K+1}{2}}^{K} (\alpha_l - \alpha_{K-l})\gamma(l) \right]}_{:=B} \tag{199}$$

Since expression $A$ in Eq. 199 does not involve $\gamma$, we only need to optimize expression $B$ in Eq. 199. That is,

$$- \frac{1}{2} \mathbb{E}_{p_1,p_2,\dots,p_K \sim \mathcal{T}}\left[ \sum_{l=\frac{K+1}{2}}^{K} (\alpha_l - \alpha_{K-l})\gamma(l) \right] \tag{200}$$

$$= - \frac{1}{2} \sum_{l=\frac{K+1}{2}}^{K} \mathbb{E}_{p_1,p_2,\dots,p_K \sim \mathcal{T}}[(\alpha_l - \alpha_{K-l})] \cdot \gamma(l) \tag{201}$$

Eq. 201 is the optimization objective we use in the experiments. We see the optimization objective is linear in $\gamma$.

Note in the general setting, $\mathcal{L}(\mathcal{D}) \sim \mathrm{PoissonBinomial}(p_1, p_2, \dots, p_K)$, where recall $\mathcal{L}(\mathcal{D})$ is the sum of observed outcomes on dataset $\mathcal{D}$, and hence, $\alpha_l = \Pr[\mathcal{L}(\mathcal{D}) = l]$ is the pmf of the Poisson Binomial distribution at $l \in \{0, 1, \dots, K\}$.

## C.2  Practical Approximation of the Objective

Since the optimization objective in Eq. 200 requires taking an expectation over $p_1, \dots, p_K$, and this invovles integrating over $K$ variables, which can be slow in practice, we propose the following approximation to

efficiently compute the objective. We start with a simple idea to compute the objective, by sampling $p_i$'s from $[0, 1]$ and take an empirical average of the objective value over all subsampled sets of $p_1, \ldots, p_K$ as the approximation of the expectation in Section C.2.1. However, we found this approach is less numerically stable. We then propose the second approach to approximate the objective in Section C.2.2, which approximates the integration over $p_i$'s using the rectangular rule instead of directly approximating the objective value. We use the second approximation approach in our experiments and empirically demonstrates its effectiveness. **Note approximating the optimization objective does not affect the privacy guarantee.**

### C.2.1 Approximation via Direct Sampling of $p_i$'s

One straightforward way of efficiently computing an approximation to the optimization objective is as follows:

---
**Algorithm 4** Straightforward Approximation of the Optimization Objective

---
1: Input: # mechanisms $K \in \mathbb{N}$, # iterations $T \in \mathbb{N}$, noise function $\gamma : \{0, 1, \ldots, K\} \to [0, 1]$
2: **for** $t = 1, 2, \ldots, T$ **do**
3:   Sample $\hat{p}_1, \hat{p}_2, \ldots, \hat{p}_K \sim \mathcal{T}$
4:   $\widehat{\mathcal{L}} \leftarrow \text{PoissonBinomail}(\hat{p}_1, \ldots, \hat{p}_K)$
5:   $\hat{\alpha}_l \leftarrow \Pr[\widehat{\mathcal{L}} = l], \forall l \{0, \ldots, K\}$
6:   $g_t \leftarrow -\frac{1}{2} \sum_{l=\frac{K+1}{2}}^{K} (\hat{\alpha}_l - \hat{\alpha}_{K-l}) \cdot \gamma(l)$
7: **end for**
8: Return $\frac{1}{T} \sum_{t=1}^{T} g_t$

---

However, we found this approximation is not very numerically stable even for $T = 10000$ in the experiments and so we propose to adopt the second approximation as follows.

### C.2.2 Approximating the Integration Over $p_i$'s

Consider the following surrogate objective:

$$-\frac{1}{2} \sum_{l=\frac{K+1}{2}}^{K} \int_{0.5}^{1} \int_{0.5}^{1} \cdots \int_{0.5}^{1} (\alpha_l - \alpha_{K-l}) dp_1 dp_2 \ldots dp_K \cdot \gamma(l) \tag{202}$$

where we approximate the integration instead of directly approximating the objective value. The approximation of the integration is based on the rectangular rule and that the Poisson Binomial distribution is invariant to the order of its probability parameters.

First, we discretize the integration over $p_i$'s: pick $\tau = 50$ points representing probabilities between $[0.5, 1)$ with equal distance $\theta = \frac{0.5}{\tau}$. Denote this set of points as $\mathcal{W}$. We pick only $\tau = 50$ samples to ensure the distance between each sample, i.e., $\theta$, is not too small; or this can cause numerical instability. For each $l \in \{\frac{K+1}{2}, \frac{K+1}{2} + 1, \ldots, K\}$, we want to compute an approximated coefficient for $\gamma(l)$ as follows:

$$\int_{0.5}^{1} \int_{0.5}^{1} \cdots \int_{0.5}^{1} (\alpha_l - \alpha_{K-l}) dp_1 dp_2 \ldots dp_K \approx \sum_{p_1 \in \mathcal{W}} \sum_{p_2 \in \mathcal{W}} \cdots \sum_{p_K \in \mathcal{W}} (\alpha_l - \alpha_{K-l}) \tag{203}$$

which approximates integration over a $K$-dimensional grid $\mathcal{W}^K$.

The idea is then to sample points from this $K$-dimensional grid $\mathcal{W}^K$ and compute an empirical mean of the integration based on the sample probabilities for $p_1, \ldots, p_K$ from $\mathcal{W}^K$ as the approximation of the integration in the objective.

Let $(s_1, s_2, \ldots, s_K)$ be randomly sampled probability values from $\mathcal{W}^K$ and we want to compute $(\alpha_l - \alpha_{K-l})$ for all $l$ based on $(p_1, \ldots, p_K) = (s_1, \ldots, s_K)$. To apply the rectangular rule, since the grid of probabilities is $K$-dimensional, the weight of $(\alpha_l - \alpha_{K-l})$ in the approximate integration is $\theta^K$. Furthermore, observe

that $\alpha_l$ is the pmf at $l$ from a Poison Binomial distribution in our case, and PoissonBinomial$(p_1, \ldots, p_K) \overset{dist.}{\sim}$ PoissonBinomial$(\pi(p_1, \ldots, p_K))$, where $\pi$ denotes a permutation of $p_1, \ldots, p_K$ and $\overset{dist.}{\sim}$ denotes "the same distribution". Hence, with a single probability sample $(s_1, \ldots, s_K)$, we can indeed compute $\alpha_l - \alpha_{K-l}$ for each $l$ at $K!$ points from the grid $\mathcal{W}^K$, since they all have the same value. Therefore, we should set the weight of $\alpha_l - \alpha_{K-l}$ in the approximate integration as $w = \theta^K \cdot K!$. Furthermore, since the order of $(p_1, \ldots, p_K)$ does not affect the objective value, there is a total of ($\tau$ choose $K$ with replacement) $= \binom{\tau + K - 1}{K} := P$ different points in the grid $\mathcal{W}^K$.

In summary, the integration based approximation of the objective proceeds as follows:

---

**Algorithm 5** Integration Based Approximation of the Optimization Objective

---

1: Input: # mechanisms $K \in \mathbb{N}$, # iterations $T = 10000 \in \mathbb{N}$, noise function $\gamma : \{0, 1, \ldots, K\} \to [0, 1]$, $\tau = 50$: # samples between $[0.5, 1)$ to form the set $\mathcal{W}$
2: $\theta \leftarrow 0.5/\tau$ distance between samples
3: $w \leftarrow \theta^K \cdot K!$
4: $P \leftarrow \binom{\tau + K - 1}{K}$
5: **for** $t = 1, 2, \ldots, T$ **do**
6: $\quad$ Sample probabilities $(s_1, s_2, \ldots, s_K) \sim \mathcal{W}^K$
7: $\quad \widehat{\mathcal{L}} \sim \text{PoissonBinomial}(s_1, s_2, \ldots, s_K)$
8: $\quad \hat{\alpha}_l \leftarrow \Pr[\widehat{\mathcal{L}} = l], \forall l \in \{0, 1, \ldots, K\}$
9: $\quad g_t \leftarrow -\frac{1}{2} \sum_{l=\frac{K+1}{2}}^{K} w \cdot (\hat{\alpha}_l - \hat{\alpha}_{K-l}) \cdot \gamma(l)$
10: **end for**
11: Return $\frac{P}{N} \sum_{t=1}^{T} g_t$

---

## C.3 Reducing # Constraints from $\infty$ to a Polynomial Set

**Lemma C.1** (Restatement of Lemma 5.1). *Consider using DaRRM (Algorithm 1) to solve Problem 1.1 and let $f$ be the privacy cost objective as defined in Lemma 3.4. Given an arbitrary noise function $\gamma$, let the worst case probabilities be*

$$(p_1^*, \ldots, p_K^*, p_1'^*, \ldots, p_K'^*) = \underset{\{(p_i, p_i')\}_{i=1}^{K}}{\arg\max} \; f(p_1, \ldots, p_K, p_1', \ldots, p_K'; \gamma)$$

*Then, each pair $(p_i^*, p_i'^*), \forall i \in [K]$ satisfies*

$$(p_i^*, p_i'^*) \in \{(0, 0), (1, 1), (0, \Delta), (\Delta, 0), (1 - \Delta, 1),$$
$$(1, 1 - \Delta), (\frac{e^\epsilon + \Delta}{e^\epsilon + 1}, \frac{1 - \Delta}{e^\epsilon + 1}), (\frac{1 - \Delta}{e^\epsilon + 1}, \frac{e^\epsilon + \Delta}{e^\epsilon + 1})\}$$

*Furthermore, when $\delta > 0$, there exists a finite vector set $\mathcal{P}$ of size $O(K^7)$ such that if $\beta = \max_{\{(p_i, p_i')\}_{i=1}^{K} \in \mathcal{P}} f(p_1, \ldots, p_K, p_1', \ldots, p_K'; \gamma)$, then $f(p_1^*, \ldots, p_K^*, p_1'^*, \ldots, p_K'^*; \gamma) \leq \beta$. When $\delta = 0$, the size of $\mathcal{P}$ can be reduced to $O(K^3)$.*

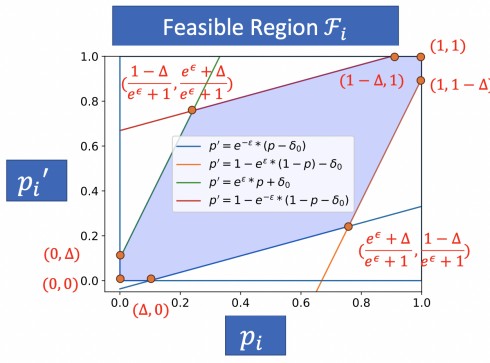

Figure 5: An illustration of the feasible region $\mathcal{F}_i$.

*Proof.* **Part I: Reducing # privacy constraints from $\infty$ to exponentially many.**

Consider $(p_i, p_i')$ for an arbitrary $i \in [K]$ and fixing $(p_j, p_j'), \forall j \neq i$. Given any noise function $\gamma$, recall the privacy cost objective $f(p_1, \ldots, p_K, p_1', \ldots, p_K'; \gamma)$ (see Lemma 3.4), is

$$f(p_1, \ldots, p_K, p_1', \ldots, p_K'; \gamma) = \sum_{l=0}^{\frac{K-1}{2}} (e^{m\epsilon}\alpha_l' - \alpha_l) \cdot \gamma(l) + \sum_{l=\frac{K+1}{2}}^{K} (\alpha_l - e^{m\epsilon}\alpha_l') \cdot \gamma(l)$$

and the privacy constraints are of the form

$$f(p_1, \ldots, p_K, p_1', \ldots, p_K'; \gamma) \leq e^{m\epsilon} - 1 + 2\delta$$

where recall that $\alpha_l = \Pr[\mathcal{L}(\mathcal{D}) = l]$ is a function of $\{p_i\}_{i=1}^K$ and $\alpha_l' = \Pr[\mathcal{L}(\mathcal{D}') = l]$ is a function of $\{p_i'\}_{i=1}^K$, $\forall l \in \{0, 1, \ldots, K\}$ and $\mathcal{L}(\mathcal{D})$, $\mathcal{L}(\mathcal{D}')$ are the sum of observed outcomes on neighboring datasets $\mathcal{D}$ and $\mathcal{D}'$. By Lemma 3.4, $\gamma$ needs to make the above privacy constraint hold for all possible $\{(p_i, p_i')\}_{i=1}^K$ to make DaRRM$_\gamma$ $(m\epsilon, \delta)$-differentially private. This is equivalent to saying, $\gamma$ needs to ensure $\max_{\{(p_i, p_i')\}_{i=1}^K} f(p_1, \ldots, p_K, p_1', \ldots, p_K'; \gamma) \leq e^{m\epsilon} - 1 + 2\delta$.

Notice that the sum of observed outcomes follows a Poisson Binomial distribution, i.e., $\mathcal{L}(\mathcal{D}) \sim$ PoissonBinomial$(p_1, \ldots, p_K)$ and $\mathcal{L}(\mathcal{D}') \sim$ PoissonBinomial$(p_1', \ldots, p_K')$. Hence, by the pmf of the Poisson Binomial distribution[6], the privacy cost objective $f$ is linear in each $p_i$ and $p_i'$, fixing all $(p_j, p_j'), \forall j \neq i$. Since each mechanism $M_i$ is $(\epsilon, \Delta)$-differentially private, by definition, $(p_i, p_i')$ satisfies all of the following:

$$p_i \leq e^\epsilon p_i' + \Delta, \quad p_i' \leq e^\epsilon p + \Delta$$
$$1 - p_i \leq e^\epsilon (1 - p_i') + \Delta, \quad 1 - p_i' \leq e^\epsilon (1 - p_i) + \Delta$$

That is, $(p_i, p_i')$ lies in a feasible region $\mathcal{F}_i$ (see Figure 5). Note the constraints on $(p_i, p_i')$, that is, the boundaries of $\mathcal{F}_i$, are linear in $p_i$ and $p_i'$. And so the optimization problem $(p_i^*, p_i'^*) = \arg\max_{(p_i, p_i')} f(p_1, \ldots, p_K, p_1', \ldots, p_K'; \gamma)$, which finds the worst case probabilities in $(p_i, p_i')$, is a Linear Programming (LP) problem in $(p_i, p_i')$ for $i \in [K]$. This implies $(p_i^*, p_i'^*)$ has to be on one of the eight corners of $\mathcal{F}_i$ — that is $(p_i^*, p_i'^*) \in \{(0, 0), (1, 1), (0, \Delta), (\Delta, 0), (1 - \Delta, 1), (1, 1 - \Delta), (\frac{e^\epsilon + \Delta}{e^\epsilon + 1}, \frac{1 - \Delta}{e^\epsilon + 1}), (\frac{1 - \Delta}{e^\epsilon + 1}, \frac{e^\epsilon + \Delta}{e^\epsilon + 1})\} := \mathcal{C}$. Since all $(p_i, p_i')$ and $(p_j, p_j')$, for $i \neq j$, are independent, we can search for the worst case probabilities by searching for $(p_i^*, p_i'^*) \in \mathcal{C}$, instead of searching for $(p_i, p_i') \in \mathcal{F}_i, \forall i \in [K]$. Therefore, the infinitely many privacy constraints are now reduced to only $8^K$ to optimize for the best $\gamma$ function that maximizes the utility of DaRRM$_\gamma$, while ensuring the output is $m\epsilon$-differentially private.

**Part II: Reducing # privacy constraints from exponentially many to a polynomial set.**

To further reduce the number of privacy constraints in optimization, observe that the Poisson Binomial distribution is invariant under the permutation of its parameters. That is, PoissonBinomial$(p_1, \ldots, p_K) \overset{dist.}{\sim}$

---

[6]See, e.g. https://en.wikipedia.org/wiki/Poisson_binomial_distribution, for the pmf of Poisson Binomial distribution.

PoissonBinomial($\pi(p_1, \ldots, p_K)$), for some permutation $\pi$ and $\overset{dist.}{\sim}$ means "follows the same distribution". Similarly, PoissonBinomial($p_1', \ldots, p_K'$) $\overset{dist.}{\sim}$ PoissonBinomial($\pi(p_1', \ldots, p_K')$).

The above observation implies if we have one privacy constraint $f(p_1 = v_1, \ldots, p_K = v_K, p_1' = v_1', \ldots, p_K' = v_K'; \gamma) \le e^{m\epsilon} - 1 + 2\delta$, for some $\{(v_i, v_i')\}_{i=1}^K \in \mathcal{C}^K$, then any privacy constraint $f(p_1 = s_1, \ldots, p_K = s_K, p_1' = s_1', \ldots, p_K' = s_K'; \gamma) \le e^{m\epsilon} - 1 + 2\delta$, where $(s_1, \ldots, s_K) = \pi_1(v_1, \ldots, v_K)$, $(s_1', \ldots, s_K') = \pi(v_1', \ldots, v_K')$, for permutations $\pi_1$ and $\pi_2$, is redundant.

Therefore, there is a vector set $\mathcal{P}$, where each probability vector $(p_1, \ldots, p_K, p_1', \ldots, p_K')$ in $\mathcal{P}$ is constructed by setting $(p_1, p_1'), (p_2, p_2'), \ldots, (p_K, p_K') = (v_1, v_2, \ldots, v_K)$, where $v_i \in \mathcal{C}, \forall i \in [K]$, such that vectors constructed by $(p_1, p_1'), (p_2, p_2'), \ldots, (p_K, p_K') = \pi(v_1, v_2, \ldots, v_K)$ is not in $\mathcal{P}$. Note $|\mathcal{P}| = $ (8 chooses K with replacement) $= \binom{K+8-1}{K} = O(K^7)$. If we can restrict our search for the worst case probabilities to this set $\mathcal{P}$ — that is, solving for $\beta := \max_{\{(p_i, p_i')\}_{i=1}^K \in \mathcal{P}} f(p_1, \ldots, p_K, p_1', \ldots, p_K'; \gamma)$, then $f(p_1^*, \ldots, p_K^*, p_1'^*, \ldots, p_K'^*; \gamma) \le \beta$. This implies we only need $O(K^7)$ privacy constraints to optimize for the best noise function $\gamma$ in DaRRM, while making sure DaRRM$_\gamma$ is $m\epsilon$-differentially private.

Note if $\Delta = 0$, i.e., the mechanism $M_i$'s are pure differentially private, the feasible region $\mathcal{F}_i$ in which $(p_i, p_i')$ lies has only 4 corners instead of 8. This implies $(p_i^*, p_i'^*) \in \mathcal{C} = \{(0,0), (1,1), (\frac{e^\epsilon}{e^\epsilon+1}, \frac{1}{e^\epsilon+1}), (\frac{1}{e^\epsilon+1}, \frac{e^\epsilon}{e^\epsilon+1})\}$. Hence, in this case, $|\mathcal{P}| = $ (4 choose $K$ with replacement) $= \binom{K+4-1}{K} = O(K^3)$, which implies we only need $O(K^3)$ privacy constraints to optimize for the best noise function $\gamma$ in DaRRM.

$\square$

# D    Full Experiment Results

## D.1    Optimized $\gamma$ in Simulations

### D.1.1    Comparison Using General Composition

The general composition (Theorem 2.3) indicates less total privacy loss than simple composition (Theorem 2.2) when the number of folds, $m$, is large, or when the failure probability $\delta$ is large. To enable meaningful comparison against general composition, we consider a larger $K$ and a larger failure probability $\delta$.

Consider $K = 35, \epsilon = 0.1, \Delta = 10^{-5}$. By general composition, if one outputs the majority of $M$ subsampled mechanisms for some $M < K$, the majority output is $(\epsilon_{opt}, \delta_{opt})$-differentially private, where

$$\epsilon_{opt} = \min\left\{ M\epsilon, \frac{(e^\epsilon - 1)\epsilon M}{e^\epsilon + 1} + \epsilon\sqrt{2M \log(e + \frac{\sqrt{M\epsilon^2}}{\delta'})}, \frac{(e^\epsilon - 1)\epsilon M}{e^\epsilon + 1} + \epsilon\sqrt{2M \log(\frac{1}{\delta'})} \right\}, \quad \delta_{opt} = 1 - (1 - \delta)^M(1 - \delta')$$

for some $\delta' \geq 0$. We set this as the privacy guarantee of all majority ensembling algorithms. That is, if we want the majority output to be $(m\epsilon, \delta)$-differentially private, we set

$$m = \frac{\epsilon_{opt}}{\epsilon} = \min\left\{ M, \frac{(e^\epsilon - 1)M}{e^\epsilon + 1} + \sqrt{2M \log(e + \frac{\sqrt{M\epsilon^2}}{\delta'})}, \frac{(e^\epsilon - 1)M}{e^\epsilon + 1} + \sqrt{2M \log(\frac{1}{\delta'})} \right\}$$

and $\delta = 1 - (1 - \delta)^M(1 - \delta')$ accordingly. The parameters $\tau$ and $\lambda$ to compute $p_{const}$ in RR (see Section A.1) are set to be

$$\tau = \min\left\{ K, \frac{(e^\epsilon - 1)K}{e^\epsilon + 1} + \sqrt{2K \log(e + \frac{\sqrt{K\epsilon^2}}{\delta'})}, \frac{(e^\epsilon - 1)K}{e^\epsilon + 1} + \sqrt{2K \log(\frac{1}{\delta'})} \right\}$$

and $\lambda = 1 - (1 - \delta)^K(1 - \delta')$.

In the experiments, we consider $M = \{10, 13, 15, 20\}$ and $\delta' = 0.1$; and $\gamma_{opt}$ is computed using a uniform prior $\mathcal{T}$.

All values of the parameters of the private ensembling algorithms we use in the experiment are listed in the table:

| # Subsampled mechanisms | $M$ | 10 | 13 | 15 | 20 |
|---|---|---|---|---|---|
| Privacy allowance | $m$ | 6.4521 | 7.5742 | 8.2708 | 9.8823 |
| Parameter of constant $\gamma$ | $\tau$ | 14.0328 | 14.0328 | 14.0328 | 14.0328 |
| Parameter of constant $\gamma$ | $\lambda$ | 0.1003 | 0.1003 | 0.1003 | 0.1003 |
| Overall privacy loss | $m\epsilon$ | 0.6452 | 0.7574 | 0.8271 | 0.9882 |
| Overall failure probability | $\delta$ | 0.1001 | 0.1001 | 0.1001 | 0.1002 |

Table 3: All parameter values. Note that all the private ensembling algorithms we compare in the experiment is required to be $(m\epsilon, \delta)$-differentially private. Here, $K = 35, \epsilon = 0.1, \Delta = 10^{-5}$ and $\delta' = 0.1$.

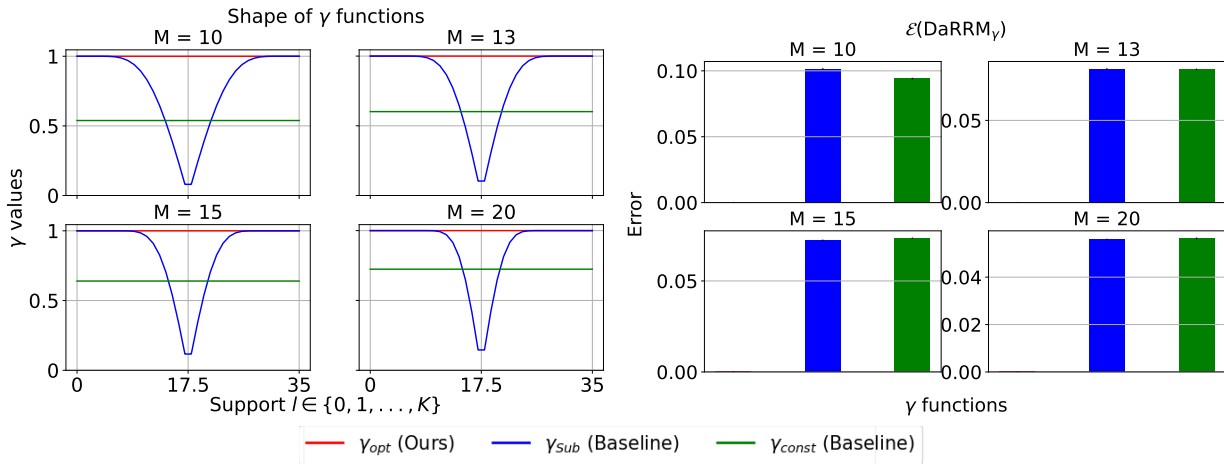

Figure 6: Plots of the shape and $\mathcal{E}(\mathsf{DaRRM}_\gamma)$ of different $\gamma$ functions: the optimized $\gamma_{Sub}$, and the baselines $\gamma_{Sub}$ (corresponding to subsampling) and $\gamma_{const}$ (corresponding to RR). Here, $K = 35, M \in \{10, 13, 15, 20\}$, $\Delta = 10^{-5}$, $\epsilon = 0.1$, $\delta' = 0.1$.

### D.1.2 Comparison in Pure Differential Privacy Settings

Consider the pure differential privacy setting, where $\Delta = \delta = 0$. Note in this setting, it is known that simple composition is tight.

To compute an optimized $\gamma_{opt}$ in $\mathsf{DaRRM}$, since we have shown the number of constraints is $O(K^3)$ if $\Delta = \delta = 0$ (see Lemma 5.1), we can set $K$ to be larger. Here, we present results for $K \in \{11, 101\}$ and $\epsilon = 0.1$.

Again, we compare the shape of different $\gamma$ and the corresponding $\mathcal{E}(\mathsf{DaRRM}_\gamma)$ under those $\gamma$ functions, fixing the total privacy loss to be $m\epsilon$. $\gamma_{opt}$ is computed using a uniform prior $\mathcal{T}$.

Since the subsampling mechanism from Section 4 with privacy amplification applies to this setting, we compare four different $\gamma$ noise functions here:

1. $\gamma_{opt}$ (Ours): optimized $\gamma$ function using our optimization framework

2. $\gamma_{Sub}$ (Baseline): the $\gamma$ function that corresponds to outputting the majority of $m$ out $K$ subsampled mechanisms

3. $\gamma_{DSub}$ (Baseline): the $\gamma$ function that corresponds to outputting $2m - 1$ subsampled mechanisms from Theorem 4.1, aka., Double Subsampling (DSub)

4. $\gamma_{const}$ (Baseline): the constant $\gamma$ function that corresponds to the classical Randomized Response (RR) algorithm

**Setting 1.** $K = 11$, $m \in \{1, 3, 5, 7, 9, 11\}$.

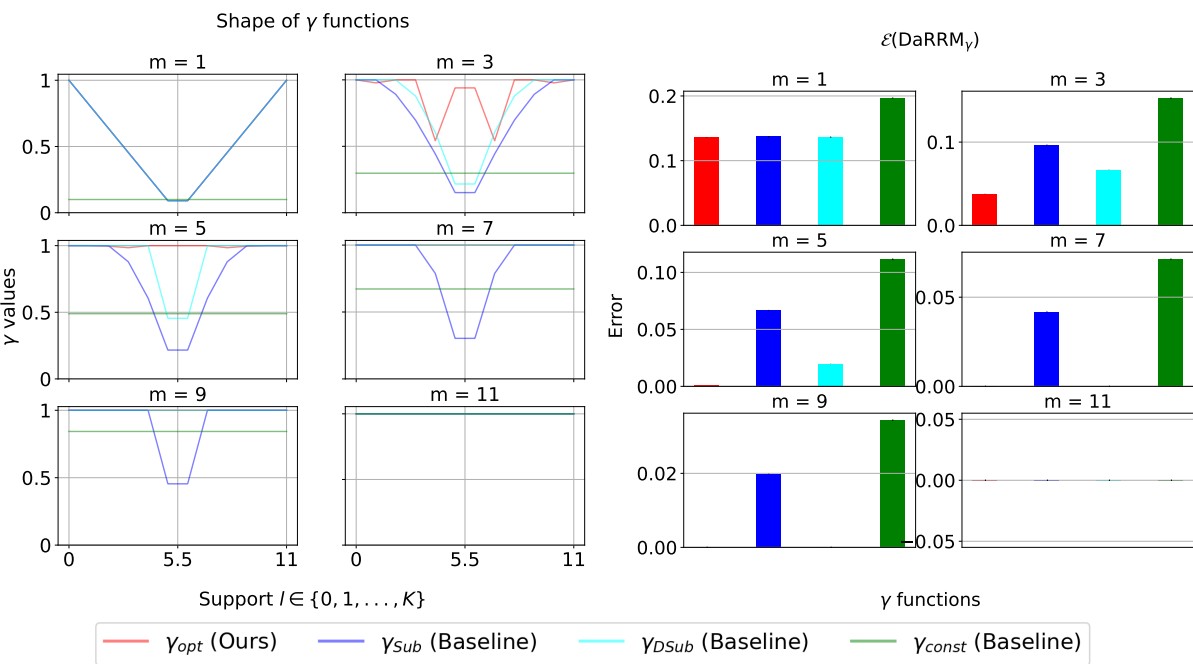

Figure 7: Plots of shape and $\mathcal{E}(\text{DaRRM}_\gamma)$ of different $\gamma$ functions: the optimized $\gamma_{Opt}$, the baselines $\gamma_{Sub}$ and $\gamma_{DSub}$ (Theorem 4.1), and the constant $\gamma_{const}$ (corresponding to RR). Here, $K = 11, m \in \{1, 3, 5, 7, 9, 11\}$, $\epsilon = 0.1$ and $\delta = \Delta = 0$. Note when $m \in \{7, 9\}$, the cyan line ($\gamma_{DSub}$) and the red line ($\gamma_{opt}$) overlap. When $m = 11$, all lines overlap. Observe that when $m \geq \frac{K+1}{2}$, that is, $m \in \{7, 9, 11\}$ in this case, the above plots suggest both $\gamma_{opt}$ and $\gamma_{DSub}$ achieve the minimum error at 0. This is consistent with our theory.

**Setting 2.** $K = 101, m \in \{10, 20, 30, 40, 60, 80\}$.

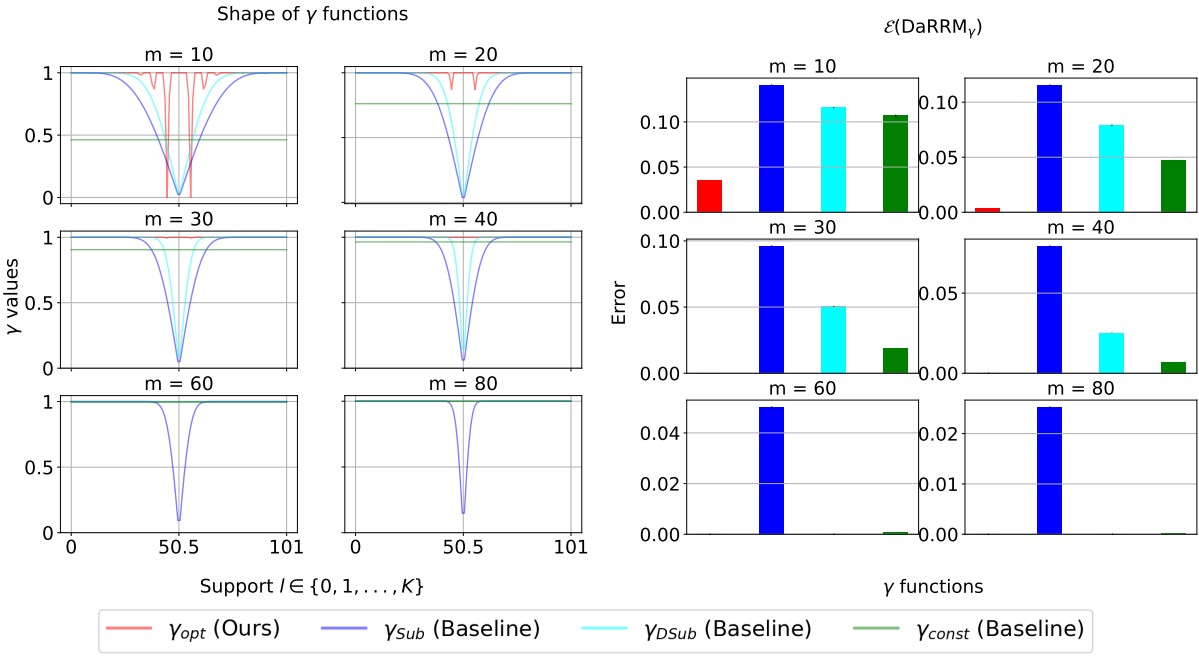

Figure 8: Plots of shape and $\mathcal{E}(\text{DaRRM}_\gamma)$ of different $\gamma$ functions: the optimized $\gamma_{Opt}$, the baselines $\gamma_{Sub}$ and $\gamma_{DSub}$ (Theorem 4.1), and the constant $\gamma_{const}$ (corresponding to RR). Here, $K = 101, m \in \{10, 20, 30, 40, 60, 80\}$, $\epsilon = 0.1$ and $\delta = \Delta = 0$.

### D.1.3 Comparison Using Different Prior Distributions

When optimizing $\gamma$ that maximizes the utility in DaRRM, recall that the objective takes an expectation over $p_i$'s for $p_i \sim \mathcal{T}$, where $\mathcal{T}$ is some distribution and $p_i = \Pr[M_i(\mathcal{D}) = 1]$. The previous experiments assume we do not have access to any prior knowledge about $p_i$'s and hence $\mathcal{T}$ is the uniform distribution, i.e., $\text{Uniform}([0,1])$. However, when one has knowledge about the mechanisms, one can set a proper prior $\mathcal{T}$ to further maximize the utility of DaRRM.

In this section, let $\mathcal{T}_U$ denote $\text{Uniform}([0,1])$ and we present results considering a different prior distribution, which we call $\mathcal{T}_P$, as follows. Suppose our prior belief is that each mechanism $M_i$ has a clear tendency towards voting 0 or 1, i.e., $p_i$ is far from 0.5. Let $\mathcal{T}_P$ be $\text{Uniform}([0,0.3] \cup [0.7,1])$.

To optimize $\gamma$ under $\mathcal{T}_P$, we change the approximate optimization objective in Eq. 202, which optimizes $\gamma$ under $\mathcal{T}_U$, to be the following,

$$-\frac{1}{2} \sum_{l=\frac{K+1}{2}}^{K} \int_{0.7}^{1} \int_{0.7}^{1} \cdots \int_{0.7}^{1} (\alpha_l - \alpha_{K-l}) dp_1 dp_2 \ldots dp_K \cdot \gamma(l) \tag{204}$$

**Setting.** $K = 11, m \in \{3, 5\}, \epsilon = 0.1, \delta = \Delta = 0$.

We compare the shape and $\mathcal{E}(\text{DaRRM}_\gamma)$ of different $\gamma$ functions:

1. $\gamma_{opt-U}$ denote the $\gamma$ function optimized under $p_i \sim \mathcal{T}_U$

2. $\gamma_{opt-P}$ denote the $\gamma$ function optimized under $p_i \sim \mathcal{T}_P$

3. $\gamma_{Sub}$, corresponding to the subsampling baseline

4. $\gamma_{const}$, corresponding to the RR baseline

Note when we compute the error, we take the expectation w.r.t. the actual $p_i$ distributions, regardless of the prior used to optimize $\gamma$. In the experiments, we consider three different actual $p_i$ distributions:"

1. "Actual: $\text{Uniform}([0,1])$": $p_i \sim \mathcal{T}_U, \forall i \in [K]$

2. "Actual: $p_i = 0.5$": $p_i = 0.5, \forall i \in [K]$

   This setting implies the mechanisms do not have a clear majority

3. "Actual: $\text{Uniform}([0,0.1])$": $p_i \sim \text{Uniform}([0,0.1]), \forall i \in [K]$

   This setting implies the mechanisms have a clear majority (i.e., 0)

Since our prior $\mathcal{T}_P$ is closer to $\text{Uniform}([0,0.1])$ (i.e., there is a clear majority), we would expect $\mathcal{E}(\text{DaRRM}_{\gamma_{opt-P}})$ to be the lowest when $p_i \sim \text{Uniform}[0,0.1]$, but to be higher than $\mathcal{E}(\text{DaRRM}_{\gamma_{opt-U}})$ when $p_i \sim \text{Uniform}([0,1])$ or $p_i = 0.5$. The results are presented in Figure 9.

## D.2 Private Semi-Supervised Knowledge Transfer

### D.2.1 More Details about the Baseline GNMax Papernot et al. (2018)

The GNMax aggregation mechanism for majority ensembling of *non-private* teachers proceeds as follows (Section 4.1 of Papernot et al. (2018)): on input $x$,

$$M_\sigma(x) = \arg \max_i \{n_i(x) + \mathcal{N}(0, \sigma^2)\} \tag{205}$$

where $n_i(x)$ is # teachers who vote for class $i$.

**How to set $\sigma$ in GNMax?**

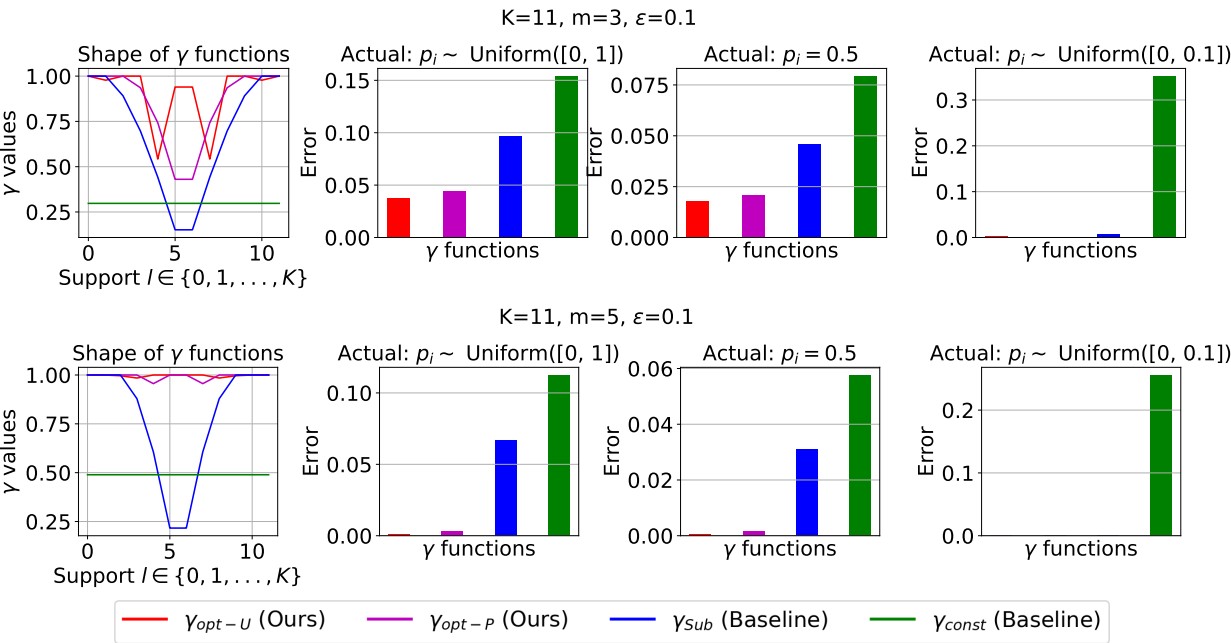

Figure 9: Comparison of the shape and $\mathcal{E}(\mathsf{DaRRM}_\gamma)$ of different $\gamma$ functions: 1) $\gamma$ optimized under prior $\mathcal{T}_U$, 2) $\gamma$ optimized under prior $\mathcal{T}_P$, 3) $\gamma_{Sub}$ (corresponding to the subsampling baseline) and 4) $\gamma_{const}$ (corresponding to the RR baseline). Here, $K = 11, m \in \{3, 5\}, \epsilon = 0.1$. Observe that if the prior $\mathcal{T}_P$ used in optimizing $\gamma$ is closer to the actual distribution of $p_i$'s, there is additional utility gain (i.e., decreased error); otherwise, we slightly suffer a utility loss (i.e., increased error), compared to optimize $\gamma$ under the $\mathcal{T}_U$ prior. Furthermore, regardless of the choice of the prior distribution $\mathcal{T}$ in optimizing $\gamma$, $\mathsf{DaRRM}_\gamma$ with an optimized $\gamma$ achieves a lower error compared to the the baselines.

Section 4.1 of Papernot et al. (2018) states the GNMax mechanism is $(\lambda, \lambda/\sigma^2)$-Renyi differentially private (RDP), for all $\lambda \geq 1$. RDP bounds can be converted to DP bounds as follows:

**Theorem D.1** (RDP to DP (Theorem 5 of Papernot et al. (2018))). *If a mechanism M guarantees $(\lambda, \epsilon)$-RDP, then M guarantees $(\epsilon + \frac{\log 1/\delta}{\lambda-1}, \delta)$-differential privacy for $\delta \in (0,1)$.*

Therefore, GNMax with parameter $\sigma^2$ guarantees $(\frac{\lambda}{\sigma^2} + \frac{\log 1/\delta}{\lambda-1}, \delta)$-differential privacy, $\forall \lambda \geq 1$. Given $m, \epsilon, \Delta$, we want to choose $\lambda$ and $\sigma^2$ here so that the output of GNMax is $(m\epsilon, m\Delta)$-differentially private. Here, $\delta = m\Delta$.

We first obtain a valid range of $\lambda$. Since $m\epsilon \geq 0$, $\frac{\lambda}{\sigma^2} + \frac{\log 1/\delta}{\lambda-1} \geq 0$ and so $\lambda \geq \frac{\log 1/\delta}{m\epsilon} + 1 := \lambda_{min}$. And $\sigma^2 = \frac{\lambda}{m\epsilon - \frac{\log 1/\delta}{\lambda-1}}$. Since the smaller $\sigma^2$ is, the higher the utility, we perform a grid search over $\lambda \in [\lambda_{min}, 500]$, with discretized $\lambda$ values of equal distance 0.5, to find the minimum $\sigma^2_{min}$. For the $(m\epsilon, m\Delta)$ values used in the experiments, we observe $\sigma^2$ decreases first and then increases as $\lambda$ increases, as shown in Figure 10. The $\lambda$ and $\sigma_{min}$ values in the RDP bound of Gaussian noise to compute the privacy loss of GNMax's output we use in the experiments are presented in Table 4.

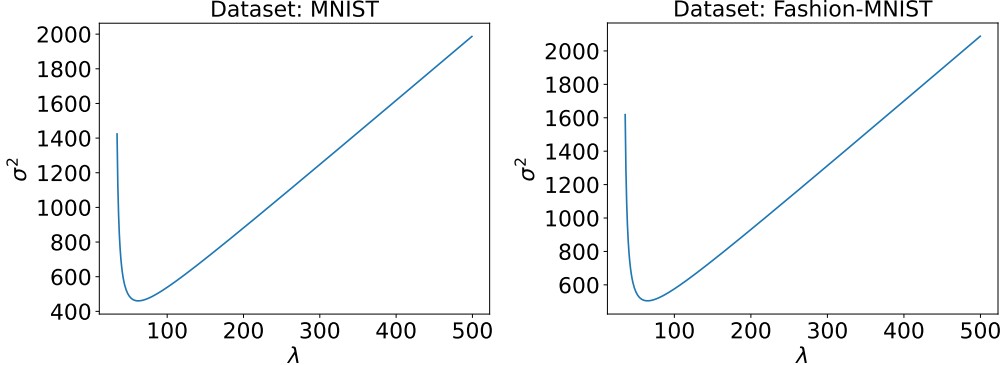

Figure 10: Plots of $\lambda$ vs. $\sigma^2$ in the Gaussian RDP privacy bound. The goal is to choose a $\lambda$ value that minimizes $\sigma^2$. It is not hard to see the value of $\sigma^2$ decreases at first and then increases as $\lambda$ increases.

|  | Privacy Loss Per Query $(m\epsilon, m\Delta)$ | $\lambda$ | $\sigma_{min}$ |
|---|---|---|---|
| MNIST | $(0.2676, 0.0003)$ | 34.31 | 21.46 |
| Fashion-MNIST | $(0.2556, 0.0003)$ | 35.74 | 22.46 |

Table 4: Parameters of the RDP bound of Gaussian noise to compute the privacy loss of GNMax's output.

**A Note on the Data-dependent Privacy Loss Bound**

Papernot et al. (2018) gives a potentially tighter data-dependent bound on the privacy loss using GNMax to output the majority of non-private teacherss votes. We give a clean pseudo-code on computing the data-dependent privacy loss bound in Algorithm 6, based on the lemmas and theorems in Papernot et al. (2018). Given privacy parameters $\sigma, \lambda$ and the teacher votes per class $\{n_i\}_{i=1}^{C}$ for $C$ classes, the data-dependent bound can be empirically evaluated and compared against the Gaussian privacy loss bound. The smaller one is the final privacy loss. We empirically find that the condition of the data-dependent bound (line 8 in Algorithm 6) is not satisfied when $K$ and the number of classes $C$ are small, e.g., $K = 11, C = 2$ as in our case, even if all teachers agree on the same output. And so in the experiments, we can only apply the Gaussian privacy loss bound (line 14).

### D.2.2 Additional Results for Private Semi-Supervised Knowledge Transfer

$m = 1$.

---

**Algorithm 6** Compute Tighter Privacy Loss

---

1: Input: Std. of Gaussian noise $\sigma$, Privacy parameter $\lambda$, # teachers $K$, # classes $C$, # votes per class $\{n_i\}_{i=1}^C$
2: $\mathcal{B} \leftarrow \{\}$ bound candidates
3: **for** $i = 1, 2, \ldots, K$ **do**
4: $\quad q^{(i)} \leftarrow \frac{1}{2} \sum_{i \neq i^*} \text{erfc}(\frac{n_{i^*} - n_i}{2\sigma})$
5: $\quad \mu_2^{(i)} \leftarrow \sigma \cdot \sqrt{\log 1/q^{(i)}}, \mu_1^{(i)} \leftarrow \mu_2^{(i)} + 1$
6: $\quad \epsilon_1^{(i)} \leftarrow \frac{\mu_1^{(i)}}{\sigma^2}, \epsilon_2^{(i)} \leftarrow \frac{\mu_2^{(i)}}{\sigma^2}$
7: $\quad q_{ub}^{(i)} \leftarrow \exp((\mu_2^{(i)} - 1)\epsilon_2^{(i)})/(\frac{\mu_1^{(i)}}{\mu_1^{(i)}-1} \cdot \frac{\mu_2^{(i)}}{\mu_2^{(i)}-1})^{\mu_2^{(i)}}$
8: $\quad$ **if** $q^{(i)} < 1$ and $\mu_1^{(i)} \geq \lambda$ and $\mu_2 > 1$ and $q^{(i)} \leq q_{ub}^{(i)}$ **then**
9: $\quad\quad A^{(i)} \leftarrow (1 - q^{(i)})/(1 - q^{(i)} \cdot \exp(\epsilon_2^{(i)})^{\frac{\mu_2^{(i)}-1}{\mu_2^{(i)}}})$
10: $\quad\quad B^{(i)} \leftarrow \exp(\epsilon_1^{(i)})/(q^{(i)})^{\frac{1}{\mu_1^{(i)}-1}}$
11: $\quad\quad \text{DataDependentBound} \leftarrow \frac{1}{\lambda-1} \cdot \left((1 - q^{(i)}) \cdot (A^{(i)})^{\lambda-1} + q^{(i)} \cdot (B^{(i)})^{\lambda-1}\right)$
12: $\quad\quad \mathcal{B} \leftarrow \mathcal{B} \cup \text{DataDependentBound}$
13: $\quad$ **else**
14: $\quad\quad \text{GaussianBound} \leftarrow \frac{\lambda}{\sigma^2}$
15: $\quad\quad \mathcal{B} \leftarrow \mathcal{B} \cup \text{GaussianBound}$
16: $\quad$ **end if**
17: **end for**
18: Return $\min \mathcal{B}$

---

| Dataset | # Queries | Privacy loss per query $(\epsilon_{query}, \delta_{query})$ | Total privacy loss over $Q$ queries $(\epsilon_{total}, \delta_{total})$ |
|---|---|---|---|
| MNIST | $Q = 20$ | | (1.704, 0.002) |
| | $Q = 50$ | (0.0892, 0.0001) | (2.837, 0.005) |
| | $Q = 100$ | | (4.202, 0.010) |
| Fashion MNIST | $Q = 20$ | | (1.620, 0.002) |
| | $Q = 50$ | (0.0852, 0.0001) | (2.695, 0.005) |
| | $Q = 100$ | | (3.988, 0.010) |

Table 5: The privacy loss per query to the teachers and the total privacy loss over $Q$ queries. Note the total privacy loss is computed by general composition, where we set $\delta' = 0.0001$.

| Dataset | MNIST | | | Dataset | Fashion-MNIST | | |
|---|---|---|---|---|---|---|---|
| | GNMax | DaRRM$_{\gamma_{Sub}}$ | DaRRM$_{\gamma_{opt}}$ | | GNMax | DaRRM$_{\gamma_{Sub}}$ | DaRRM$_{\gamma_{opt}}$ |
| # Queries | (Baseline) | (Baseline) | (Ours) | # Queries | (Baseline) | (Baseline) | (Ours) |
| $Q = 20$ | 0.54 (0.11) | 0.68 (0.07) | **0.74 (0.08)** | $Q = 20$ | 0.56 (0.10) | **0.92 (0.05)** | 0.89 (0.06) |
| $Q = 50$ | 0.51 (0.07) | **0.67 (0.05)** | 0.66 (0.05) | $Q = 50$ | 0.52 (0.05) | 0.89 (0.04) | **0.92 (0.03)** |
| $Q = 100$ | 0.57 (0.03) | **0.71 (0.03)** | 0.69 (0.04) | $Q = 100$ | 0.56 (0.04) | 0.89 (0.04) | **0.91 (0.04)** |

Table 6: Accuracy of the predicted labels of $Q$ query samples on datasets MNIST (on the left) and Fashion-MNIST (on the right). We report the mean and one std. in parentheses over 10 random draws of the query samples from the test dataset. Note each prediction on the query sample is $(\epsilon_{total}, \delta_{total})$-differentially private. Note in this case where $m = 1$, by Lemma 3.2, subsampling achieves the optimal error/utility. Hence, there is not much difference in terms of accuracy between DaRRM$_{\gamma_{Sub}}$ and DaRRM$_{\gamma_{opt}}$ as expected.

$m = 5$.

| Dataset | # Queries | Privacy loss per query $(\epsilon_{query}, \delta_{query})$ | Total privacy loss over $Q$ queries $(\epsilon_{total}, \delta_{total})$ |
|---|---|---|---|
| MNIST | $Q = 20$ | $(0.4460, 0.0005)$ | $(8.920, 0.010)$ |
|  | $Q = 50$ |  | $(18.428, 0.025)$ |
|  | $Q = 100$ |  | $(28.926, 0.049)$ |
| Fashion MNIST | $Q = 20$ | $(0.4260, 0.0005)$ | $(8.520, 0.010)$ |
|  | $Q = 50$ |  | $(17.398, 0.025)$ |
|  | $Q = 100$ |  | $(27.223, 0.049)$ |

Table 7: The privacy loss per query to the teachers and the total privacy loss over $Q$ queries. Note the total privacy loss is computed by general composition, where we set $\delta' = 0.0001$.

| Dataset | MNIST | | | Dataset | Fashion-MNIST | | |
|---|---|---|---|---|---|---|---|
|  | GNMax | DaRRM$_{\gamma_{Sub}}$ | DaRRM$_{\gamma_{opt}}$ |  | GNMax | DaRRM$_{\gamma_{Sub}}$ | DaRRM$_{\gamma_{opt}}$ |
| # Queries | (Baseline) | (Baseline) | (Ours) | # Queries | (Baseline) | (Baseline) | (Ours) |
| $Q = 20$ | 0.73 (0.11) | 0.76 (0.09) | **0.84 (0.07)** | $Q = 20$ | 0.72 (0.10) | 0.96 (0.04) | **0.97 (0.04)** |
| $Q = 50$ | 0.75 (0.07) | 0.82 (0.04) | **0.83 (0.04)** | $Q = 50$ | 0.72 (0.08) | 0.96 (0.02) | **0.97 (0.02)** |
| $Q = 100$ | 0.72 (0.04) | 0.79 (0.05) | **0.83 (0.03)** | $Q = 100$ | 0.72 (0.06) | **0.97 (0.01)** | **0.97 (0.01)** |

Table 8: Accuracy of the predicted labels of $Q$ query samples on datasets MNIST (on the left) and Fashion-MNIST (on the right). We report the mean and one std. in parentheses over 10 random draws of the query samples from the test dataset. Note each prediction on the query sample is $(\epsilon_{total}, \delta_{total})$-differentially private. With the same per query privacy loss (and hence the same total privacy loss over $Q$ samples), DaRRM$_{\gamma_{opt}}$ achieves the highest accuracy compared to the other two baselines.

$m = 7$.

| Dataset | # Queries | Privacy loss per query $(\epsilon_{query}, \delta_{query})$ | Total privacy loss over $Q$ queries $(\epsilon_{total}, \delta_{total})$ |
|---|---|---|---|
| MNIST | $Q = 20$ | $(0.6244, 0.0007)$ | $(12.488, 0.014)$ |
|  | $Q = 50$ |  | $(28.392, 0.035)$ |
|  | $Q = 100$ |  | $(45.683, 0.068)$ |
| Fashion MNIST | $Q = 20$ | $(0.5964, 0.0007)$ | $(11.928, 0.014)$ |
|  | $Q = 50$ |  | $(26.738, 0.035)$ |
|  | $Q = 100$ |  | $(42.873, 0.068)$ |

Table 9: The privacy loss per query to the teachers and the total privacy loss over $Q$ queries. Note the total privacy loss is computed by general composition, where we set $\delta' = 0.0001$.

| Dataset | MNIST | | | Dataset | Fashion-MNIST | | |
|---|---|---|---|---|---|---|---|
|  | GNMax | DaRRM$_{\gamma_{Sub}}$ | DaRRM$_{\gamma_{opt}}$ |  | GNMax | DaRRM$_{\gamma_{Sub}}$ | DaRRM$_{\gamma_{opt}}$ |
| # Queries | (Baseline) | (Baseline) | (Ours) | # Queries | (Baseline) | (Baseline) | (Ours) |
| $Q = 20$ | 0.79 (0.07) | 0.80 (0.09) | **0.85 (0.08)** | $Q = 20$ | 0.79 (0.07) | 0.95 (0.04) | **0.96 (0.04)** |
| $Q = 50$ | 0.80 (0.05) | 0.82 (0.05) | **0.85 (0.04)** | $Q = 50$ | 0.79 (0.05) | 0.96 (0.03) | **0.97 (0.03)** |
| $Q = 100$ | 0.80 (0.04) | 0.80 (0.04) | **0.83 (0.03)** | $Q = 100$ | 0.79 (0.03) | 0.96 (0.02) | **0.96 (0.02)** |

Table 10: Accuracy of the predicted labels of $Q$ query samples on datasets MNIST (on the left) and Fashion-MNIST (on the right). We report the mean and one std. in parentheses over 10 random draws of the query samples from the test dataset. Note each prediction on the query sample is $(\epsilon_{total}, \delta_{total})$-differentially private. With the same per query privacy loss (and hence the same total privacy loss over $Q$ samples), DaRRM$_{\gamma_{opt}}$ achieves the highest accuracy compared to the other two baselines.

