# OpenReview forum: "Optimized Tradeoffs for Private Prediction with Majority Ensembling"
_TMLR — Accepted by TMLR_

### Review · Reviewer_KYVF · 2024-06-20

**Summary Of Contributions:**

This paper studies a classical problem for computing a differentially private majority from K-differentially private algorithms. They introduce a Data-dependent Randomized Response Majority (DaRRM) algorithm.

They show that an (mϵ, δ)-private majority algorithm with maximal utility can be computed tractably for any m ≤ K by a novel structural result that reduces the privacy constraints. In some settings, They show DaRRM provably enjoys a privacy gain of a factor of 2 over common baselines. They demonstrate the strong empirical effectiveness of the algorithm when compared against several baselines.

**Audience:**

Yes

**Broader Impact Concerns:**

This paper works for privacy, and it doesn't have ethical problems.

**Claims And Evidence:**

No

**Requested Changes:**

1. In Algorithm 1 Step 5, it should be $\gamma(L)$ rather than $\gamma(S)$. What is the function for $\gamma(L)$? as the $\gamma_{sub}$ in Lemma 3?
2. It says the DaRRM has a non-constant success probability $p_\gamma$, the success probability should be as least $p_\gamma$, since Step 10 also has$\frac{1}{2}(1-p_\gamma)$ success probability.
3. Why the probability use some combinations? Please explain intuitively.

Overall, the contribution of this paper is that it gets a better algorithm for privacy majority problem compared with traditional algorithms. But the comparisons are not clear, such as authors can show the error bound for this algorithm and previous algorithms.

**Strengths And Weaknesses:**

#### Strengths:
They give effective algorithm for differentially private majority problem.

#### Weakness:
1. The contributions should be summaried as short and to the points.
2. The notations of $\gamma$ and $\gamma_{sub}$ are mixed used. In some cases, they denote for the same.
3. Theorem 4.1 says it amplifies privacy by 2, the baseline is not clear here.
4. There are no error bounds analysis for the algorithm.
5. I have a question for Theorem 4.1. It say when $m \ge \frac{K+1}{2}$, just set $\gamma(l)=1$ to output the true majority with no noise. Can this protect $m\epsilon$ differential privacy? I try to understand that. This may because of Lemma 3.1, which says "the majority of m out of K-subsampled mechanisms without replacement and the output of data-dependent RR algorithm have the same distribution." So the algorithm's output distribution is the same as m samples' distribution, then how is it to get $e^{m\epsilon}$
6. The title is "optimized tradeoff for ...", and it doesn't show and prove the optimal results.

---

> ### Author Response · Authors · 2024-08-22
> **Review Response**
>
> We would like to thank the reviewer for his/her valuable feedback.
>
> Response to weakness:
>
> 1. We significantly shortened the contribution section and leave only the key points in the updated draft. We hope this gives the readers a more succinct picture of the paper.
>
> 2. $\gamma$ refers to a generic noise function used in our proposed DaRRM framework to compute a private majority, while $\gamma_{Sub}$ refers to the specific noise function that makes the output distribution of DaRRM the same as that of the subsampling baseline. In the context of discussing the subsampling baseline, we set $\gamma = \gamma_{Sub}$, while in other context, $\gamma$ can be other functions.
> We believe this should be clear in the draft. We would appreciate it if the reviewer could point to the places where $\gamma$ and $\gamma_{Sub}$ are mixed and potentially cause confusion.
>
> 3. The baseline Theorem 4.1 compares against is the subsampling baseline discussed in Section 3. This is mentioned in the introduction.
> We further clarified this in Section 4 and in the interpretation of Theorem 4.1 in the updated draft.
>
> 4. The error bound, or the utility bound, can be derived as closed form expressions straightforwardly using Definition 2.4. However, it is not straightforward to directly compare the error / utility bounds of different algorithms. Hence, we empirically compare the error of different algorithms in simulations in Section 6.1 and omit the actual messy utility expression to avoid confusion.
> However, we note that it is easy to compare the utility of different subsampling algorithms. Intuitively, the more outcomes the algorithm subsamples, the better the utility (and the less the error). We formalize this intuition in Lemma B.11 in Appendix B.3.
> Lemma B.11 can be used to compare the results in Theorem 4.1 against the subsampling baseline. Theorem 4.1 suggests in pure differential privacy settings, under a fixed privacy budget, one can subsample 2x more outcomes compared to the subsampling baseline, and hence, this implies an improved utility compared to the subsampling baseline.
>
> 5. The main reason that privacy is still possible even when $\gamma = 1$ is that each of the underlying mechanism is already private and so the truth-telling algorithm can be viewed as subsampling with $m = K$ . The difference between our algorithm in Theorem 4.1 and the subsampling baseline, which outputs the majority of $m$ outcomes/samples, is in the privacy analysis. If one uses only simple composition (which for a general problem is tight in terms of privacy analysis in the pure differential privacy setting) to reason about the privacy loss, then one cannot set $\gamma(l) = 1$ if $m < K$. The key here is that we know we are solving a specific problem, i.e., private majority ensembling, and this enables us to derive a tighter analysis on the privacy loss.  Essentially note that the majority of a set of mechanisms can be determined by revealing only half of the mechanisms' outputs, although such a choice needs to be non-random and requires revealing only the mechanisms with the majority vote. However, since simple composition works with adaptive non-random choices of mechanisms, therefore the privacy leakage is essentially halved.
>
> 6. We meant optimizing/improving the utility by "optimized" in this title, which is different from "optimal". In the updated draft, we are more careful about the notion of "optimality".

---

> > ### Author Response · Authors · 2024-08-22
> > **Review Response (Cont'd)**
> >
> > Response to requested changes:
> >
> > 1. Note that $\mathcal{S} = \{S_1, \dots, S_K\}$ is the set of outcomes from the $K$ underlying private mechanisms/teachers, and $\mathcal{L} = \sum_{i=1}^{K} S_i$ is the sum of the observed outcomes. We stated the proposed framework DaRRM (Algorithm 1) in a generic way at line 5, where $\gamma$, the noise function, can be defined on each individual outcome, i.e., $\gamma(\mathcal{S})$. However, as mentioned in section 3 in \textit{Designing the $\gamma$ Function}, the sum of outcomes $\mathcal{L}$ is a sufficient static for computing the majority and hence, defining $\gamma$ functions on $\mathcal{L}$ suffices. This simplifies the choice and analysis of different $\gamma$ noise functions. To make this clear, we explicitly stated in line 5 Algorithm 1 that ``in out setting, $p_\gamma\leftarrow \gamma(\mathcal{L})$''.
> > The function $\gamma(\mathcal{L})$ in Algorithm 1 is a hyperparameter. Choosing different $\gamma(\mathcal{L})$ functions leads to different algorithms computing the private majority and the goal is to choose one that satisfies the given privacy loss while maximizing the utility. For example, choosing the $\gamma$ functions as $\gamma_{Sub}$ in Lemma 3.1 recovers the subsampling baseline. In our proposed method to maximize the utility in section 5, $\gamma(\mathcal{L})$ is a function, which has a range of $K+1$ values, that is the output from the optimization procedure without a closed form expression.
> >
> > 2. As a convention from the classical Randomized Response mechanism,
> > the success probability here refers to the probability of faithfully revealing the true majority of the $K$ underlying mechanisms. In the proposed DaRRM framework, it is the probability that the biased coin (Algorithm 1 line 6) lands on head, i.e., $p_{\gamma}$.
> > The success probability is not the probability that the algorithm outputs 1.
> >
> > 3. By combinations in the probability, do you mean the sum of $j = \frac{m}{2}, \dots, m$ in Lemma 3.1 and the sum of $i = m, \dots, 2m-1$ in Theorem 4.1, where $m$ is the privacy budget? Here, the probability from each component $j$ (or $i$) denotes the probability of the algorithm outputting 1, conditioning on seeing the sum of outcomes $\mathcal{L} = l$ for some $l\in \{0, \dots, K\}$. Since in Lemma 3.1 the algorithm subsamples $m$ outcomes from the $K$ private mechanisms/teachers (similarly, in Theorem 4.1, the algorithm subsamples $2m-1$ outcomes), the algorithm outputs 1 when $j \geq \frac{m}{2}$ (or $i \geq \frac{2m-1}{2}$) and hence one gets such components in the probability.

---

### Review · Reviewer_Fx2m · 2024-07-30

**Summary Of Contributions:**

This manuscript proposes a randomized response framework that optimizes utility for differentially private majority ensembling algorithms. The proposed approach employs a data-dependent noise function, and its utility optimization is evaluated in the context of differential private label ensembling for image classification applications.

**Audience:**

Yes

**Broader Impact Concerns:**

No concerns about the ethical implications of the work

**Claims And Evidence:**

No

**Requested Changes:**

1)  Although the authors provide a thorough overview of the background on private aggregation of predictions, the motivation for applying private prediction in data-adaptive settings needs to be better explained. Including a real-world example involving a data-adaptive setting where the prediction ensembling should preserve privacy would significantly enhance the audience's understanding of the importance of the problem.


2)  The first sentence of the section “1.1 Our Contributions” needs clarification: “We give a (perhaps surprising) affirmative answer to this question: by using ..”. It is not clear which question the authors are referring to.

3) Although this work is theoretically supported, and the authors claim they “demonstrate the strong empirical effectiveness” of the proposed framework, the only datasets used for empirical evaluation are MNIST and FashionMNIST. I strongly suggest the authors extend their experiments by including more complex datasets such as SVHN.

4) The “Introduction” section mentions that this work focuses on “ data-adaptive settings where the underlying dataset is changing slowly over time.” However, it is unclear to me how this setting is incorporated into the experiments conducted on MNIST and FashionMNIST.  Could you please clarify how the experiments address this online dataset scenario?

5)  As a minor point, the readability of the sentence, “We emphasize in data-adaptive settings where the underlying.. ”  (on page 2, Introduction section), would improved by either removing “in” or replacing it with “on”.

**Strengths And Weaknesses:**

Strengths
- Studying privacy-utility trade-offs and developing an optimization framework for it is a compelling effort.

Weaknesses
- Limited experimental evaluation
- Some sections require clarification and revision

---

> ### Author Response · Authors · 2024-08-22
> **Review Response**
>
> We would like to thank the reviewer for his/her valuable feedback.
>
> Response to requested changes:
>
> 1/4/5. The main focus of the work is not a "data-adaptive setting'', or an online setting.
> The only place where the word "data-adaptive'' appears is in the following sentence in Section 1 Introduction: ``We emphasize in data-adaptive settings ...''. This sentence is merely to emphasize an interesting setting similar to the online setting where private prediction could be useful. We have re-written this sentence to avoid confusion.
>
> Instead, we focus on "data-dependent'' private majority ensembling algorithm design, where the amount of noise is added dependent on the dataset to maximize the utility. More specifically, it is dependent on the set of observed outcomes from the underlying private mechanisms, $S$, which is a random variable of the dataset and is a proxy.
> The draft already includes motivation and real-world examples of private majority ensembling in Section 1.
>
> 2. We explicitly stated the question before Contribution 1.1 in the updated draft. Hope this clarifies the first sentence of Contribution 1.1.
>
> 3. Our algorithm's performance does not directly depend on the dataset itself but instead depends on the performance of the underlying private mechanisms/teachers. We used CNN models in the experiments due to limited computational resources. Training on SVHN would require more complex models and more training time. These experiments primarily serve as a proof of concept, demonstrating the practical value of our theoretical framework.

---

### Review · Reviewer_kUU3 · 2024-08-05

**Summary Of Contributions:**

The paper considers the problem of releasing a majority vote from $K$ $(\epsilon, \Delta)$-DP algorithms that satisfies $(m \epsilon, \delta)$-DP for $1 <=m <=K$ and $0 <= \Delta <= \delta <1$. The authors propose a new randomized response (RR) variant, which uses a parameterized noise function. The main novelty in the formulation is that the noise function can be optimized for different queries, which allows RR to add differing amounts of noise depending on the input (i.e., it allows to add less or no noise when there is a clear-enough majority). The authors show that the proposed formulation is quite general, and can be in some sense optimal. The paper also includes some empirical experiments showing that the proposed method tends to work better than some existing baselines.

**Audience:**

Yes

**Broader Impact Concerns:**

I have no broader impact concerns for this paper.

**Claims And Evidence:**

No

**Requested Changes:**

In more or less decreasing order of importance:

1) On the optimality and efficiency of the proposed method: in several places (e.g., in the abstract, Sec 1.1 on p2 and p3, Sec 5 on p7, Sec7 on p11) it is claimed that the noise function $\gamma$ can be optimized efficiently, and as a result, the proposed method achieves optimal utility. However, looking at the comments in Lemma 5.1 about the optimization (Linear Optimization Objective on p7, Appendix E.2), finding the optimal value involves calculating a multidimensional integral, and since this is not really doable with larger $K$, an approximation is discussed in the Appendix and used in the experiments. To me this looks like the proposed method with any larger $K$ is not actually tractable, and hence when the noise function is only approximately optimal, the proposed method has no optimality guarantees (furthermore, the non-optimality does not seem to be only a theoretical possibility, as under some settings e.g. in Fig. 8 and in Table 6 the proposed method can perform worse than the baselines).

2) Please move at least the entire Related work discussing the closely-related existing methods to the main body of the text.

3) Please add missing existing RR optimality results in the related work and when discussing your results (at least Kairouz et al. 2015, Holohan et al. 2016).

4) On composition: besides basic and advanced composition, there is also the exact bound from Kairouz et al. 2015. Is there some reason for not using it in the experiments? Please also use the exact composition bound when stating results on the provable privacy amplification (Sec 4).

5) Can you characterize how does your optimality results relate to optimal count query release (which seems also like a natural baseline, i.e., directly release the (subsampled) noisy count with geometric mechanism, see e.g. Kairouz et al. 2015 for optimality discussion)?

6) Please state the neighborhood relation (e.g. replace?) explicitly, or that your results do not depend on the exact relation, to avoid misunderstandings.

### References:

Holohan et al. 2016: Optimal Differentially Private Mechanisms for Randomised Response.

Kairouz et al. 2015: The Composition Theorem for Differential Privacy.

**Strengths And Weaknesses:**

### Strengths

i) The proposed RR variant seem interesting: it seems to have a nice theoretical motivation (at least when looking at it without properly going through all of the proofs), and the reported performance is good under specific parameter ranges.

ii) General DP mechanism optimality results are very interesting and important, both in theory and for practical use cases.


### Weaknesses

i) The paper is generally not easy to read and could be clearly improved by clarifying the writing (e.g., there are plenty of breaks in the flow of thought [such as the first sentence of Sec1.1: what question is this talking about?], parts of the related work and many basic defs etc. are only in the appendix, in several places it is claimed that something is "hard", and then in the next sentence it is claimed that the hard problem can be easily solved). While shortening the paper by pushing most content into the Appendix is somewhat understandable, if lamentable, in a short-format conference paper, TMLR does not have such limits.

ii) I am not sure if all the claims about the optimality and efficiency of the proposed method are well-founded.

iii) There is not much discussion about the scalability of the proposed approach beyond some mentions, or on how the quality of the approximations depends on the available compute. Given that it seems to be maybe the most important bottle-neck for the proposed method, this seems a bit odd.

---

> ### Author Response · Authors · 2024-08-23
> **Review Response**
>
> We would like to thank the reviewer for his/her valuable feedback.
>
> Response to weakness:
>
> 1: We have revised the draft, paying specific attention to points raised by the reviewer. We hope this new version is more clear and more comprehensible. Please let us know if there are places that might cause confusion or where the presentation can be further improved.
>
> We want to clarify on the comment "the non-optimality does not seem to be only a theoretical possibility, as under some settings e.g. in Fig. 8 and in Table 6 the proposed method can perform worse than the baselines".
>
> In the right-hand side plot of Figure 8 on the error, we do see that the proposed method achieves the lowest error compared to the baselines. The non-optimality the reviewer thinks may come from the fact that in the left-hand side plot on the shape of $\gamma$ function, the optimized $\gamma$ does not have the highest probability compared to the baselines at all possible values of the support $l\in\{0, \dots, K\}$. This is because the error or utility used we optimize for is the average error across the support values. Hence, "optimality" refers to the best average utility across possible outcomes of the underlying mechanisms.
>
> Table 6 shows the results in the semi-supervised knowledge transfer, when the privacy allowance is $m = 1$. Note that the proposed approach achieves similar performance compared to the subsampling baseline in this case. Indeed, when $m=1$, subsampling is optimal in the pure DP case. We think it might also be optimal in the approximate DP case. Hence, we do not expect to see improvement when $m=1$.
>
> 2. See response to requested changes 1 below.
>
> 3. We discussed the limitations on scalability of going to a large $K$ in section 5 when we formulate the optimization problem.
> Scalability might not be a big issue since we expect $K$ to be modest in practice and one can pre-compute $\gamma$ and reuse it for all subsequent predictions.

---

> > ### Author Response · Authors · 2024-08-23
> > **Review Response (Cont'd)**
> >
> > Response to changes:
> >
> > 1. We have updated the draft to be more careful about "optimality" and "efficiency". We agree with the reviewer that though theoretically solving the proposed optimization problem (via a direct non-approximation approach) shall lead to an optimal utility, in practice, one needs to rely on an approximated version of the objective function in the optimization problem to make it computationally feasible. We hence loosen our optimality guarantee in practice by making all claims and discussion more precise, changing "optimal utility" to be "maximizing the utility" and "efficient" to be "tractable".
> >
> >
> > 2. We have moved the entire related work section from the Appendix to the main paper.
> >
> > 3. We added the following discussion about the optimality results of Randomized Response (RR) from two above mentioned work, [Holohan et al. 2016] and [Kairouz et al. 2015], in Section 2.1 on related work:
> >
> > [Holohan et al. 2016] and [Kairouz et al. 2015] show that the classical Randomized Response (RR) mechanism with a constant probability of faithfully revealing the true answer is optimal in certain private estimation problems.
> > Our proposed DaRRM framework and our problem setting is a generalized version of the ones considered in both [Holohan et al. 2016] and [Kairouz et al. 2015], which not only subsumes RR but also enables a data-dependent probability, or noise addition.
> >
> >
> > While RR with a constant probability can be shown optimal in problems such as private count queries or private estimation of trait possession in a population, it is not optimal in other problems, such as private majority ensembling, since unlike the former problems, changing one response of the underlying mechanisms does not necessarily change the output of the majority.
> > To explicitly compute the minimum amout of noise required, one needs the output distributions of the underlying mechanisms but this is unknown.
> > To resolve this, our proposed DaRRM framework adds the amount of noise dependent on the set of observed outcomes from the underlying private mechanisms, $\mathcal{S}$, which is a random variable of the dataset and is hence a proxy. This enables DaRRM to calibrate the amount of noise based on whether the majority output is likely to change. The amount of noise is automatically reduced when the majority output is not likely to change.
> >
> > Second, [Holohan et al. 2016] and [Kairouz et al. 2015] both consider a special case in our setting where all $K$ private mechanisms are i.i.d., while our approach focuses on the more general setting where each private mechanism can have a different output distribution.
> >
> > 4. We updated all results in the experiments section by using the optimal composition bound from [Kairouz et al. 2015] instead of the advanced composition bound.
> > Specifically, we updated the results in simulations (section 6.1, Appendix D.1) and in the application on private semi-supervised knowledge transfer (section 6.2 and Appendix D.2.2).
> > We observe no major difference in terms of the conclusion.
> >
> > Note in Section 4, Theorem 4.1 on provable privacy amplification, we focus on the pure differential privacy setting, i.e., $\delta = \Delta = 0$, and both advanced composition and the exact composition bound from [Kairouz et al. 2015] do not apply. Simple composition is indeed tight in this setting.
> >
> > 5. See the discussion in point 3.
> > Count query release with geometric mechanism is optimal for the general count query problem, which is different from majority ensembling. Changing the output of one underlying mechanism/teacher changes the aggregated output in count query, but it does not necessarily change the majority output.
> > Hence, the geometric mechanism, which again is randomized response with a constant probability, is likely not optimal in majority ensembling.
> >
> > 6. We have moved important DP related definitions from the Appendix to the main paper. The neighborhood relation here can be either arbitrary replacement or deletion. The only requirement is that the underlying $K$ mechanisms that vote for the majority need to satisfy the DP definition based on whichever neighborhood relation.
> > The privacy guarantee of the proposed algorithm will then be DP under the same neighborhood relation.

---

> > ### Comment · Reviewer_kUU3 · 2024-09-05
> > **Some further comments**
> >
> > Thanks for the updates and clarifications, and sorry for the delay. I still have some further questions/comments:
> >
> > * Optimality: In which cases does the subsampling baseline provide optimal utility? The question is asked on p7, paragraph 1 but I do not see any obvious answer. Please state this explicitly in the paper, and add a comment on this also to the experimental results.
> >
> > * Some minor details: can you switch the error plots in Sec6 to be something more immediately readable than single points to avoid confusion, e.g., barplots would be pretty standard. Also, add some legends to the result plots.

---

> > > ### Author Response · Authors · 2024-09-23
> > > **Further Review Response**
> > >
> > > Sorry for the late response.
> > >
> > > 1. In terms of optimality, we included a new proof on the lower bound of the error at $m=1$ in the updated draft, which applies to both pure and approximate DP settings. Please check out Lemma 3.2 and Appendix A.3 for a full proof. This lower bound implies that subsampling gives the optimal error when $m=1$. This is also reflected in the experiments. For example in Figure 2 and Figure 8, when $m=1$, the optimized $\gamma_{Opt}$ function indeed overlaps with the $\gamma_{Sub}$ function, which corresponds to subsampling, and they have the same error. We also made a comment on this in the experiment section in the main draft.
> > >
> > > 2. We have changed all error plots from scatter plots to barplots. The legends were below the plots in a separate plot. We now removed those and put the legends directly in the result plots. Please check out the updated version of the draft.
> > >
> > > Thank you.

---

> > > > ### Comment · Reviewer_kUU3 · 2024-09-25
> > > > **Final(?) comments**
> > > >
> > > > Thanks again for the updates, here are still some (mostly minor) comments:
> > > >
> > > > * Note that contrary to what you claim in your Thm2.3, Kairouz et al Thm 3.4 is *not* optimal (and this then leads to some weird statements, e.g., on when the optimal composition is worse than the basic composition in the footnote on p6, and some in Sec6, as well as in Appendix D.1.1). Can you please use the actual optimal composition (Thm3.3 in Kairouz et al) as there should not be any problems with the number of compositions you are doing.
> > > >
> > > > * On Table 6: I do not think claiming that your method achieves the highest accuracy is ok when it seems to lose/get at most comparable results compared to a baseline in several settings.
> > > >
> > > > * On Lemma 3.2: when you say "$g(\mathcal S)$ is the prob of the non-private.." do you mean $Pr(g(\mathcal S))$ is the prob. of the locally DP majority output being 1?
> > > >
> > > > * On the figs: thanks for updating these, but please revert to using legend under the fig; when making the comment I was looking at Fig10 which happened to be the only one without a legend. Apologies!

---

> > > > > ### Author Response · Authors · 2024-09-30
> > > > > **Response to Final(?) Comments**
> > > > >
> > > > > 1. Thm 3.3 in Kairouz et al. does not provide a closed-form expression for privacy loss, making it difficult to evaluate in practice. In contrast, Thm 3.4 in Kairouz et al. has a more straightforward, closed-form expression.
> > > > >
> > > > > Apologies for the confusing comment in the footnote on page 6; it was left over from the previous comparison between advanced and simple composition, and we neglected to remove it.
> > > > >
> > > > > We revisited Thm 3.4 in Kairouz et al. and compared it with simple and advanced composition. In the regime of interest (small $K$ and $m$), the privacy loss (i.e., total $\epsilon$) by Thm 3.4 matches that of simple composition, which is $m\epsilon$ under $m$-fold composition. In this case, one can indeed minimize $\delta$ by setting $\delta'$ (a hyperparameter in both optimal composition and advanced composition) to be 0, as the privacy loss does not depend on $\delta'$. By doing so, optimal composition yields a slightly lower $\delta$ than simple composition. Previously in Section D.1.1 before, we did not optimize this $\delta$. Now one can consistently rely on Thm 3.4 without needing simple composition or advanced composition. We have revised Sec 6 (experiments) accordingly and removed Section D.1.1 in the Appendix, as it is no longer relevant. Note that the optimized $\delta$ from optimal composition has minimal impact on the experiment results.
> > > > >
> > > > > 2. Sorry the caption of Table 6 was copied and pasted from the other tables.
> > > > > Since Table 6 represents the case where $m=1$ and subsampling is already optimal (as shown in Lemma 3.2), we have updated the caption to clarify that, as expected, there is no difference between subsampling and our optimized method in this scenario.
> > > > >
> > > > > 3. Yes, you are correct. The "non-private majority" might be a bit confusing. We now changed it to be "the true majority" in Lemma 3.2.
> > > > >
> > > > > 4. We reverted the previous setting using legend under the figs and added a new legend for Fig 10 (now Fig 9).
> > > > >
> > > > > Thank you for the comments.

---

> > > > > > ### Comment · Reviewer_kUU3 · 2024-10-09
> > > > > > **Thanks for the update again**
> > > > > >
> > > > > > To be clear, I think the current issue with accounting is not a major one, but it should be done somehow consistently and clearly. In general, doing the proofs using (non-optimal Thm3.4 in Kairouz et al.) bound is fine, since as you note, the actual tight bound is a bit nasty, as long as the paper does not claim optimality in accounting where it is not optimal. From your comment above I gather that these issues are now fixed (the draft has not been updated after the last comments, but please make sure that this is done; I give the recommendation assuming this is so):
> > > > > > * Thm2.3 (Thm3.4 in Kairouz et al) is not claimed to be optimal composition, and any reference to this thm also does not claim optimality.
> > > > > > * Clearly state which accounting the experiments use.

---

> > > > > > > ### Author Response · Authors · 2024-10-14
> > > > > > > **Response to the update**
> > > > > > >
> > > > > > > Instead of calling Thm 2.3 (Thm 3.4 in Kairouz et al.) "optimal composition", we now call it "tight composition". We added additional pointers in the experiment section to point to Thm 2.3 and hopefully made it clear that Thm 2.3 is the composition theorem used in the experiments.
> > > > > > >
> > > > > > > We should have updated the draft after the last comments. We just uploaded the latest version. Please let us know if you cannot see the most up-to-date version. Thank you.

---

> > > > > > > > ### Comment · Reviewer_kUU3 · 2024-10-14
> > > > > > > > **TIght and optimal are usually the same**
> > > > > > > >
> > > > > > > > Please do not call it that: tight accounting generally means optimal, i.e., Thm3.3 in Kairouz et al. is tight as in there is no slack in the accounting, Thm3.4 is not.

---

> > > > > > > > > ### Author Response · Authors · 2024-10-14
> > > > > > > > >
> > > > > > > > > Okay, we changed the reference to Thm 3.4 in Kairouz et al. to be "general composition" and have updated the draft accordingly. Thm 3.4 does not have an official name in Kairouz et al. Feel free to leave suggestions if you think some other name referring to this composition is better.

---

### Author Response · Authors · 2024-08-23
**Updated Version of the Draft**

Dear reviewers and AE,

We really appreciate all helpful feedback and comments from the reviewers.
We have uploaded a revised version of the draft, trying to incorporate all these comments. The revised places are colored in blue. Feel free to let us know if there is anything we could do to further improve the draft.

Thank you.

---

### Decision · Action_Editor_qwFS · 2024-10-18

**Recommendation:** Accept with minor revision

**Comment:**

While I think the paper is generally well written, I want to authors to provide some final clarifications to the camera ready, which I will list below.

- You mention that if there is a clear majority of outcomes (either 0s or 1s) in $\mathcal{S}$, then it should remain so even for an adjacent data set. Please clarify why that is in the final version. I would imagine this depends somehow on the individual mechanisms.
- Proof of Lemma 3.2 is not clear, and needs to be clarified for the final version (NOTE: since Lemma 3.2 is not a crucial part of the other theoretical contributions of this work, I don't see it as a breaking chance)
    * First of all, the Lemma 3.2 in the main paper is titled "Lower Bound on Error when $m=1$". However, the statement of the Lemma, neither in the main paper of appendix, specifies that the Lemma is specific to $m=1$ or even subsampled mechanism. Hence I read the Lemma as a general guarantee for $(\epsilon, \delta)$-DP aggregation algorithm $\mathcal{A}$. I believe the $m=1$ result just shows that the Lemma is tight in this case. If I'm correct in my assessment, please consider dropping the ".. when $m=1$" from the title of the Lemma.
    * In the beginning of the proof you seem to focus on setting where $p_i = p, \forall i \in [K]$, but this doesn't assumption does not seem to be generalized for the differing $p_i$ anywhere in the proof. It needs to be clarified if and why studying this case is sufficient for the general statement of the Lemma.
    * Next, you say "Consider a dataset $\mathcal{D}_0$ such that $\Pr[\mathcal{M}_i(\mathcal{D}_0)=1] = \frac{1}{2}$". Is there any guarantee that such data set exists for arbitrary mechanism $\mathcal{M}_i : \mathcal{D} \rightarrow \{0, 1\}$?
    * Regarding the way you choose $p$ in the 3rd paragraph: I'm not sure if such $L$ is guaranteed to exist, and depends on the $\epsilon$ value. E.g. if $\epsilon = 1$ then even $L = 1$ would set $p > 1$.
- Proof of Lemma 3.4: The last bit of the proof, eq. 45 implying 46, is bit challenging to follow. I wonder if it would help the exposition if you would write the $\alpha$s in eq. 50 as $\tilde{\alpha}$, corresponding to the $\Pr[\mathcal{L}^\beta = l]$. Since the eq. 45 would hold for all $\alpha$, the eq. 50 would hold for all $\tilde{\alpha}$ in the K-simplex, and therefore also for $\alpha$ and thus you would get your claim.
- While the paper is mainly theoretical, I agree with reviewer Fx2m that further experiments on different data sets would strengthen the claims. Instead of using a computationally expensive model, authors could e.g. employ a simple CNN model to evaluate their approach on SVHN data set as suggested by the reviewer.

Typos:
- Proof of Lemma 3.2:
    * 1st paragraph of the proof $\Pr[g(\mathcal{S}] = 1$ should be $\Pr[g(\mathcal{S}) = 1]$.
    * Should the first $\mathcal{A}(\mathcal{D}_0)$ after $\Pr[\mathcal{A}(\mathcal{D}_2)=1] \leq $ be $\mathcal{A}(\mathcal{D}_1)$?
- Step 3 of Alg. 1, lower case $k$ in subscript
- Proof of Lemma 3.4
    * Eq. 34 is has conditional probability with condition $\mathcal{L}(\mathcal{D}) = 1$, while I believe this should be $\mathcal{L}(\mathcal{D}) = l$. The same mistake is in all the proceeding equations.
    * Eq. 40 uses similarly wrong condition but now it is both parts of the law of total probability.
- "maixmizes"

**Audience:**

The problem studied in the paper is interesting and has practical importance, especially for the DP audience.

**Claims And Evidence:**

This paper proposes a novel method for aggregating responses of an ensemble of differentially private prediction mechanisms. Paper shows theoretically, that aggregating the majority vote based on a subsample of the predictions is equivalent to applying a randomized response mechanism on the majority of vote of all the predictions with a response probability relative to the size of the subsample (Lemma 3.1). Authors propose a new majority vote aggregation strategy (Alg. 1) that releases the majority based on RR with response probability parametrized with noise function $\gamma$. Authors further demonstrate a theoretical model on choosing the optimal noise function $\gamma$ for the problem.

Authors verify their theoretical findings in a simulated setting, where they show that the optimized noise function outperforms the subsampled variant. Furthermore, authors apply the method on a real world problem, where they show that again the optimized noise function outperforms the subsampled one as well as a previous majority voting algorithm.

In summary, the main contribution of the paper is the theoretical work which is supported by rigorous proofs. The empirical results further support the findings, though in somewhat limited setting.

---

> ### Author Response · Authors · 2024-11-23
> **Response to AE comments**
>
> We apologize for the delayed response. We have updated the draft to incorporate your feedback. Additionally, we have reverted the blue-colored text to black and disclosed the authors' identities in this final version.
>
> 1. "You mention that ..." This is because the mechanisms are differentially private by themselves, and so they cannot change their output by much on neighboring datasets by definition. We added additional clarification of this in the last paragraph on page 2.
>
> 2. "Proof of Lemma 3.2 ..."
>
> - No, Lemma 3.2 is specific to the $m=1$ case, as stated in the draft. This is because the probability of outputting 1 when subsampling $m=1$ teacher is exactly $\frac{1}{K}\sum_{i=1}^{K} p_i$; but this probability is different for other $m$'s. When $m=1$, the probability of picking the $i$-th mechanism is $\frac{1}{K}$ and then the majority is just the output of the $i$-th mechanism, which has a probability of $p_i$. However, when $m = 3$, for example, the probability of subsampling three mechanisms $i, j, k$, for $i, j, k \in [K]$, is $\frac{1}{{K \choose 3}}$. The output has a majority of 1, when two of the subsampled mechanisms output 1 or when all three subsampled mechanisms output 1. And so the probability of outputting a majority of 1 is $p_ip_jp_k + p_ip_j(1-p_k) + p_i(1-p_j)p_k + (1-p_i)p_jp_k$. This means the overall probability of outputting a majority of 1 when $m=3$ is $\frac{1}{{K \choose3}} \sum_{i,j,k} ( p_ip_jp_k + p_ip_j(1-p_k) + p_i(1-p_j)p_k + (1-p_i)p_jp_k )$, which is different from $\frac{1}{K}\sum_{i=1}^{K} p_i$. We added additional explanation in the paragraph before stating Lemma 3.2 on page 7 to make it clear that the lower bound only applies to the case when $m=1$.
>
> - When showing a lower bound, the objective is to identify a specific bad instance that results in a large error (e.g., a high TV distance in our case). This contrasts with proving an upper bound, which must apply to all cases.
>             The $p_i = p, \forall i \in [K]$ case is a special case in the more general setting where $p_i$'s can be different. Hence, a lower bound in the $p_i = p$ case also indicates a lower bound for the more general case. We clarified this point further in the proof of Lemma 3.2 in Appendix A.3.
>
> - Again, we do not claim that such a dataset $\mathcal{D}_0$ exists for any arbitrary mechanism. Rather, we construct a specific dataset $\mathcal{D}_0$ and $K$ teachers, such that $\Pr[M_i(\mathcal{D}_0) = 1]$, to serve as a bad instance. We made this more clear in the proof of Lemma 3.2 in Appendix A.3.
>
> - The idea is that as long as there exists at least one set of $(\epsilon, \delta, L)$ value satisfying the construction, the bad instance is non-void and the lower bound holds. This $L$ does not need to  exist for all $\epsilon, \delta$. In the original statement, we specified "$0\leq \epsilon < c$ for some constant $c > 0$", with the expectation that this $c$ is relatively small. To make this regime for $\epsilon$ more explicit, we have now clarified it in the lemma statement as $\epsilon \in (0, \frac{1}{2})$.
>
> 3. "Proof of Lemma 3.4 ..." We revised the wording in the proof on page 24 based on your suggestion.  Hope this makes the proof more clear.
>
> 4. We tried to train CNN and ResNet models on the SVHN dataset using DP-SGD in a private setting but got some problems. We found that this dataset is particularly difficult to train privately.
>     For $\epsilon < 1$,
>     the test accuracy of the teacher model for binary classification (using 1000 samples from class 5 and 1000 test samples from class 8, ensuring a baseline accuracy of 50\%) ranged between 55\% and 63\%,
>     and we have tried to tune different hyperparmeters in training, including batch size, clip norm and noise std., but ended up having similar results.
>     This low accuracy makes the teacher models highly noisy, and the 11 teachers we trained often do not agree with each other. This makes it a bad setting to demonstrate the effectiveness of the proposed DaRRM compared to the baselines, because in such cases a large amount of noise is required to achieve privacy guarantees.
>     As explained in the intuition from the introduction section (see our response to your first point), some level of consensus among the teachers is necessary to reduce noise and improve utility.
>
> 5. We fixed all typos as mentioned in the comment.